# DyGB: Dynamic Gradient Boosting Decision Trees with In-Place Updates for Efficient Data Addition and Deletion

## Abstract

Gradient Boosting Decision Tree (GBDT) is one of the most popular machine learning algorithm in various applications. However, in the traditional settings, all data should be simultaneously accessed in the training procedure: it does not allow to add or delete any data instances after training. In this paper, we propose DyGB (**Dy**namic **GB**DT), a novel framework that enables efficient support for both incremental and decremental learning within GBDT. To reduce the learning cost, we present a collection of optimizations for DyGB, so that it can add or delete a small fraction of data on the fly. We theoretically show the relationship between the hyper-parameters of the proposed optimizations, which enables trading off accuracy and cost on incremental and decremental learning. Empirical results on backdoor and membership inference attacks demonstrate that DyGB can effectively add and remove data from a well-trained model through incremental and decremental learning. Furthermore, experiments on public datasets validate the effectiveness and efficiency of the proposed DyGB framework and optimizations.

## 1 Introduction

Gradient Boosting Decision Tree (GBDT) has demonstrated outstanding performance across a wide range of applications (Sudakov et al., 2019; Biau et al., 2019; Rao et al., 2019; Liu & Yu, 2007). It outperforms deep learning models on many datasets in accuracy and provides interpretability for the trained models. In particular, GBDT has become the de facto choice for modeling tabular and categorical data, where it consistently achieves state-of-the-art performance. However, in traditional setting, all data is simultaneously accessed in training procedure, making its application limited. To address this, we introduce Dynamic Learning, which combines incremental learning (adding new data) (Tian et al., 2024; He, 2024) and decremental learning (removing outdated data) (Liu et al., 2025; Wang et al., 2024). This is essential for applications like recommender systems, which must dynamically adapt to the latest user behaviors (Wang et al., 2023; Shi et al., 2024).

**Incremental Learning.** There are some challenges for incremental learning in GBDT due to its natural properties (Friedman et al., 2000). Traditional GBDT trains over an entire dataset, and each node is trained on the data reaching it to achieve the best split for optimal accuracy. Adding unseen data may affect node splitting results, leading to catastrophic performance changes. Moreover, training gradient boosting models involves creating trees for each iteration, with tree fitting based on the residual of previous iterations. More iterations create more trees, increasing model sizes and hurting inference throughput. This also prohibits tasks like fine-tuning or transfer learning without substantially increasing model sizes.

Recent studies have explored incremental learning on random forest (RF), and gradient boosting (GB). Wang et al. (2009) presented an incremental random forest for online learning with small streaming data. Beygelzimer et al. (2015a) extended gradient boosting theory for regression to online learning. Zhang et al. (2019) proposed iGBDT for incremental learning by "lazily" updating, but it may require retraining many trees when the new data size is large. It is important to note that prior studies on online gradient boosting (Beygelzimer et al., 2015a; Chen et al., 2012; Beygelzimer et al., 2015b) and incremental gradient boosting (Zhang et al., 2019; Hu et al., 2017) do not support decremental learning.

**Algorithm 1** Robust LogitBoost Algorithm.

1: $F_{i,k} = 0, p_{i,k} = \frac{1}{K}, k = 0$ to $K-1, i = 1$ to $N$
2: **for** $m = 0$ to $M-1$ **do**
3:    **for** $k = 0$ to $K-1$ **do**
4:      $\hat{D}_{tr} = \{r_{i,k} - p_{i,k}, \ \mathbf{x}_i\}_{i=1}^N$
5:      $w_{i,k} = p_{i,k}(1 - p_{i,k})$
6:      $\{R_{j,k,m}\}_{j=1}^J = J$-terminal node regression tree from $\hat{D}_{tr}$, with weights $w_{i,k}$, using the tree split gain formula Eq. equation 5.
7:      $\beta_{j,k,m} = \frac{K-1}{K} \frac{\sum_{\mathbf{x}_i \in R_{j,k,m}} r_{i,k} - p_{i,k}}{\sum_{\mathbf{x}_i \in R_{j,k,m}} (1 - p_{i,k}) p_{i,k}}$
8:      $f_{i,k} = \sum_{j=1}^J \beta_{j,k,m} \mathbf{1}_{\mathbf{x}_i \in R_{j,k,m}}, \quad F_{i,k} = F_{i,k} + \nu f_{i,k}$
9:    **end for**
10:   $p_{i,k} = \exp(F_{i,k}) / \sum_{s=1}^K \exp(F_{i,s})$
11: **end for**

**Algorithm 2** Online Learning in Gradient Boosting

1: $D' = D_{in}$ **if** incremental learning **else** $D_{de}$
2: **for** $m = 0$ to $M-1$ **do**
3:    **for** $k = 0$ to $K-1$ **do**
4:      Calculate $p_{i,k}, r_{i,k}, g_{i,k} = r_{i,k} - p_{i,k}$, and $w_{i,k} = p_{i,k}(1 - p_{i,k})$ for $\mathbf{x}_i \in D'$
5:      Define $\hat{D}' = \{\mathbf{x}_i, g_{i,k}, w_{i,k}\}_{i=1}^{|D'|}$
6:      **if** incremental learning **then**
7:        $\left\{\hat{R}_{j,k,m}\right\}_{j=1}^J = \text{incr}(\{R_{j,k,m}\}_{j=1}^J, \hat{D}')$
8:      **else**
9:        $\left\{\hat{R}_{j,k,m}\right\}_{j=1}^J = \text{decr}(\{R_{j,k,m}\}_{j=1}^J, \hat{D}')$
10:      **end if**
11:      Update $F_{i,k}$ with $\left\{\hat{R}_{j,k,m}\right\}_{j=1}^J$
12:    **end for**
13: **end for**

**Decremental Learning.** Decremental learning is more complex and less studied than incremental learning (Dilworth, 2025). Wang et al. (2025) and Brophy & Lowd (2020) provided dynamic methods for data addition and removal in RF. While dynamic learning has emerged as a popular topic recently, it has been barely investigated on GBDT. Wu et al. (2023); Lin et al. (2023) are among the latest studies in decremental learning for GBDT. Wu et al. (2023) presented DeltaBoost, a GBDT-like model enabling data deletion. DeltaBoost divides the training dataset into disjoint sub-datasets, training each iteration's tree on a different sub-dataset, reducing the inter-dependency of trees. However, this simplification may impact model performance. Lin et al. (2023) proposed an unlearning framework in GBDT without simplification, unlearning specific data using recorded auxiliary information from training. It optimizes to reduce unlearning time, making it faster than retraining from scratch, but introduces many hyper-parameters and performs poorly on extremely large datasets.

In this paper, we propose DyGB (**Dy**namic **GB**DT), an efficient framework for both incremental and decremental learning in GBDT. To the best of our knowledge, DyGB is the first approach to support in-place learning for both adding and removing data in GBDT models. Furthermore, DyGB introduces a unified mechanism for incremental and decremental updates for efficient implementation.

Unlike incremental-only methods such as iGBDT (Zhang et al., 2019), DyGB supports both data addition and removal within a unified framework. Furthermore, compared to DeltaBoost Wu et al. (2023), which facilitates unlearning by restricting each tree's training to disjoint data partitions, DyGB performs in-place updates on the complete model structure without data partitioning. This ensures that the model maintains high functional similarity to a model retrained from scratch, avoiding structural constraints that may compromise predictive performance.

**Contributions.** (1) We introduce DyGB, an efficient in-place dynamic learning framework for gradient boosting models supporting incremental and decremental learning. (2) We present optimizations to reduce the cost of incremental and decremental learning, making adding or deleting a small data fraction substantially faster than retraining. (3) We theoretically show the relationship among optimization hyper-parameters, enabling trade-offs between accuracy and cost. (4) We experimentally evaluate DyGB on public datasets, confirming its effectiveness and efficiency. (5) We release an open-source implementation of DyGB[1].

## 2 DYNAMIC GBDT FRAMEWORK

### 2.1 GBDT PRELIMINARY

GBDT is an powerful ensemble technique that combines multiple decision tree to produce an accurate predictive model (Friedman et al., 2000; Friedman, 2001). Given a dataset $D_{tr} = \{y_i, \mathbf{x}_i\}_{i=1}^N$, where $N$ is the size of training dataset, and $\mathbf{x}_i$ indicates the $i^{th}$ data vector and $y_i \in \{0, 1, ..., K-1\}$ denotes the label for the $i^{th}$ data point. For a GBDT model with $M$ iteration, the probability $p_{i,k}$ for $i^{th}$ data and class $k$ is:

$$p_{i,k} = \mathbf{Pr}\left(y_i = k | \mathbf{x}_i\right) = \frac{e^{F_{i,k}(\mathbf{x_i})}}{\sum_{s=1}^K e^{F_{i,s}(\mathbf{x_i})}}, \qquad i = 1, 2, ..., N \tag{1}$$

where $F$ is a combination of M terms:

$$F^{(M)}(\mathbf{x}) = \sum_{m=0}^{M-1} \rho_m h(\mathbf{x}; \mathbf{a}_m) \tag{2}$$

---

[1] https://anonymous.4open.science/r/DyGB

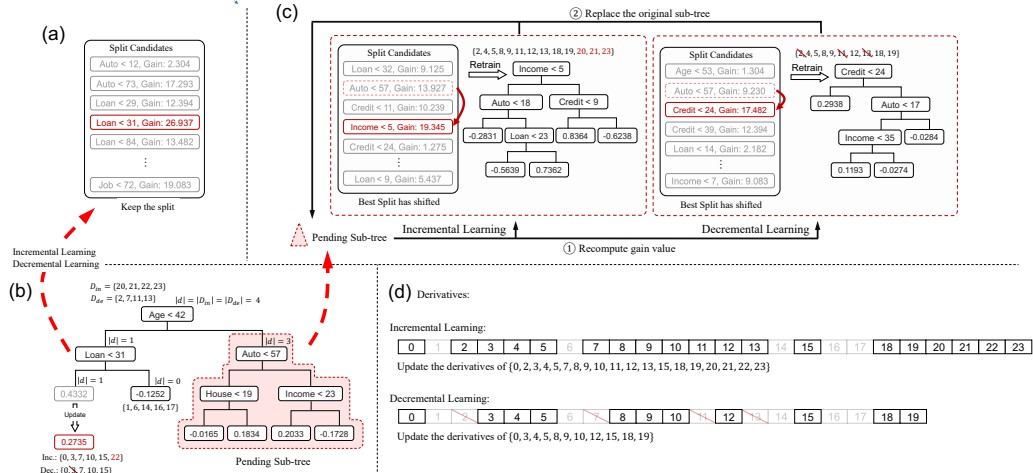

Figure 1: An example for the incremental learning and decremental learning procedure in DyGB. (a) For the node of `Loan < 31`, the current split is still the best after dynamic learning. Thus, the split does not need to change. (b) An already well-trained tree in $D_{tr}$. (c) For the node of `Auto < 57`, the best split has shifted after dynamic learning. (d) Incremental update for derivatives – only update the derivatives for those data reaching the changed terminal nodes.

where $h(\mathbf{x}; \mathbf{a}_m)$ is a regression tree, and $\rho_m$ and $\mathbf{a}_m$ denote the tree parameters that learned by minimizing the *negative log-likelihood*:

$$L = \sum_{i=1}^{N} L_i, \qquad L_i = -\sum_{k=0}^{K-1} r_{i,k} \log p_{i,k} \tag{3}$$

where $r_{i,k} = \begin{cases} 1, & \text{if } y_i = k \\ 0, & \text{otherwise} \end{cases}$. The training procedures require calculating the derivatives of loss function $L$ with respect to $F_{i,k}$:

$$g_{i,k} = \frac{\partial L_i}{\partial F_{i,k}} = -\left(r_{i,k} - p_{i,k}\right), \quad h_{i,k} = \frac{\partial^2 L_i}{\partial F_{i,k}^2} = p_{i,k}\left(1 - p_{i,k}\right). \tag{4}$$

In GBDT training, to solve numerical instability problem (Friedman et al., 2000; Friedman, 2001; Friedman et al., 2008), we apply **Robust LogitBoost** algorithm (Li, 2010) as shown in Algorithm 1, which has three parameters, the number of terminal nodes $J$, the shrinkage $\nu$ and the number of boosting iterations $M$. To find the optimal split for a decision tree node, we first sort the $N$ data by the feature values being considered for splitting. We then iterate through each potential split index $s$, where $1 \leq s < N$, to find the best split that minimizes the weighted squared error (SE) between the predicted and true labels. Specifically, we aim to find an split $s$ to maximize the gain function:

$$Gain(s) = \frac{\left(\sum_{i=1}^{s} g_{i,k}\right)^2}{\sum_{i=1}^{s} h_{i,k}} + \frac{\left(\sum_{i=s+1}^{N} g_{i,k}\right)^2}{\sum_{i=s+1}^{N} h_{i,k}} - \frac{\left(\sum_{i=1}^{N} g_{i,k}\right)^2}{\sum_{i=1}^{N} h_{i,k}}. \tag{5}$$

## 2.2 PROBLEM SETTING

For classic GBDT, all training data must be loaded during training, and adding/deleting instances is not allowed afterwards. This work proposes DyGB, enabling in-place addition/deletion of specific data instances to/from a trained model through incremental/decremental learning.

**Problem Statement.** Given a trained gradient boosting model $T(\theta)$ on training dataset $D_{tr}$, where $\theta$ indicates the parameters of model $T$, an incremental learning dataset $D_{in}$, and/or a decremental learning dataset $D_{de}$ ($D_{de} \subseteq D_{tr}$), our goal is to find a tree model $T(\theta')$ that fits dataset $D_{tr} \cup D_{in} \setminus D_{de}$, where $|\theta| = |\theta'|$ (the parameter size and the number of trees stay unchanged).

**Problem Statement.** Given a trained gradient boosting model $T(\theta)$ on training dataset $D_{tr}$, where $\theta$ indicates the parameters of model $T$, an incremental learning dataset $D_{in}$, and/or a decremental learning dataset $D_{de}$ ($D_{de} \subseteq D_{tr}$), our goal is to find a tree model $T(\theta')$ that fits dataset $D_{tr} \cup D_{in} \setminus D_{de}$, where $|\theta| = |\theta'|$ (the parameter size and the number of trees stay unchanged). We assume

the original training dataset $D_{tr}$ remains available (retrievable) and accessible by the algorithm for potential use in conditional subtree retraining.

The most obvious way is to retrain the model from scratch on dataset $D_{tr} \cup D_{in} \setminus D_{de}$. However, retraining is time-consuming and resource-intensive. Especially for dynamic learning applications, rapid retraining is not practical. The key question of this problem is: ***Can we obtain the model $T(\theta')$ based on the learned knowledge of the original model $T(\theta)$ without retraining the entire model?***

The proposed DyGB aims to find a tree model $T(\theta')$ as close to the model retraining from scratch as possible based on the learned knowledge of the model $T(\theta)$. In addition, this dynamic learning algorithm is in a "warm-start" manner, because it learns a new dataset $D_{in}$ or removes a learned sub-dataset $D_{de} \subseteq D_{tr}$ on a model that is already well-trained on training dataset $D_{tr}$.

Let $\mathcal{A}$ denotes the initial GBDT learning algorithm , then we have $\mathcal{A}(D_{tr}) \in \mathcal{H}$, where $\mathcal{H}$ is the hypothesis space. A dynamic learning algorithm $\mathcal{L}$ for incremental learning or decremental learning can be used to learn dataset $D_{in}$ or remove dataset $D_{de} \subseteq D_{tr}$.

### 2.3 DyGB: Framework Overview

The goal of this work is to propose a dynamic GBDT framework that supports incremental and decremental learning for any collection of data.

**Dynamic Learning in GBDT.** The Algorithm 2 shows the dynamic learning procedure in GBDT. At first, the GBDT model is a well-trained model on the training dataset $D_{tr}$. Recall that the GBDT model is frozen and can not be changed after training—no training data modification. In this proposed framework, the user can do (1) incremental learning: update a new dataset $D_{in}$ to the model, and (2) decremental learning: remove a learned dataset $D_{de} \subseteq D_{tr}$ and its effect on the model.

---

**Algorithm 3** Incremental / Decremental Learning on One Tree

---

**Require:** Tree nodes $\{R_{j,k,m}\}_{j=1}^{J}$, updated data $D' = \{(\mathbf{x}_i, y_i)\}_{i=1}^{|D'|}$

1: **for** each non-terminal node $t$ in $\{R_{j,k,m}\}_{j=1}^{J}$ with ascending depths **do**
2:     Construct residual set $\widehat{D}' = \{(r_{i,k} - p_{i,k}, \mathbf{x}_i)\}_{i=1}^{|D'|}$
3:     $s \leftarrow$ current split of node $t$
4:     $s' \leftarrow$ best split by Eq. equation 5 using $(r_{i,k}, w_{i,k})$ after adding/removing $\widehat{D}'$
5:     **if** $s' \neq s$ **then**
6:         Retrain the subtree rooted at $t$
7:     **end if**
8: **end for**
9: Update prediction values $\{\beta_{j,k,m}\}_{j=1}^{J}$ for all terminal nodes

---

As shown in Algorithm 2, it is similar to the learning process, but it only needs to compute $r_{i,k}$ and $p_{i,k}(1 - p_{i,k})$ for target dataset $D'$ without touching the training dataset $D_{tr}$. Then, it will call the function of incremental learning or decremental learning to obtain $\left\{\hat{R}_{j,k,m}\right\}_{j=1}^{J}$. Finally, we update $F_{i,k}$ with new $\left\{\hat{R}_{j,k,m}\right\}_{j=1}^{J}$. Here we use the same notion to design the function of incremental learning and decremental learning – decremental learning is the inverse process of incremental learning for dataset $D'$. Therefore, we describe them in the Algorithm 3 at the same time.

**Incremental & Decremental Learning on One Tree.** Algorithm 3 describes the detailed process for incremental and decremental learning, which are almost the same as decremental learning is the inverse of incremental learning for dataset $D'$. The main difference is at Line 3. First, we traverse all non-terminal nodes layer by layer from root to leaves. For each node, let $s$ denote the current split. We recompute the new best gain value with $r_{i,k}$ and $p_{i,k}(1 - p_{i,k})$ after adding $D'$ for incremental learning or removing $D'$ for decremental learning. If the current split $s$ matches the new best split $s'$ (after adding/removing $D'$), we keep the current split (Figure 1(a)). Otherwise, if the current best split has changed ($s \neq s'$, Figure 1(c)), we retrain the sub-tree rooted on this node and replace it with the new sub-tree. After testing all nodes, node splits remain on the best split. Finally, we recompute the prediction value on all terminal nodes. Appendix E provides a detailed explanation of Figure 1.

## 3 Optimizing Learning Time

In this section, we introduce optimizations for the proposed DyGB to reduce computation overhead and costs. The key step is deciding whether a node should be kept or replaced: *Can we design an algorithm to quickly test whether the node should be retained or retrained without touching the*

*training data?* Our most important optimization is to avoid touching the full training dataset. We apply incremental update and split candidates sampling concepts from (Lin et al., 2023), extend them to support dynamic learning, and provide evidence of the relationship between hyper-parameters of different optimizations, enabling trade-offs between accuracy and cost. Additionally, we design optimizations specific to DyGB: 1) adaptive lazy update for residuals and hessians to decrease dynamic learning time; 2) adaptive split robustness tolerance to reduce the number of retrained nodes.

### 3.1 UPDATE WITHOUT TOUCHING TRAINING DATA

To reduce computation overhead and dynamic learning time, we target to avoid touching the original training dataset $D$, and only focus on the dynamic learning dataset $D'$. Following the study (Lin et al., 2023), we extend the optimization of updating statistical information to the scenarios of dynamic learning: (1) Maintain Best Split; (2) Recomputing Prediction Value; (3) Incremental Update for Derivatives, and the computation cost is reduced from $O(D \pm D')$ to $O(D')$ by these optimizations. The implementation of these optimizations are included in Appendix G.

### 3.2 ADAPTIVE LAZY UPDATE FOR DERIVATIVES

Although incremental update can substantially reduce dynamic learning time, we can take it a step further: if no retraining occurs, the changes to the derivatives will be very small. *How can we effectively utilize the parameters already learned to reduce dynamic learning time?*

Gradient Accumulation (Li et al., 2014; Goyal et al., 2017; Ruder, 2016) is widely used in DNN training. After computing the loss and gradients for each mini-batch, the system accumulates these gradients over multiple batches instead of updating the model parameters immediately. Inspired by this techniques, we introduce an adaptive lazy update for the proposed DyGB. Unlike Lin et al. (2023), which perform updates after a fixed number of batches, we update the derivatives only when retraining occurs. This approach uses more outdated derivatives for gain computation but significantly reduces the cost of derivative updates.

### 3.3 SPLIT CANDIDATES SAMPLING

From the above optimizations, if retraining is not required, we can keep the current best split. In this case, we only need to iterate over the dynamic learning dataset $D'$ and update the prediction values to accomplish dynamic learning, whether it involves adding or removing data. However, if the sub-tree rooted in this node requires retraining, it is necessary to train the new sub-tree on the data from the dataset $D_{tr} \pm D'$ that reaches this node. It is clear that retraining incurs more resource consumption and takes a longer execution time. In the worst case, if retraining is required in the root node, it has to retrain the entire new tree on full dataset $D_{tr} \pm D'$.

To reduce time and resource consumption of dynamic learning, a straightforward approach is to minimize retraining frequency. Therefore, we introduce split candidate sampling to reduce frequent retraining by limiting the number of splits, benefiting both training and dynamic learning. All features are discretized into integers in $0, 1, \cdots, B - 1$, as shown in Appendix C. This discretization process constructs the feature histograms, which is the foundational histogram-based implementation used in high-performance libraries like LightGBM (Ke et al., 2017) and the ABC-Boost (Li, 2010) framework. The original training procedure enumerates all $B$ potential splits, then obtains the best split with the greatest gain value. In split candidates sampling, we randomly select $\lceil \alpha B \rceil$ splits as candidates and only perform gain computing on these candidates. As $\alpha$ decreases, the number of split candidates decreases, resulting in larger distances between split candidates. Consequently, the best split is less likely to change.

**Definition 1** (Distance Robust). *Let $s$ be the current best split, and let $\lambda = |D'|/|D_{\mathrm{tr}}|$ denote the dynamic update ratio. Let $N_\Delta = \|t - s\|$ be the distance between $s$ and its nearest competing split $t$ under the same feature. We say that $s$ is* distance robust *if*

$$N_\Delta > \lambda \, Gain(s) \, C_s, \tag{6}$$

*where the node structural coefficient $C_s$ is defined as*

$$C_s := \left( \frac{1}{N_{ls}} \frac{\left(\sum_{\mathbf{x}_i \in l_s} g_{i,k}\right)^2}{\sum_{\mathbf{x}_i \in l_s} h_{i,k}} + \frac{1}{N_{rs}} \frac{\left(\sum_{\mathbf{x}_i \in r_s} g_{i,k}\right)^2}{\sum_{\mathbf{x}_i \in r_s} h_{i,k}} \right)^{-1}. \tag{7}$$

Here, $l_s$ and $r_s$ denote the left and right child nodes of split $s$, with sizes $N_{l_s} = |l_s|$ and $N_{r_s} = |r_s|$, respectively. This definition implies that the expected distance $\mathbb{E}[N_\Delta] = 1/\alpha$ under split sampling rate $\alpha$. Hence a smaller sampling rate enlarges $N_\Delta$ and makes the split more robust, reducing the probability that the best split changes after dynamic updates. The same conclusion holds for incremental updates.

**Definition 2** (Robustness Split) *For a best split $s$ and an arbitrary split $t, t \neq s$, and dynamic learning data rate $\frac{|D'|}{|D_{tr}|} = \lambda$, the best split $s$ is robust split if*

$$Gain(s) > \frac{1}{1 - \lambda} Gain(t) \tag{8}$$

Robustness split shows that, as $\lambda = \frac{|D'|}{|D_{tr}|}$ decreases, the splits are more robust, decreasing the frequency of retraining. In conclusion, decreasing either $\alpha$ or $\lambda$ makes the split more robust, reducing the change occurrence in the best split, and it can significantly reduce the dynamic learning time. We provide the proof of *Distance Robust* and *Robustness Split* in Appendix F.

### 3.4 Adaptive Split Robustness Tolerance

Recall the retraining condition for a node that we mentioned previously: we retrain the sub-tree rooted at a node if the best split changes. Although the best split may have changed to another one, the gain value might only be slightly different from the original best split. We show the observation of the distance of best split changes (the changes in the ranking of the best split) in Figure 2. The top row illustrates the distance of best split changes observed in the Adult and Covtype datasets for incremental learning, while the bottom row depicts same in Letter and SUSY datasets for decremental learning. Similar patterns are observed across various other datasets.

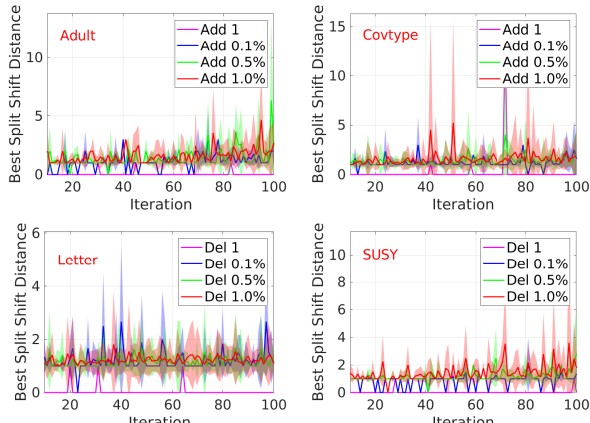

Figure 2: Observation of distance of best split changes. The lines represents the average changes of best split distance, and the shaded region is the standard error.

Prior theoretical work, such as Delta-Boost (Wu et al., 2023), has established that the best split often remains invariant under single-point deletion. However, our empirical results in Figure 2 extend this insight in two critical ways that distinguish DyGB from previous methods: (1) We observe this stability holds equally for incremental learning (data addition) involving unseen distributions; (2) Crucially, even when the invariance is broken (e.g., as $|D'|$ increases to 1%), Figure 2 quantifies that the split index typically shifts only slightly to the 2nd or 3rd best rank rather than jumping randomly.

This specific empirical observation, namely that split shifts are quantifiable and small, is the fundamental basis for our Adaptive Split Robustness Tolerance ($\sigma$). Unlike methods that trigger retraining immediately upon any split change, Figure 2 justifies that we can safely tolerate these minor rank shifts. To further validate this design, we explicitly examine the impact of sub-optimal splits on model performance in Appendix T, confirming that this tolerance mechanism maintains high accuracy. The distance of the best split changes is usually small. Tolerating its variation within a certain range and continuing to use the original split significantly accelerates dynamic learning. We propose adaptive split robustness tolerance: for a node with $\lceil \alpha B \rceil$ potential splits, if the current split is among the top $\lceil \sigma \alpha B \rceil$, we continue using it, where $\sigma$ ($0 \leq \sigma \leq 1$) is the robustness tolerance. $\sigma = 0$ selects only the best split, while $\sigma = 1$ avoids retraining. Higher $\sigma$ indicates greater tolerance, making the split robust and less likely to retrain. We recommend setting $\sigma$ to around 0.1.

## 4 Experimental Evaluation

In this section, we compare 1) our incremental learning with OnlineGB (onl, 2014; Leistner et al., 2009) and iGBDT (Zhang et al., 2019); 2) decremental learning with DeltaBoost (Wu et al., 2023)

Table 1.1: Total **incremental learning** time (seconds) and speedup v.s. baselines.

| Dataset | $|D'|$ | Total Time (Seconds) | | | Speedup v.s. | | | | | |
|---|---|---|---|---|---|---|---|---|---|---|
| | | OnlineGB | iGBDT | Ours | OnlineGB | iGBDT | XGBoost | LightGBM | CatBoost | ThunderGBM (GPU) |
| Adult | 1 | 0.265 | 0.595 | 0.035 ± 0.001 | 7.6x | 17.0x | 39.4x | 23.3x | 43.8x | 8.9x |
| | 0.1% | 9.020 | 1.145 | 0.105 ± 0.003 | 85.9x | 10.9x | 13.1x | 7.8x | 14.6x | 3.0x |
| | 0.5% | 44.650 | 1.296 | 0.215 ± 0.003 | 207.7x | 6.0x | 6.4x | 3.8x | 7.1x | 1.4x |
| | 1% | 98.000 | 1.573 | 0.340 ± 0.013 | 288.2x | 4.6x | 4.1x | 2.4x | 4.5x | 0.9x |
| CreditInfo | 1 | 29.000 | 0.475 | 0.111 ± 0.008 | 261.3x | 4.3x | 22.7x | 18.8x | 31.1x | 2,185.9x |
| | 0.1% | 3,386 | 1.391 | 0.245 ± 0.012 | 13,821.4x | 5.7x | 10.3x | 8.5x | 14.1x | 990.4x |
| | 0.5% | 28,875 | 1.428 | 0.336 ± 0.010 | 85,937.5x | 4.3x | 7.5x | 6.2x | 10.3x | 722.1x |
| | 1% | 336,000 | 1.568 | 0.384 ± 0.026 | 875,000.0x | 4.1x | 6.6x | 5.4x | 9.0x | 631.9x |
| SUSY | 1 | OOM | 12.037 | 1.683 ± 0.033 | - | 7.2x | 56.8x | 62.3x | 64.7x | 3.0x |
| | 0.1% | OOM | 53.460 | 7.990 ± 0.182 | - | 6.7x | 12.0x | 13.1x | 13.6x | 0.6x |
| | 0.5% | OOM | 55.380 | 13.428 ± 0.410 | - | 4.1x | 7.1x | 7.8x | 8.1x | 0.4x |
| | 1% | OOM | 57.680 | 20.238 ± 0.610 | - | 2.9x | 4.7x | 5.2x | 5.4x | 0.2x |
| HIGGS | 1 | OOM | 45.250 | 5.495 ± 0.235 | - | 8.2x | 40.9x | 44.4x | 55.2x | 2.3x |
| | 0.1% | OOM | 132.460 | 26.651 ± 0.779 | - | 5.0x | 8.4x | 9.2x | 11.4x | 0.5x |
| | 0.5% | OOM | 165.340 | 43.383 ± 1.621 | - | 3.8x | 5.2x | 5.6x | 7.0x | 0.3x |
| | 1% | OOM | 171.160 | 66.961 ± 1.463 | - | 2.6x | 3.4x | 3.6x | 4.5x | 0.2x |
| Optdigits | 1 | 0.032 | 0.174 | 0.011 ± 0.002 | 2.9x | 15.8x | 87.7x | 86.4x | 16.1x | 234.9x |
| | 0.1% | 0.091 | 0.181 | 0.015 ± 0.003 | 6.1x | 12.1x | 64.3x | 63.3x | 11.8x | 172.3x |
| | 0.5% | 0.559 | 0.191 | 0.029 ± 0.007 | 19.3x | 6.6x | 33.3x | 32.8x | 6.1x | 89.1x |
| | 1% | 1.403 | 0.196 | 0.044 ± 0.011 | 31.9x | 4.5x | 21.9x | 21.6x | 4.0x | 58.7x |
| Pendigits | 1 | 0.014 | 0.181 | 0.015 ± 0.003 | 0.9x | 12.1x | 58.7x | 77.6x | 12.2x | 190.2x |
| | 0.1% | 0.082 | 0.224 | 0.027 ± 0.006 | 3.0x | 8.3x | 32.6x | 43.1x | 6.8x | 105.7x |
| | 0.5% | 0.427 | 0.234 | 0.043 ± 0.009 | 9.9x | 5.4x | 20.5x | 27.1x | 4.3x | 66.3x |
| | 1% | 0.820 | 0.235 | 0.054 ± 0.011 | 15.2x | 4.4x | 16.3x | 21.6x | 3.4x | 52.8x |
| Letter | 1 | 0.033 | 0.102 | 0.017 ± 0.002 | 1.9x | 6.0x | 241.9x | 249.3x | 13.6x | 381.2x |
| | 0.1% | 0.551 | 0.167 | 0.039 ± 0.008 | 14.1x | 4.3x | 105.5x | 108.7x | 5.9x | 166.2x |
| | 0.5% | 2.768 | 0.187 | 0.070 ± 0.009 | 39.5x | 2.7x | 58.8x | 60.5x | 3.3x | 92.6x |
| | 1% | 5.680 | 0.201 | 0.124 ± 0.014 | 45.8x | 1.6x | 33.2x | 34.2x | 1.9x | 52.3x |
| Covtype | 1 | 0.090 | 1.321 | 0.290 ± 0.025 | 0.3x | 4.6x | 115.1x | 64.0x | 21.2x | 10.6x |
| | 0.1% | 21.408 | 6.391 | 0.658 ± 0.056 | 32.5x | 9.7x | 50.7x | 28.2x | 9.3x | 4.7x |
| | 0.5% | 105.688 | 7.765 | 1.051 ± 0.044 | 100.6x | 7.4x | 31.8x | 17.7x | 5.8x | 2.9x |
| | 1% | 214.188 | 8.088 | 1.732 ± 0.094 | 123.7x | 4.7x | 19.3x | 10.7x | 3.5x | 1.8x |
| Abalone | 1 | 0.013 | 0.331 | 0.028 ± 0.005 | 0.5x | 11.8x | 2.6x | 1.9x | 19.0x | 38.9x |
| | 0.1% | 0.026 | 0.356 | 0.029 ± 0.006 | 0.9x | 12.3x | 2.5x | 1.8x | 18.4x | 37.6x |
| | 0.5% | 0.170 | 0.338 | 0.050 ± 0.007 | 3.4x | 6.8x | 1.5x | 1.1x | 10.7x | 21.8x |
| | 1% | 0.354 | 0.366 | 0.054 ± 0.009 | 6.6x | 6.8x | 1.4x | 1.0x | 9.9x | 20.2x |
| WineQuality | 1 | 0.014 | 0.239 | 0.018 ± 0.002 | 0.8x | 13.3x | 5.9x | 4.0x | 47.7x | 44.1x |
| | 0.1% | 0.057 | 0.262 | 0.027 ± 0.005 | 2.1x | 9.7x | 3.9x | 2.7x | 31.8x | 29.4x |
| | 0.5% | 0.296 | 0.282 | 0.044 ± 0.007 | 6.7x | 6.4x | 2.4x | 1.6x | 19.5x | 18.0x |
| | 1% | 0.608 | 0.276 | 0.053 ± 0.008 | 11.5x | 5.2x | 2.0x | 1.4x | 16.2x | 15.0x |

and MUinGBDT (Lin et al., 2023); 3) training cost with popular GBDT libraries, such as XG-Boost (Chen & Guestrin, 2016), LightGBM (Ke et al., 2017), CatBoost (Dorogush et al., 2018) and ThunderGBM (Wen et al., 2020).

**Implementation Details.** Experimental settings are detailed in Appendix D. We employ one thread for all experiments to have a fair comparison, and run ThunderGBM on a NVIDIA A100 40GB GPU, since it does not support only CPU (thu, 2018). Unless explicitly stated otherwise, our default parameter settings are: $M = 100$, $J = 20$, $B = 1024$, $|D'| = 0.1\% \times |D_{tr}|$, $\alpha = 0.1$, and $\sigma = 0.1$.

**Datasets.** We utilize 10 public datasets in the experiments. The specifications of these datasets are presented in Table 2. The smallest dataset, Optdigits, consists of 3,822 training instances, while the largest dataset, HIGGS, contains a total of 11 million instances. The number of dimensions or features varies between 8 and 87 across the datasets.

Table 2: Dataset specifications.

| Dataset | # Train | # Test | # Dim | # Class |
|---|---|---|---|---|
| Adult | 36,139 | 9,034 | 87 | 2 |
| CreditInfo | 105,000 | 45,000 | 10 | 2 |
| SUSY | 2,500,000 | 2,500,000 | 18 | 2 |
| HIGGS | 5,500,000 | 5,500,000 | 28 | 2 |
| Optdigits | 3,822 | 1,796 | 64 | 10 |
| Pendigits | 7,493 | 3,497 | 16 | 10 |
| Letter | 15,000 | 5,000 | 16 | 26 |
| Covtype | 290,506 | 290,506 | 54 | 7 |
| Abalone | 2,785 | 1,392 | 8 | Reg. |
| WineQuality | 4,332 | 2,165 | 12 | Reg. |

## 4.1 TRAINING TIME AND MEMORY OVERHEAD

Since the proposed DyGB stores statistical information during training, this impacts both the training time and memory usage. Table 3 presents a report of the total training time and memory overhead.

**Training Time.** Table 3 shows the total training time across methods. DyGB is much faster than OnlineGB, DeltaBoost, and XGBoost, and slightly slower than iGBDT. While slower than Light-GBM on smaller datasets, it outperforms on larger ones like SUSY and HIGGS, with training times

Table 1.2: Total **decremental learning** time (seconds) and speedup v.s. baselines.

| Dataset | $\|D'\|$ | Total Time (Seconds) | | | Speedup v.s. | | | | | |
|---|---|---|---|---|---|---|---|---|---|---|
| | | DeltaBoost | MUinGBDT | Ours | DeltaBoost | MUinGBDT | XGBoost | LightGBM | CatBoost | ThunderGBM (GPU) |
| Adult | 1 | 1.159 | 0.217 | 0.034 ± 0.001 | 34.1x | 6.4x | 40.6x | 24.0x | 45.1x | 9.1x |
| | 0.1% | 2.860 | 0.751 | 0.104 ± 0.003 | 27.5x | 7.2x | 13.3x | 7.9x | 14.7x | 3.0x |
| | 0.5% | 2.158 | 1.059 | 0.219 ± 0.008 | 9.9x | 4.8x | 6.3x | 3.7x | 7.0x | 1.4x |
| | 1% | 1.975 | 1.276 | 0.381 ± 0.014 | 5.2x | 3.3x | 3.6x | 2.1x | 4.0x | 0.8x |
| CreditInfo | 1 | 52.493 | 0.113 | 0.056 ± 0.005 | 937.4x | 2.0x | 45.0x | 37.3x | 61.6x | 4,332.8x |
| | 0.1% | 49.458 | 0.426 | 0.138 ± 0.010 | 358.4x | 3.1x | 18.3x | 15.1x | 25.0x | 1,758.3x |
| | 0.5% | 51.892 | 0.824 | 0.230 ± 0.022 | 225.6x | 3.6x | 11.0x | 9.1x | 15.0x | 1,055.0x |
| | 1% | 58.349 | 1.065 | 0.338 ± 0.036 | 172.6x | 3.2x | 7.5x | 6.2x | 10.2x | 717.9x |
| SUSY | 1 | 180.390 | 1.707 | 1.283 ± 0.159 | 140.6x | 1.3x | 74.5x | 81.8x | 84.9x | 3.9x |
| | 0.1% | 169.545 | 23.999 | 6.146 ± 0.642 | 27.6x | 3.9x | 15.6x | 17.1x | 17.7x | 0.8x |
| | 0.5% | 184.025 | 53.962 | 15.472 ± 1.122 | 11.9x | 3.5x | 6.2x | 6.8x | 7.0x | 0.3x |
| | 1% | 176.730 | 77.76 | 26.666 ± 1.241 | 6.6x | 2.9x | 3.6x | 3.9x | 4.1x | 0.2x |
| HIGGS | 1 | OOM | 4.967 | 3.223 ± 0.034 | - | 1.5x | 69.7x | 75.8x | 94.2x | 3.9x |
| | 0.1% | OOM | 55.265 | 19.402 ± 0.272 | - | 2.8x | 11.6x | 12.6x | 15.6x | 0.7x |
| | 0.5% | OOM | 152.095 | 49.419 ± 1.451 | - | 3.1x | 4.5x | 4.9x | 6.1x | 0.3x |
| | 1% | OOM | 251.224 | 79.418 ± 1.407 | - | 3.2x | 2.8x | 3.1x | 3.8x | 0.2x |
| Optdigits | 1 | 0.286 | 0.015 | 0.010 ± 0.001 | 28.6x | 1.5x | 96.5x | 95.0x | 17.7x | 258.4x |
| | 0.1% | 0.182357 | 0.032 | 0.014 ± 0.002 | 13.0x | 2.3x | 68.9x | 67.9x | 12.6x | 184.6x |
| | 0.5% | 0.226884 | 0.067 | 0.030 ± 0.004 | 7.6x | 2.2x | 32.2x | 31.7x | 5.9x | 86.1x |
| | 1% | 0.187428 | 0.085 | 0.051 ± 0.005 | 3.7x | 1.7x | 18.9x | 18.6x | 3.5x | 50.7x |
| Pendigits | 1 | 0.171722 | **0.013** | 0.017 ± 0.001 | 10.1x | 0.8x | 51.8x | 68.5x | 10.8x | 167.8x |
| | 0.1% | 0.166171 | **0.022** | 0.025 ± 0.004 | 6.6x | 0.9x | 35.2x | 46.6x | 7.3x | 114.1x |
| | 0.5% | 0.181613 | 0.089 | 0.040 ± 0.007 | 4.5x | 2.2x | 22.0x | 29.1x | 4.6x | 71.3x |
| | 1% | 0.168511 | 0.129 | 0.055 ± 0.007 | 3.1x | 2.3x | 16.0x | 21.2x | 3.3x | 51.9x |
| Letter | 1 | 0.353374 | 0.017 | 0.012 ± 0.002 | 29.4x | 1.4x | 342.8x | 353.2x | 19.3x | 540.0x |
| | 0.1% | 0.35043 | **0.032** | 0.060 ± 0.009 | 5.8x | 0.5x | 68.6x | 70.6x | 3.9x | 108.0x |
| | 0.5% | 0.352653 | **0.066** | 0.104 ± 0.012 | 3.4x | 0.6x | 39.5x | 40.8x | 2.2x | 62.3x |
| | 1% | 0.43276 | **0.094** | 0.143 ± 0.013 | 3.0x | 0.7x | 28.8x | 29.6x | 1.6x | 45.3x |
| Covtype | 1 | 12.384 | 0.562 | 0.163 ± 0.013 | 76.0x | 3.4x | 204.7x | 113.9x | 37.7x | 18.8x |
| | 0.1% | 11.995 | 3.44 | 0.552 ± 0.040 | 21.7x | 6.2x | 60.5x | 33.6x | 11.1x | 5.6x |
| | 0.5% | 12.34 | 5.519 | 1.205 ± 0.047 | 10.2x | 4.6x | 27.7x | 15.4x | 5.1x | 2.5x |
| | 1% | 13.485 | 6.917 | 1.958 ± 0.174 | 6.9x | 3.5x | 17.0x | 9.5x | 3.1x | 1.6x |
| Abalone | 1 | 0.322 | 0.069 | 0.028 ± 0.004 | 11.5x | 2.5x | 2.6x | 1.9x | 19.0x | 38.9x |
| | 0.1% | 0.314 | 0.263 | 0.029 ± 0.005 | 10.8x | 9.1x | 2.5x | 1.8x | 18.4x | 37.6x |
| | 0.5% | 0.498 | 0.372 | 0.052 ± 0.005 | 9.6x | 7.2x | 1.4x | 1.0x | 10.3x | 21.0x |
| | 1% | 0.43 | 0.417 | 0.048 ± 0.007 | 9.0x | 8.7x | 1.5x | 1.1x | 11.1x | 22.7x |
| WineQuality | 1 | 0.354 | 0.022 | 0.015 ± 0.002 | 23.6x | 1.5x | 7.1x | 4.8x | 57.2x | 52.9x |
| | 0.1% | 0.299 | 0.196 | 0.025 ± 0.002 | 12.0x | 7.8x | 4.2x | 2.9x | 34.3x | 31.8x |
| | 0.5% | 0.492 | 0.298 | 0.040 ± 0.003 | 12.3x | 7.5x | 2.7x | 1.8x | 21.5x | 19.9x |
| | 1% | 0.435 | 0.333 | 0.051 ± 0.006 | 8.5x | 6.5x | 2.1x | 1.4x | 16.8x | 15.6x |

similar to MUinGBDT. Overall, DyGB offers achieves fast training while remaining competitive with popular GBDT libraries.

**Memory Overhead.** Memory usage is crucial for practical applications. Most incremental and decremental learning methods store auxiliary information or learned knowledge during training, occupying significant memory. As shown in Table 3, our DyGB's memory usage is significantly lower than OnlineGB, iGBDT, and DeltaBoost, while OnlineGB and DeltaBoost encountered OOM.

## 4.2 DYNAMIC LEARNING TIME

Retraining from scratch can be time-consuming, but in some cases, the cost of dynamic learning outweighs the benefits compared to retraining from scratch, making dynamic learning unnecessary. Thus, evaluating the cost of dynamic learning is crucial for practical applications. Table 1.1 and Table 1.2 shows the total dynamic learning time (s) and speedup vs. baselines, comparing OnlineGB & iGBDT for incremental learning, and DeltaBoost & MUinGBDT for decremental learning.

In incremental learning, compared to OnlineGB and iGBDT, which also support incremental learning, adding a single data instance can be up to 261.3x faster, respectively. Furthermore, compared to retraining from scratch on XGBoost, LightGBM, CatBoost, and ThunderGBM (GPU), it can achieve speedups of up to 241.9x, 249.3x, 64.7x, and 2,185.9x, respectively. In decremental learning, deleting a data is 937.4x and 9.1x faster than DeltaBoost and MUinGBDT, and 342.8x, 353.2x, 84.9x, and 4,332.8x faster than XGBoost, LightGBM, CatBoost, and ThunderGBM (GPU).

Interestingly, we observed that when $\|D'\|$ is small, decremental learning is faster than incremental learning. However, as $\|D'\|$ increases, incremental learning becomes faster than decremental learning. For decremental learning, the data to be removed has already been learned, and their derivatives have been stored from training. However, the deleted data often exists discretely in memory. On

Table 3: Comparison of total training time (in seconds) and memory usage (total allocated, MB).

| | Method | Adult | CreditInfo | SUSY | HIGGS | Optdigits | Pendigits | Letter | Covtype | Abalone | WineQuality |
|---|---|---|---|---|---|---|---|---|---|---|---|
| Training Time (Seconds) | iGBDT | 1.88 | 1.79 | 63.13 | 180.46 | 0.26 | 0.35 | 0.26 | 9.16 | 1.43 | 1.05 |
| | OnlineGB | 6,736.18 | 330,746.80 | OOM | OOM | 130.70 | 87.36 | 771.99 | 19,938.80 | 39.87 | 62.03 |
| | DeltaBoost | 114.64 | 247.44 | 5,494.03 | OOM | 12.25 | 25.98 | 43.41 | 724.21 | 4.54 | 6.38 |
| | MU in GBDT | 1.29 | 1.65 | 58.55 | 175.95 | 0.26 | 0.35 | 0.29 | 6.45 | 1.43 | 1.03 |
| | XGBoost | 1.38 | 2.52 | 95.63 | 224.59 | 0.97 | 0.88 | 4.11 | 33.37 | 0.07 | 0.11 |
| | LightGBM | 0.82 | 2.09 | 104.89 | 244.20 | 0.95 | 1.16 | 4.24 | 18.57 | 0.05 | 0.07 |
| | CatBoost | 1.53 | 3.45 | 108.95 | 303.56 | 0.18 | 0.18 | 0.23 | 6.14 | 0.53 | 0.86 |
| | ThunderGBM (GPU) | 0.31 | 242.64 | 5.05 | 12.66 | 2.58 | 2.85 | 6.48 | 3.07 | 1.09 | 0.79 |
| | Ours | 2.67 | 1.82 | 64.94 | 177.10 | 0.28 | 0.37 | 0.35 | 9.34 | 0.58 | 0.43 |
| Memory Usage (MB) | iGBDT | 1,153.13 | 2,192.13 | 31,320.40 | 31,724.40 | 2,161.20 | 3,917.61 | 3,370.38 | 18,381.10 | 1,767.23 | 1,281.08 |
| | OnlineGB | 35,804.10 | 58,119.61 | OOM | OOM | 7,493.97 | 6,488.75 | 13,067.75 | 19,699.62 | 582.97 | 345.83 |
| | DeltaBoost | 86,750.73 | 584,955.00 | 780,328.40 | OOM | 6,707.68 | 2,580.91 | 8,374.99 | 40,485.53 | 1,043.78 | 710.42 |
| | MU in GBDT | 570.78 | 1,095.70 | 16,576.50 | 34,380.90 | 1,080.49 | 1,959.02 | 1,805.22 | 9,637.65 | 1,711.02 | 1,194.82 |
| | XGBoost | 335.88 | 249.97 | 2,210.68 | 7,479.52 | 227.18 | 189.95 | 292.77 | 854.45 | 185.16 | 208.09 |
| | LightGBM | 233.61 | 278.54 | 2,847.48 | 9,830.89 | 248.49 | 234.41 | 252.26 | 836.46 | 27.95 | 11.81 |
| | CatBoost | 83.02 | 129.09 | 1,503.93 | 3,090.55 | 29.41 | 36.64 | 99.79 | 595.27 | 40.97 | 27.91 |
| | ThunderGBM (GPU) | 453.30 | 0.90 | 2,122.18 | 7,418.06 | 166.13 | 383.26 | 397.30 | 1,299.85 | 373.58 | 368.53 |
| | Ours | 577.18 | 1,096.71 | 16,576.40 | 24,333.30 | 1,081.15 | 1,959.49 | 1,805.76 | 9,665.21 | 762.78 | 531.88 |

the other hand, for incremental learning, the data to be added are unseen, and derivatives need to be computed during the incremental learning process. Nevertheless, we append the added data at the end, ensuring that the added data are stored contiguously in memory. With a small $|D'|$, derivatives can be reused in decremental learning, whereas derivatives need to be computed in incremental learning. Therefore, decremental learning is faster. However, as $|D'|$ grows, contiguous memory access in incremental learning becomes faster, making it more efficient.

## 4.3 BATCH ADDITION & REMOVAL

In the traditional setting, GBDT models must be trained in one step with access to all training data, and they cannot be modified after training – data cannot be added or removed. In our proposed dynamic learning framework, DyGB support both incremental and decremental learning, allowing continual batch learning (data addition) and batch removal, similar to mini-batch learning in DNNs.

We conducted experiments on continual batch addition and removal by dividing the data into 20 equal parts, each with $5\%|D_{tr}|$. Figure 3 (left) shows a GBDT model incrementally trained from $5\%$ to $100\%$ of the data, then decrementally reduced back to $5\%$. We retrained models for comparison. Figure 3 (right) depicts a model decrementally reduced from $100\%$ to $5\%$, then incrementally trained back to $100\%$. We also report the accuracy of XGBoost and LightGBM. The overlapping curves highlight DyGB's effectiveness. Due to space limit, results are shown for three datasets.

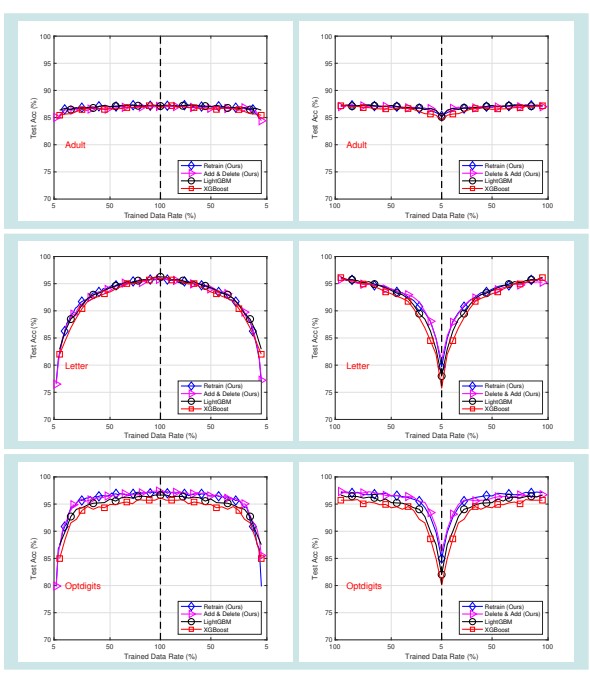

Figure 3: The impact of tuning data size on the number of retrained nodes for each iteration in incremental learning.

## 4.4 VERIFYING BY BACKDOOR ATTACKING

Backdoor attacks in machine learning refers to a type of malicious manipulation of a trained model, which is designed to modify the model's behavior or output when it encounters a specific, predefined trigger input pattern (Salem et al., 2022; Saha et al., 2020). In this evaluation, we demonstrate that DyGB can successfully inject and remove backdoor in a well-trained, clean GBDT model using incremental learning and decremental learning. The details of backdoor attack experiments are provided in Appendix L.

In this evaluation, we randomly selected a subset of the training dataset and injected triggers into it to create a backdoor training dataset, leaving the rest as the clean training dataset. The test dataset was similarly divided into backdoor and clean subsets. We report the accuracy for clean test dataset

Table 4: Accuracy for clean test dataset and attack successful rate for backdoor test dataset.

| Dataset | Train Clean | | Train Backdoor | | Add Backdoor | | Remove Backdoor | |
|---|---|---|---|---|---|---|---|---|
| | Clean | Backdoor | Clean | Backdoor | Clean | Backdoor | Clean | Backdoor |
| Optdigits | 96.21% | 8.91% | 96.27% | 100% | 95.94% | 100% | 95.82% | 9.69% |
| Pendigits | 96.11% | 3.97% | 96.43% | 100% | 96.48% | 100% | 96.51% | 5.55% |
| Letter | 93.9% | 1.38% | 94.08% | 100% | 93.62% | 100% | 93.78% | 3.48% |
| Covtype | 78.4% | 47.83% | 78.32% | 100% | 78.38% | 100% | 78.38% | 51.71% |

and attack successful rate (ASR) for backdoor test dataset in Table 4. Initially, we trained a model on the clean training data ("Train Clean"), which achieved high accuracy on the clean test dataset but low ASR on the backdoor test dataset. We then incrementally add the backdoor training data with triggers in to the model ("Add Backdoor"). After incremental learning, the model attained 100% ASR on the backdoor test dataset, demonstrating effective learning of the backdoor data. For comparison, training a model on the combined clean and backdoor training datasets ("Train Backdoor") yielded similar results to "Add Backdoor". Finally, we removed the backdoor data using decremental learning ("Remove Backdoor"), reducing the ASR to the level of the clean model and confirming the successful removal of backdoor data.

## 4.5 ADDITIONAL EVALUATIONS

To further validate our method's effectiveness and efficiency, we have included comprehensive additional evaluations in the Appendix due to page limitations:

• **Time Complexity Analysis:** We analyze the computational complexity of our proposed framework compared to retraining from scratch in Appendix H.

• **Test Error Rate:** We compare the test error rate between our proposed method and several baseline approaches, with detailed results provided in Appendix I.

• **Real-world Time Series Evaluation:** To confirm DyGB's performance on real-world datasets with varying data distributions, we report the experiments on two time series datasets in Appendix J.

• **Extremely High-dimensional Datasets:** To confirm the scalability of DyGB, we report the experiments for two extremely high-dimensional datasets in Appendix O.

• **Model Functional Similarity:** We evaluate the similarity between the model learned by dynamic learning and the one retrained from scratch in Appendix K.

• **Approximation Error of Leaf Scores:** Since DyGB might use the outdated derivatives in the gain computation, to assess the effect of outdated derivatives, we report the approximation error of leaf scores in Appendix Q.

• **Different Base Learners:** We include the experiments on various base learners in Appendix N.

• **Recommender System:** We report a practical use case of recommender system in Appendix R.

• **Data Addition with More Classes:** DyGB supports incremental learning for previously unseen classes. Detailed results and analysis are provided in Appendix P.

• **Membership Inference Attack:** We also confirm the effectiveness of our method on adding/deleting data by membership inference attack (MIA) in Appendix M.

• **Ablation Study:** We report the detailed ablation study results for different hyper-parameter settings and their effects in Appendix S.

• **Impact of Sub-optimal Splits:** To empirically validate the safety of our tolerance mechanism and the robustness of accuracy against sub-optimal splits, we report an experiment in Appendix T.

## 5 CONCLUSION

In this paper, we propose DyGB, an in-place dynamic learning framework for GBDT that support incremental and decremental learning: it enables us to dynamically add a new dataset to the model and delete a learned dataset from the model. It support continual batch addition/removal, and data additional with unseen classes. We present a collection of optimizations on DyGB to reduce the cost of dynamic learning. Adding or deleting a small fraction of data is substantially faster than retraining from scratch. Our extensive experimental results confirm the effectiveness and efficiency of DyGB and optimizations – successfully adding or deleting data while maintaining accuracy.

ETHICS STATEMENT

This research proposes a framework for dynamic gradient boosting decision trees that allows efficient addition and removal of data. The work does not involve human subjects, sensitive personal data, or proprietary datasets; all experiments are conducted using publicly available benchmark datasets. We carefully follow applicable data usage policies to ensure compliance with privacy and licensing requirements.

A potential ethical concern is the possibility that dynamic learning techniques could be misused for malicious purposes, such as unauthorized data manipulation or backdoor insertion. To address this, our study explicitly evaluates such scenarios to raise awareness of these risks and to demonstrate how our framework can also enable secure data removal when necessary. We affirm that this work is intended for advancing trustworthy and responsible machine learning research, and we disclose no conflicts of interest or external influences.

REPRODUCIBILITY STATEMENT

We have made extensive efforts to ensure that the findings of this paper are reproducible. All datasets used are publicly available, and their details are clearly specified. The methods are described at both the conceptual and algorithmic levels, including hyperparameters and evaluation protocols.

For transparency, we provide an anonymized implementation of the proposed framework, along with scripts and instructions to reproduce the reported experiments. Together with the detailed descriptions in the main paper and supplementary materials, this ensures that independent researchers can reliably replicate our results.

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

# APPENDIX

## Table of Contents

## A    THE USE OF LARGE LANGUAGE MODELS

Large Language Models (LLMs) were used in the preparation of this paper. Specifically, LLMs were employed to aid and polish the writing, helping to refine grammar, clarity, and readability of the text. No part of the research ideation, experimental design, implementation, or analysis relied on LLMs. The responsibility for all content presented in this paper rests fully with the authors.

## B    RELATED WORK

**Incremental Learning** is a technique in machine learning that involves the gradual integration of new data into an existing model, continuously learning from the latest data to ensure performance on new data (van de Ven et al., 2022). It has been a open problem in machine learning, and has been studied in convolutional neural network (CNN) (Polikar et al., 2001; Kuzborskij et al., 2013; Zhou et al., 2022), DNN (Hussain et al., 2023; Dekhovich et al., 2023), SVM (Chen et al., 2019; Cauwenberghs & Poggio, 2000) and RF (Wang et al., 2009; Brophy & Lowd, 2020). In gradient boosting, iGBDT offers incremental updates (Zhang et al., 2019), while other methods (Beygelzimer et al., 2015a; Babenko et al., 2009) extend GB to dynamic learning. However, these methods do not support removing data.

**Decremental Learning** allows for the removal of trained data and eliminates their influence on the model, which can be used to delete outdated or privacy-sensitive data (Bourtoule et al., 2021; Nguyen et al., 2022; Sekhari et al., 2021; Xu et al., 2024). It has been researched in various models, including CNN (Poppi et al., 2023; Tarun et al., 2021), DNN (Chen et al., 2023; Thudi et al., 2022), SVM (Karasuyama & Takeuchi, 2009; Cauwenberghs & Poggio, 2000), Naive Bayes (Cao & Yang, 2015), K-means (Ginart et al., 2019), RF (Schelter et al., 2021; Brophy & Lowd, 2021), and GB (Wu et al., 2023; Zhang et al., 2023). In random forests, DaRE (Brophy & Lowd, 2021) and a decremental learning algorithm (Schelter et al., 2021) are proposed for data removal with minimal retraining.

However, in GBDT, trees in subsequent iterations rely on residuals from previous iterations, making decremental learning more complicated. DeltaBoost Wu et al. (2023) simplified the dependency for data deletion by dividing the dataset into disjoint sub-datasets, while a recent study Lin et al. (2023) proposed an efficient unlearning framework without simplification, utilizing auxiliary information to reduce unlearning time. Although effective, its performance on large datasets remains unsatisfactory.

## C    FEATURE DISCRETIZATION.

The preprocessing step of feature discretization plays a crucial role in simplifying the implementation of Eq. equation 5 and reducing the number of splits that need to be evaluated. This process involves sorting the data points based on their feature values and assigning them to bins, taking into account the distribution of the data, as shown in Figure 4 and Algorithm 4.

Our implementation explicitly adopts the adaptive histogram construction method from the Fast ABC-Boost package (Li & Zhao, 2022), utilizing its specific adaptive bin-width doubling mechanism (Li et al., 2007). While this approach shares the foundational histogram-based philosophy found in LightGBM (Ke et al., 2017) and other baselines, it dynamically adjusts granularity to fit the data distribution.

As detailed in Algorithm 4, the process starts with a small initial bin-width (e.g., $10^{-10}$) and a predetermined maximum number of bins $B$ (e.g., 1024). It assigns bin numbers to the data points from the smallest to the largest, considering the presence of data points in each bin.

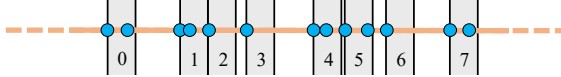

Figure 4: Feature discretization example. For a feature, all its values are grouped into 8 bins, i.e., the original feature values become integers between 0 to 7 assigned to the nearest bin.

---

**Algorithm 4** Discretize Feature

---

1: $v_{\{1..N\}}$ = sorted feature values, $bin\_width = 10^{-10}$
2: **while true do**
3:     $cnt = 0$, $curr\_idx = 0$
4:     **for** $i = 1$ **to** $N$ **do**
5:         **if** $v_i - v_{curr\_idx} > bin\_width$ **then**
6:             $cnt = cnt + 1$, $cur\_idx = i$
7:             **if** $cnt > B$ **then**
8:                 $bin\_width = bin\_width * 2$
9:                 **break**
10:             **end if**
11:         **end if**
12:         $v'_i = cnt$
13:     **end for**
14:     **if** $cnt <= B$ **then break**
15: **end while**
16: **return** $v'$ as discretized feature values

---

In cases where the number of required bins surpasses the maximum limit, the bin-width is doubled, and the entire process is repeated. This adaptive discretization approach proves particularly effective for boosted tree methods, ensuring that feature values are mapped to integers within a specific range. Consequently, after the discretization mapping is established, each feature value is assigned to the nearest bin. After this discretization preprocessing, all feature values are integers within $\{0, 1, 2, \cdots, B - 1\}$.

The advantage of this discretization technique becomes evident during the gain searching step. Instead of iterating over all $N$ feature values, the algorithm only needs to consider a maximum of $B$ splits for each feature. This substantial reduction in the number of splits to evaluate leads to a significant decrease in the computational cost, transforming it from being dependent on the dataset size $N$ to a manageable constant $B$.

Table 5: Hyper-parameters for experiments. ($*$ indicates the parameter is default or recommended from original sources).

| Methods | Learning Rate | Iterations | Max Leaf Num | Depth | Num Bins | Others |
|---|---|---|---|---|---|---|
| OnlineGB | - | 100 | - | - | - | The tree grown automatically. |
| iGBDT | 0.1 | 100 | - | 5 | - | |
| DeltaBoost | 1* | 100 | - | 5 | 100* | All other parameters remain default |
| MU in GBDT | 0.1* | 100 | 20 | - | 1024* | update_freq = 20, sample_rate = 0.1, $L_r = 0$ |
| XGBoost | 0.1 | 100 | 20 | - | 128 | |
| LightGBM | 0.1 | 100 | 20 | - | 128 | |
| CatBoost | 0.1 | 100 | 20 | - | - | |
| ThunderGMB (GPU) | 0.1 | 100 | - | 5 | 128 | A100 40GB GPU * 1 |
| Ours | 1 | 100 | 20 | - | 1024 | Sampling Rate = 0.1, Robustness Tolerance = 0.1 |

## D    EXPERIMENT SETTING

The experiments are performed on a Linux computing node running Red Hat Enterprise Linux 7, utilizing kernel version 5.10.155-1.el7.x86_64. The CPU employed was an Intel(R) Xeon(R) Gold 6150 CPU operating at a clock speed of 2.70GHz, featuring 18 cores and 36 threads. The system was equipped with a total memory capacity of 376 GB. We have built a prototype of our dynamic learning framework using C++11. The code is compiled with g++-11.2.0, utilizing the "O3" optimization.

**Hyper-parameter Configuration.** To ensure a fair and reproducible comparison, we detail the hyper-parameter settings for all methods in Table 5. Unless explicitly stated otherwise, our default parameter settings are: $J = 20$, $B = 1024$, $|D'| = 0.1\% \times |D_{tr}|$, $\alpha = 0.1$, and $\sigma = 0.1$. For baseline methods, including OnlineGB, iGBDT, DeltaBoost, MU in GBDT, XGBoost, LightGBM, CatBoost, and ThunderGBM, we adopted either their default parameters or the specific values recommended in their respective original papers (denoted by $*$ in the table). Specifically, for standard GBDT libraries (XGBoost, LightGBM, ThunderGBM), we set the max bin size to 128 to optimize their training efficiency as per common practice, while maintaining a higher resolution ($B = 1024$)

for our method to demonstrate its efficiency even under more demanding discretization. For our DyGB's incremental and decremental learning scenarios, we additionally run five independent trials and report the mean and variance to ensure statistical robustness. We report the ablation study for different settings in Appendix S.

## E   FRAMEWORK OVERVIEW

Figure 1 is a visual example of incremental and decremental learning of our proposed framework. Figure 1(b) is one tree of the GBDT model and has been well-trained on dataset $D_{tr} = \{0, 1, 2, 3..., 19\}$. Every rectangle in the tree represents a node, and the labels inside indicate the splitting criteria. For instance, if the condition `Age < 42` is met, the left-child node is followed; otherwise, the right-child node is chosen. The numbers within the rectangles represent the prediction value of the terminal nodes. Please note that here the feature `42` is a discretized value, instead of the raw feature. Our dynamic learning framework has the capability to not only incrementally learn a new dataset $D_{in}$, but also decrementally delete a learned dataset $D_{de} \subseteq D_{tr}$.

**Example for Incremental Learning.** Here, we would like to add a new dataset $D' = D_{in} = \{20, 21, 22, 23\}$ to the original model, so we will call the function of incremental learning. $|d|$ denotes how many data of the $D'$ reach this node. As shown in Algorithm 3, we traverse all non-terminal nodes (non-leaf nodes) in the tree at first. For example, we are going to test the node of `Loan < 31`. Its current best split is `Loan < 31`. One of the new data instances $\{22\}$ reaches this node. After adding this data and recomputing the gain value, `Loan < 31` is still best split with the greatest gain value of 26.937, and meets $s = s'$, as shown in Figure 1(a). Thus, we can keep this split and do not need to do any changes for this node. Then we are going to test the node of `Auto < 57` and the remaining three new data instances $\{20, 21, 23\}$ reach this node. As shown in the left side of Figure 1(c), we recompute the gain value for this node, but the best split changes to `Income < 5`. Therefore, we retrain the pending sub-tree rooted on `Auto < 57` after adding new data instances to obtain a new sub-tree rooted on `Income < 5`. Then we replace the pending sub-tree with the new one. Finally, we update the prediction value on terminal nodes (leaf nodes). For example, 0.4322 is updated to 0.2735 because of adding data $\{22\}$; $-0.1252$ has no change because the data of this node are still the same.

**Example for Decremental Learning.** Similar to incremental learning, we would like to delete a learned dataset $D_{de} = \{2, 7, 11, 13\}$ and its effect on the model. The best split of node `Loan < 31` does not change, so we keep the split. For `Auto < 57`, as shown in the right side of Figure 1(c), after removing data instances $\{2, 11, 13\}$, the best split changes from `Auto < 57` to `Credit < 24`, so we retrain the pending sub-tree rooted on `Loan < 31` and then replace it with the new sub-tree. For terminal nodes, the prediction value changes if any data reaching this node is removed.

## F   SPLIT CANDIDATES SAMPLING: THEORETICAL ANALYSIS

All symbols used in the theoretical analysis are defined in Table 6.

### F.1   ROBUSTNESS VIA DISTANCE AND GAIN MARGIN

**Definition 1** (Distance Robust). *Let $s$ be the current best split and let $\lambda = |D'|/|D_{\mathrm{tr}}|$ denote the dynamic update ratio. Let $N_\Delta$ be the distance between $s$ and its nearest alternative split $t$. We say that $s$ is* distance robust *if*

$$N_\Delta > \lambda \, Gain(s) \, C_s, \tag{9}$$

*where the node structural coefficient $C_s$ is defined as*

$$C_s := \left( \frac{1}{N_{ls}} \frac{\left(\sum_{x_i \in l_s} g_{i,k}\right)^2}{\sum_{x_i \in l_s} h_{i,k}} + \frac{1}{N_{rs}} \frac{\left(\sum_{x_i \in r_s} g_{i,k}\right)^2}{\sum_{x_i \in r_s} h_{i,k}} \right)^{-1}. \tag{10}$$

Table 6: Notation table used throughout the theoretical analysis.

| Symbol | Meaning |
|--------|---------|
| $D_{\text{tr}}$ | Training dataset before dynamic update |
| $D'$ | Added/removed subset during dynamic update |
| $\lambda = \|D'\|/\|D_{\text{tr}}\|$ | Dynamic update ratio |
| $s$ | Current best split candidate |
| $t$ | A competing split under the same feature |
| $N_\Delta = \|t - s\|$ | Distance between split $s$ and nearest competitor $t$ |
| $l_s, r_s$ | Left / right child regions of split $s$ |
| $N_{ls}, N_{rs}$ | Number of samples in $l_s$ and $r_s$ |
| $g_{i,k}, h_{i,k}$ | First-order and second-order gradients of sample $i$ at iteration $k$ |
| $G_L = \sum_{x_i \in l_s} g_{i,k}$ | Gradient sum of left child |
| $G_R = \sum_{x_i \in r_s} g_{i,k}$ | Gradient sum of right child |
| $H_L = \sum_{x_i \in l_s} h_{i,k}$ | Hessian sum of left child |
| $H_R = \sum_{x_i \in r_s} h_{i,k}$ | Hessian sum of right child |
| $Gain(s)$ | Split gain of split $s$ |
| $Gain'(s)$ | Split gain after dynamic update |
| $n_{ls}, n_{rs}$ | Number of removed samples in $l_s$ and $r_s$ |
| $\overline{g}_{ls}, \overline{h}_{ls}$ | Average gradient / Hessian in $l_s$ |
| $\alpha$ | Split sampling rate |
| $\sigma$ | Robustness tolerance threshold |

**Proof.** In decremental learning, for a fixed $\lambda$, we have

$$(1 - \lambda)\, Gain(s) - Gain(s + N_\Delta) \tag{11}$$

$$\approx (1 - \lambda) \left( \frac{\left(\sum_{\mathbf{x}_i \in l_s} g_{i,k}\right)^2}{\sum_{\mathbf{x}_i \in l_s} h_{i,k}} + \frac{\left(\sum_{\mathbf{x}_i \in r_s} g_{i,k}\right)^2}{\sum_{\mathbf{x}_i \in r_s} h_{i,k}} - \frac{\left(\sum_{\mathbf{x}_i \in l_s \cup r_s} g_{i,k}\right)^2}{\sum_{\mathbf{x}_i \in l_s \cup r_s} h_{i,k}} \right)$$

$$- \left( \left(1 - \frac{N_\Delta}{N_{ls}}\right) \frac{\left(\sum_{\mathbf{x}_i \in l_s} g_{i,k}\right)^2}{\sum_{\mathbf{x}_i \in l_s} h_{i,k}} + \left(1 - \frac{N_\Delta}{N_{rs}}\right) \frac{\left(\sum_{\mathbf{x}_i \in r_s} g_{i,k}\right)^2}{\sum_{\mathbf{x}_i \in r_s} h_{i,k}} \right.$$

$$\left. - \frac{\left(\sum_{\mathbf{x}_i \in l_s \cup r_s} g_{i,k}\right)^2}{\sum_{\mathbf{x}_i \in l_s \cup r_s} h_{i,k}} \right) \tag{12}$$

where $l$ represents the left child of split $s$, and it contains the samples belonging to this node, while $r$ represent the right child, $N_{ls}$ denotes $|l_s|$, and $N_{rs}$ denotes $|r_s|$.

Let $(1 - \lambda)Gain(s) - Gain(s + N_\Delta) > 0$, we have

$$\overset{approx}{\Rightarrow} (1 - \lambda)Gain(s) - \left( \left(1 + \frac{N_\Delta}{N_{ls}}\right) \frac{\left(\sum_{\mathbf{x}_i \in l_s} g_{i,k}\right)^2}{\sum_{\mathbf{x}_i \in r_s} h_{i,k}} \right.$$

$$\left. + \left(1 - \frac{N_\Delta}{N_{rs}}\right) \frac{\left(\sum_{\mathbf{x}_i \in r_s} g_{i,k}\right)^2}{\sum_{\mathbf{x}_i \in r_s} h_{i,k}} - \frac{\left(\sum_{\mathbf{x}_i \in l_s \cup r_s} g_{i,k}\right)^2}{\sum_{\mathbf{x}_i \in l_s \cup r_s} h_{i,k}} \right) \tag{13}$$

$$\Rightarrow \frac{N_\Delta}{N_{ls}} \frac{\left(\sum_{\mathbf{x}_i \in l_s} g_{i,k}\right)^2}{\sum_{\mathbf{x}_i \in l_s} h_{i,k}} + \frac{N_\Delta}{N_{rs}} \frac{\left(\sum_{\mathbf{x}_i \in r_s} g_{i,k}\right)^2}{\sum_{\mathbf{x}_i \in r_s} h_{i,k}} - \lambda Gain(s) > 0 \tag{14}$$

$$\Rightarrow N_\Delta > \frac{\lambda Gain(s)}{\frac{1}{N_{ls}} \frac{\left(\sum_{\mathbf{x}_i \in l_s} g_{i,k}\right)^2}{\sum_{\mathbf{x}_i \in l_s} h_{i,k}} + \frac{1}{N_{rs}} \frac{\left(\sum_{\mathbf{x}_i \in r_s} g_{i,k}\right)^2}{\sum_{\mathbf{x}_i \in r_s} h_{i,k}}} \tag{15}$$

where $l$ represents the left child of split $s$, and it contains the samples belonging to this node, while $r$ represents the right child, $N_{ls}$ denotes $|l_s|$, and $N_{rs}$ denotes $|r_s|$.

Solving for $N_\Delta$ yields

$$N_\Delta > \lambda\, Gain(s) \left( \frac{1}{N_{ls}} \frac{\left(\sum_{\mathbf{x}_i \in l_s} g_{i,k}\right)^2}{\sum_{\mathbf{x}_i \in l_s} h_{i,k}} + \frac{1}{N_{rs}} \frac{\left(\sum_{\mathbf{x}_i \in r_s} g_{i,k}\right)^2}{\sum_{\mathbf{x}_i \in r_s} h_{i,k}} \right)^{-1}. \tag{16}$$

To simplify notation, we define the *node structural coefficient*

$$C_s := \left( \frac{1}{N_{ls}} \frac{\left(\sum_{\mathbf{x}_i \in l_s} g_{i,k}\right)^2}{\sum_{\mathbf{x}_i \in l_s} h_{i,k}} + \frac{1}{N_{rs}} \frac{\left(\sum_{\mathbf{x}_i \in r_s} g_{i,k}\right)^2}{\sum_{\mathbf{x}_i \in r_s} h_{i,k}} \right)^{-1}, \tag{17}$$

so that the distance-robustness condition takes the clean and compact form

$$N_\Delta > \lambda\, Gain(s)\, C_s. \tag{18}$$

$\square$

The above expression characterizes when the best split $s$ cannot be overtaken by its nearest competitor $t$ after removing a random subset $D'$. To further simplify the analysis, we next derive a general bound on the gain perturbation.

**Lemma 1** (Gain Perturbation Bound). *For any fixed split $s$, removing a random subset $D'$ with rate $\lambda = |D'|/|D_{\mathrm{tr}}|$ changes its gain by at most $O(\lambda)$:*

$$\left| Gain'(s) - Gain(s) \right| \leq \lambda \left( \frac{\left(\sum_{\mathbf{x}_i \in l_s} g_{i,k}\right)^2}{\sum_{\mathbf{x}_i \in l_s} h_{i,k}} + \frac{\left(\sum_{\mathbf{x}_i \in r_s} g_{i,k}\right)^2}{\sum_{\mathbf{x}_i \in r_s} h_{i,k}} \right) = O(\lambda). \tag{19}$$

**Proof.** For the left child of split $s$, after deleting $n_{ls}$ samples,

$$G'_L = \sum_{x_i \in l_s} g_{i,k} - \sum_{x_i \in l_s \cap D'} g_{i,k} \approx G_L - n_{ls}\, \overline{g}_{ls}, \tag{20}$$

$$H'_L = \sum_{x_i \in l_s} h_{i,k} - \sum_{x_i \in l_s \cap D'} h_{i,k} \approx H_L - n_{ls}\, \overline{h}_{ls}. \tag{21}$$

Then

$$\frac{(G'_L)^2}{H'_L} \approx \frac{(G_L - n_{ls}\overline{g}_{ls})^2}{H_L - n_{ls}\overline{h}_{ls}} = \frac{G_L^2}{H_L}\left(1 - \frac{n_{ls}}{N_{ls}}\right) + O\left(\frac{n_{ls}}{N_{ls}}\right). \tag{22}$$

Because random deletion implies

$$\frac{n_{ls}}{N_{ls}} \approx \lambda, \quad \frac{n_{rs}}{N_{rs}} \approx \lambda, \tag{23}$$

we obtain

$$Gain'(s) = \frac{(G'_L)^2}{H'_L} + \frac{(G'_R)^2}{H'_R} - \frac{(G'_L + G'_R)^2}{H'_L + H'_R} \tag{24}$$

$$\approx (1-\lambda)\left( \frac{G_L^2}{H_L} + \frac{G_R^2}{H_R} \right) - (1-\lambda)\frac{(G_L + G_R)^2}{H_L + H_R} + O(\lambda). \tag{25}$$

$$= (1-\lambda)Gain(s) + O(\lambda). \tag{26}$$

Thus,

$$\left| Gain'(s) - Gain(s) \right| \leq \lambda \left( \frac{G_L^2}{H_L} + \frac{G_R^2}{H_R} \right) = O(\lambda). \tag{27}$$

$\square$

We next analyze a complementary notion of robustness based on the gain margin between competing splits, which captures how large the gain advantage of the current best split must be in order to remain stable under dynamic updates.

**Definition 2** (Robustness Split). *For the best split $s$ and any other split $t \neq s$ under the same feature, $s$ is a* robust split *under deletion rate $\lambda = |D'|/|D_{\text{tr}}|$ if*

$$Gain(s) > \frac{1}{1-\lambda}Gain(t). \tag{28}$$

**Proof.** Initially, we have

$$Gain(s) = \frac{\left(\sum_{\mathbf{x}_i \in l_s} g_{i,k}\right)^2}{\sum_{\mathbf{x}_i \in l_s} h_{i,k}} + \frac{\left(\sum_{\mathbf{x}_i \in r_s} g_{i,k}\right)^2}{\sum_{\mathbf{x}_i \in r_s} h_{i,k}} - \frac{\left(\sum_{\mathbf{x}_i \in l_s \cup r_s} g_{i,k}\right)^2}{\sum_{\mathbf{x}_i \in l_s \cup r_s} h_{i,k}} \tag{29}$$

After decremental learning, we get

$$Gain'(s) = \frac{\left(\sum_{\mathbf{x}_i \in l_s} g_{i,k} - \sum_{\mathbf{x}_i \in l_s \cap D'} g_{i,k}\right)^2}{\sum_{\mathbf{x}_i \in l_s} h_{i,k} - \sum_{\mathbf{x}_i \in l_s \cap D'} h_{i,k}} + \frac{\left(\sum_{\mathbf{x}_i \in r_s} g_{i,k} - \sum_{\mathbf{x}_i \in r_s \cap D'} g_{i,k}\right)^2}{\sum_{\mathbf{x}_i \in r_s} h_{i,k} \sum_{\mathbf{x}_i \in r_s \cap D'} h_{i,k}}$$
$$- \frac{\left(\sum_{\mathbf{x}_i \in l_s \cup r_s} g_{i,k} - \sum_{\mathbf{x}_i \in (l_s \cup r_s) \cap D'} g_{i,k}\right)^2}{\sum_{\mathbf{x}_i \in l_s \cup r_s} h_{i,k} - \sum_{\mathbf{x}_i \in (l_s \cup r_s) \cap D'} h_{i,k}} \tag{30}$$

For any possible split $t$ ($t \neq s$), the split $s$ is robust only and only if $Gain(s) > Gain(t)$ and $Gain'(s) > Gain'(t)$. First, let's analyze the first term of $Gain'(s)$. Suppose $\frac{|D'|}{|D_{tr}|} = \lambda$, and $D'$ is randomly selected from $D$. Here we consider the leaf child $l_s$ of split $s$, and let the $|l_s \cap D'|$ to be $n_{ls}$, $|l_s|$ to be $N_{ls}$. Then we have

$$\frac{\left(\sum_{\mathbf{x}_i \in l_s} g_{i,k} - \sum_{\mathbf{x}_i \in l_s \cap D'} g_{i,k}\right)^2}{\sum_{\mathbf{x}_i \in l_s} h_{i,k} - \sum_{\mathbf{x}_i \in l_s \cap D'} h_{i,k}} \xrightarrow{approx} \frac{\left(\sum_{\mathbf{x}_i \in l_s} g_{i,k} - n_{ls}\overline{g}_{ls}\right)^2}{\sum_{\mathbf{x}_i \in l_s} h_{i,k} - n_{ls}\overline{h}_{ls}} \tag{31}$$

$$\Rightarrow \left(1 - \frac{n_{ls}}{N_{ls}}\right)\frac{\left(\sum_{\mathbf{x}_i \in l_s} g_{i,k}\right)^2}{\sum_{\mathbf{x}_i \in l_s} h_{i,k}} \tag{32}$$

where $\overline{g}$ and $\overline{h}$ denote the average of the $g_{i,k}$ and $h_{i,k}$ respectively.

Similarly, we can get all three terms for $Gain(s)$, $Gain'(s)$, $Gain(t)$, and $Gain'(t)$ in a similar form. For $Gain'(s) > Gain'(t)$, finally, we have $Gain(s) > Gain(t) + C$, where

$$C = \left(\frac{n_{ls}}{N_{ls}}\frac{\left(\sum_{\mathbf{x}_i \in l_s} g_{i,k}\right)^2}{\sum_{\mathbf{x}_i \in r_s} h_{i,k}} + \frac{n_{rs}}{N_{rs}}\frac{\left(\sum_{\mathbf{x}_i \in r_s} g_{i,k}\right)^2}{\sum_{\mathbf{x}_i \in r_s} h_{i,k}}\right.$$
$$\left. - \frac{n_{ls} + n_{rs}}{N_{ls} + N_{rs}}\frac{\left(\sum_{\mathbf{x}_i \in l_s \cup r_s} g_{i,k}\right)^2}{\sum_{\mathbf{x}_i \in l_s \cup r_s} h_{i,k}}\right) - \left(\frac{n_{lt}}{N_{lt}}\frac{\left(\sum_{\mathbf{x}_i \in l_t} g_{i,k}\right)^2}{\sum_{\mathbf{x}_i \in r_t} h_{i,k}}\right.$$
$$\left. + \frac{n_{rt}}{N_{rt}}\frac{\left(\sum_{\mathbf{x}_i \in r_t} g_{i,k}\right)^2}{\sum_{\mathbf{x}_i \in r_t} h_{i,k}} - \frac{n_{lt} + n_{rt}}{N_{lt} + N_{rt}}\frac{\left(\sum_{\mathbf{x}_i \in l_t \cup r_t} g_{i,k}\right)^2}{\sum_{\mathbf{x}_i \in l_t \cup r_t} h_{i,k}}\right) \tag{33}$$

The upper bound of $C$ is $\lambda Gain(s)$. Further, we have

$$Gain(s) > \frac{1}{1-\lambda}Gain(t) \tag{34}$$

$\square$

In order to connect Definition 2 with a margin condition, we first upper-bound the correction term $C$ arising from the removal of $D'$.

**Lemma 2** (Upper Bound on the Correction Term $C$). *Let $C$ be the correction term defined in the proof of Definition 2. Under random deletion with rate $\lambda$,*

$$C \leq \lambda\, Gain(s). \tag{35}$$

**Proof.** From the approximation in the proof of Definition 2, each gain term takes the form

$$\frac{\left(\sum_{\mathbf{x}_i \in l_s} g_{i,k} - \sum_{\mathbf{x}_i \in l_s \cap D'} g_{i,k}\right)^2}{\sum_{\mathbf{x}_i \in l_s} h_{i,k} - \sum_{\mathbf{x}_i \in l_s \cap D'} h_{i,k}} \approx \left(1 - \frac{n_{ls}}{N_{ls}}\right) \frac{G_L^2}{H_L}, \tag{36}$$

and similarly for the right child and the merged node. Thus the perturbation satisfies

$$C = \frac{n_{ls}}{N_{ls}} \frac{G_L^2}{H_L} + \frac{n_{rs}}{N_{rs}} \frac{G_R^2}{H_R} - \frac{n_{ls} + n_{rs}}{N_{ls} + N_{rs}} \frac{(G_L + G_R)^2}{H_L + H_R} \tag{37}$$

$$\leq \lambda \left(\frac{G_L^2}{H_L} + \frac{G_R^2}{H_R} - \frac{(G_L + G_R)^2}{H_L + H_R}\right) \tag{38}$$

$$= \lambda \, Gain(s). \tag{39}$$

$$\square$$

We are now ready to provide a clean robustness condition.

**Proposition 1** (Margin Condition for Robustness)**.** *Let $s$ be the best split and $t$ any competitor. If*

$$Gain(s) - Gain(t) > \lambda \, Gain(s), \tag{40}$$

*then after deleting $D'$ with rate $\lambda$, we still have*

$$Gain'(s) > Gain'(t). \tag{41}$$

*Moreover,*

$$Gain(s) - Gain(t) > \lambda Gain(s) \iff Gain(s) > \frac{1}{1-\lambda} Gain(t), \tag{42}$$

*i.e., this margin condition is equivalent to Definition 2.*

**Proof.** By Lemma 2,

$$Gain'(s) \geq Gain(s) - \lambda Gain(s) = (1 - \lambda) Gain(s), \tag{43}$$
$$Gain'(t) \leq Gain(t). \tag{44}$$

Thus

$$Gain'(s) > Gain'(t) \Leftarrow (1 - \lambda) Gain(s) > Gain(t), \tag{45}$$

which is equivalent to

$$Gain(s) > \frac{1}{1-\lambda} Gain(t). \tag{46}$$

$$\square$$

**Discussion.** Definitions 1 and 2, together with Lemmas 1 and 2 and Proposition 1, establish that: (i) the gain of any fixed split changes only by $O(\lambda)$ under dynamic updates, and (ii) if the original gain margin exceeds this perturbation, the best split remains stable. This explains both the empirical robustness of DyGB under small data modifications and the reduced retraining frequency when either $\lambda$ or the sampling rate $\alpha$ decreases.

**Summary.** Our robustness analysis consists of five complementary components. (1) *Distance Robustness* characterizes the stability of a split based on how far its nearest competing split lies in the sorted feature space: a split remains stable when the neighbor distance exceeds the perturbation-scaled threshold $N_\Delta > \lambda \, Gain(s) \, C_s$. (2) *Gain Perturbation Bound* shows that dynamic updates modify the gain of any fixed split by at most $O(\lambda)$, providing a quantitative limit on how much the split metric can drift. (3) *Robustness Split* formalizes the requirement that the best split must retain a higher gain than any competitor even after the update. (4) Using our perturbation analysis, we upper bound the total correction term $C$ and show that its worst-case effect is no more than $\lambda \, Gain(s)$. (5) These results together yield the *Margin Condition for Robustness*, which states that a split remains optimal if and only if its original gain margin exceeds the maximum perturbation: $Gain(s) - Gain(t) > \lambda \, Gain(s)$, or equivalently, $Gain(s) > Gain(t)/(1-\lambda)$.

## G  UPDATE W/O TOUCHING TRAINING DATA

**Maintain Best Split.** The split gain is calculated by Eq. equation 5. There are three terms: the gain for the left-child, the gain for the right-child, and subtracting the gain before the split. Each gain is computed as the sum of the squared first derivatives $\left(\left[\sum_{i=1}^{N}(r_{i,k}-p_{i,k})\right]^2\right)$ divided by the sum of the second derivatives $\left(\sum_{i=1}^{N}p_{i,k}(1-p_{i,k})\right)$ for all the data in the node. To compute these terms, it is necessary to iterate over all the data that reaches the current node. The most straightforward way for dynamic learning to obtain the split gain is to directly compute these three terms for dataset $D_{tr} \pm D'$. In the worst case, which is the root node, the computation cost for gain computing is $|D_{tr}| + |D_{in}|$ or $|D_{tr}| - |D_{de}|$ because the root node contains all the training data.

We calculate the split gain for $D_{tr} \pm D'$ without touching the $D_{tr}$. In this optimization, during the training process, we store the $S_{rp} = \sum_{i=1}^{N}(r_{i,k}-p_{i,k})$ and $S_{pp} = \sum_{i=1}^{N}p_{i,k}(1-p_{i,k})$ for the training dataset $D_{tr}$ for every potential split. In incremental learning process, we can only calculate the $S'_{rp}$ and $S'_{pp}$ for $D_{in}$. To obtain the new split gain based on Eq. equation 5, we add it to the stored $S_{rp}$ and $S_{pp}$. Similarly, for decremental learning, we can only calculate the $S'_{rp}$ and $S'_{pp}$ for $D_{de}$ to obtain the new split gain. In this manner, we successfully avoid the original training data for split gain computation and reduce the computation cost from $O(D_{tr} \pm D')$ to $O(D')$.

**Recomputing Prediction Value.** For the terminal node (leaf node), if there are no data of $D'$ reaching this node, we can skip this node and do not need to change the prediction value. Otherwise, we have to calculate a prediction value $f$ as shown in line 5 of the Algorithm 1. Similar to split gain computing, it is required to iterate over all the data that reaches this terminal node. Here we store $S_{rp} = \sum_{\mathbf{x}_i \in R_{j,k,m}}(r_{i,k}-p_{i,k})$ and $S_{pp} = \sum_{\mathbf{x}_i \in R_{j,k,m}}(1-p_{i,k})p_{i,k}$ for training dataset $D_{tr}$ in training process. Thus, in dynamic learning process, we only need to calculate $S'_{rp}$ and $S'_{pp}$ for dynamic learning dataset $D'$.

**Incremental Update for Derivatives.** After conducting dynamic learning on a tree, we need to update the derivatives and residuals for learning the next tree. From the perspective of GBDT training, each tree in the ensemble is built using the residuals learned from the trees constructed in all previous iterations: Modifying one of the trees affects all the subsequent trees. A trivial method is to update the derivatives and residuals for all data instances of $D_{tr} \pm D'$ in every tree, but it is time-consuming.

When performing dynamic learning on a tree, not all terminal nodes will be changed—some terminal nodes remain unchanged because there is no data from $D'$ that reaches these terminal nodes. Note that our goal is to find a model close to the model retraining from scratch. In the dynamic learning scenario, all trees have already been well-trained on $D_{tr}$. Intuitively, the derivative changes for data in those unchanged terminal nodes should be minimal. Therefore, as shown in Figure 1(d), we only update the derivatives for those data reaching the changed terminal nodes. For example, the terminal node with a prediction value of $-0.1252$ does not meet any data in $D'$ in both incremental learning and decremental learning, so the prediction value of this node does not need to be changed. Therefore, we do not need to update the derivatives of the data $\{1, 6, 14, 16, 17\}$ reaching this terminal node.

## H  TIME COMPLEXITY

We compare the time complexity of retraining from scratch and our dynamic learning approach in Table 7. Training a tree involves three key steps: Derivatives Computing, Gain Computing & Split Finding, and Prediction Computing. Let $B$ represent the number of bins, $J$ the number of leaves, $|D_{tr}|$ the number of training data points, and $|D'|$ the number of dynamic learning data points ($|D'| \ll |D_{tr}|$).

**Derivatives Computing.** In retraining, each point is assigned to one of the $B$ bins, which take $O(\|D_{tr}\|)$ time. In our method, we optimize updates without touching training data, directly adding or subtracting derivatives for the dynamic data points, which takes $O(\|D'\|)$ time.

**Gain Computing & Split Finding.** In training, to identify the optimal split for each node, we compute the potential split gains for each bin. As a binary tree is constructed with $2J - 1$ nodes, the total computational complexity for split finding across the entire tree is $O(B(2J - 1)) = O(BJ)$. In our approach, Split Candidates Sampling reduces the number of split candidates from $B$ to $\alpha B$, where $\alpha$ denotes the split sample rate ($0 < \alpha \leq 1$). Additionally, let $P_\sigma$ represent the probability of a split change being within the robustness tolerance, indicating the likelihood that a node does not require retraining (with larger $\sigma$, $P_\sigma$ increases). If retraining is not required, the time complexity for checking a node is $O(|D'|)$. Conversely, if retraining is required, the complexity to retrain a node is $O(\alpha B)$. Consequently, the total time complexity for the entire tree is $O(J|D'| \cdot P_\sigma + J\alpha B \cdot (1 - P_\sigma))$. For $P_\sigma \to 1$, no nodes require retraining, simplifying the complexity to $O(J|D'|)$. Conversely, for $P_\sigma \to 0$, all nodes require retraining, and the complexity becomes $O(J\alpha B)$.

**Predicted Value Computing.** During training, after the tree is built, the predicted value for each leaf is updated. This involves traversing the leaf for the data points that reach it, with the total number being equivalent to all training data points, resulting in a complexity of $O(|D_{tr}|)$. In our method, we update the predicted value only for leaves reached by at least one dynamic data point, and adjust by adding/subtracting the impact of dynamic data points, resulting in a complexity of $O(|D'|)$.

Table 7: Time complexity comparison between retraining and dynamic learning.

| Step | Training Time | Optimization | Dynamic Learning Time |
|---|---|---|---|
| Derivatives Computing | $O(|D_{tr}|)$ | Update without Touching Training Data | $O(|D'|)$ |
| Gain Computing & Split Finding | $O(BJ)$ | Split Candidates Sampling, Split Robustness Tolerance | $O(\alpha BJ\sigma)$ |
| Predition Computing | $O(|D_{tr}| \log J)$ | Update without Touching Training Data | $O(|D'|)$ |

## I  TEST ERROR RATE

Table 8 presents the test error for different methods, defined as (1 - accuracy) for classification tasks and Mean Squared Error (MSE) for regression tasks. We have omitted the results for OnlineGB, as its excessively long learning time makes it relatively insignificant compared to the other methods. Three scenarios are considered: (1) Training, reporting the test error for models trained on the full dataset $D$; (2) Incremental Learning, performing incremental learning to add a randomly selected portion $D'$ into a model pre-trained on $D - D'$; and (3) Decremental Learning, conducting decremental learning to remove $D'$ from a model trained on the full dataset $D$. As shown in Table 8, The proposed DyGB achieved the best error rates in most cases.

## J  REAL-WORLD TIME SERIES EVALUATION

To confirm the performance of our methods on real-world datasets with varying data distributions, we conducted experiments on two real-world time series datasets from Kaggle:

• **GlobalTemperatures** (Glo, 2017): This dataset records the average land temperatures from 1750 to 2015.

• **WebTraffic** (Web, 2024): This dataset tracks hourly web requests to a single website over five months.

For this experiment, we constructed the input data $X$ using the time series values from the previous 15 time steps, with the goal of predicting the corresponding output value $y$. Initially, we randomly sample 10% of the data as the test dataset, with the remaining 90% used as the training dataset. Similar to Section 4.3, we evenly divided the training data into 10 subsets, each containing 10% of the training samples. It is important to note that we did not shuffle these time series datasets, meaning the 10 subsets were arranged sequentially from older to more recent data. We trained an initial model using the first subset, then incrementally added each subsequent subset one by one. After incorporating all training data, we sequentially removed each subset in reverse order. As expected, since the test dataset spans all time steps, the error rate decreases as more subsets are added to the model. This is because the model learns the updated distribution from the newly added subsets. After removing each subset, the error rate increases, reflecting the loss of information associated

Table 8: The test error after training, adding, and deleting.

| Task | | Method | Adult | CreditInfo | SUSY | HIGGS | Optdigits | Pendigits | Letter | Covtype | Abalone (×10⁻³) | WineQuality (×10⁻²) |
|---|---|---|---|---|---|---|---|---|---|---|---|---|
| | | iGBDT | 0.1276 | 0.0629 | 0.1987 | 0.2742 | 0.0290 | 0.0295 | 0.0418 | 0.1702 | 5.7721 | 1.2085 |
| | | DeltaBoost | 0.1938 | 0.1830 | 0.2428 | OOM | 0.0830 | 0.0824 | 0.1704 | 0.3024 | 8.3571 | 1.3475 |
| | | MU in GBDT | 0.1276 | 0.0629 | 0.1987 | 0.2742 | 0.0307 | 0.0294 | 0.0418 | 0.1702 | 5.7721 | 1.2085 |
| Test Error | | XGBoost | 0.1375 | 0.0659 | 0.1976 | 0.2676 | 0.0395 | 0.0355 | 0.0384 | 0.1717 | 5.7657 | 1.1193 |
| | | LightGBM | 0.1287 | 0.0631 | 0.1985 | 0.2726 | 0.0334 | 0.0355 | 0.0374 | 0.1700 | 5.9304 | 1.1995 |
| | | CatBoost | 0.2928 | 0.1772 | 0.4324 | 0.5384 | 0.0618 | 0.0440 | 0.0655 | 0.1572 | 5.7265 | 1.2457 |
| | | ThunderGBM (GPU) | 0.2405 | 0.0660 | 0.4576 | 0.4698 | 0.0546 | 0.0515 | 0.0940 | 0.2135 | 8.1791 | 1.6482 |
| | | Ours | 0.1276 | 0.0629 | 0.1987 | 0.2742 | 0.0307 | 0.0294 | 0.0418 | 0.1702 | 5.7721 | 1.2085 |
| | 1 | iGBDT | 0.1279 | 0.0633 | 0.1987 | 0.2769 | 0.0301 | 0.0286 | 0.0418 | 0.1696 | 5.8801 | 1.1953 |
| | | Ours | 0.1275 | 0.0630 | 0.1988 | 0.2742 | 0.0295 | 0.0297 | 0.0404 | 0.1685 | 5.8110 | 1.2079 |
| Incre. Learning | 0.10% | iGBDT | 0.1267 | 0.0630 | 0.1995 | 0.2742 | 0.0323 | 0.0363 | 0.0446 | 0.1777 | 6.2531 | 1.2680 |
| | | Ours | 0.1269 | 0.0626 | 0.1989 | 0.2747 | 0.0295 | 0.0297 | 0.0406 | 0.1686 | 5.9000 | 1.2040 |
| | 0.50% | iGBDT | 0.1287 | 0.0636 | 0.2012 | 0.2795 | 0.0390 | 0.0440 | 0.0572 | 0.1788 | 7.6510 | 1.2907 |
| | | Ours | 0.1294 | 0.0632 | 0.1988 | 0.2734 | 0.0290 | 0.0295 | 0.0394 | 0.1681 | 5.7701 | 1.2198 |
| | 1% | iGBDT | 0.1291 | 0.0630 | 0.2014 | 0.2780 | 0.0529 | 0.0603 | 0.0875 | 0.1868 | 8.5324 | 1.4462 |
| | | Ours | 0.1267 | 0.0632 | 0.1990 | 0.2740 | 0.0262 | 0.0283 | 0.0440 | 0.1683 | 5.8378 | 1.2209 |
| | 1 | DeltaBoost | 0.1971 | 0.1852 | 0.2460 | OOM | 0.0837 | 0.0812 | 0.1755 | 0.3103 | 8.5831 | 1.3380 |
| | | MU in GBDT | 0.1280 | 0.0629 | 0.1987 | 0.2742 | 0.0306 | 0.0295 | 0.0408 | 0.1702 | 5.8025 | 1.2095 |
| | | Ours | 0.1276 | 0.0628 | 0.1987 | 0.2742 | 0.0306 | 0.0295 | 0.0416 | 0.1702 | 5.8723 | 1.2143 |
| Decre. Learning | 0.10% | DeltaBoost | 0.2003 | 0.1788 | 0.2387 | OOM | 0.0848 | 0.0835 | 0.1690 | 0.2966 | 8.5054 | 1.3681 |
| | | MU in GBDT | 0.1285 | 0.0634 | 0.1988 | 0.2742 | 0.0301 | 0.0295 | 0.0444 | 0.1734 | 5.9727 | 1.2202 |
| | | Ours | 0.1284 | 0.0633 | 0.1988 | 0.2747 | 0.0295 | 0.0283 | 0.0432 | 0.1712 | 5.8744 | 1.2109 |
| | 0.50% | DeltaBoost | 0.1920 | 0.1870 | 0.2476 | OOM | 0.0821 | 0.0843 | 0.1728 | 0.2998 | 8.4328 | 1.3227 |
| | | MU in GBDT | 0.1309 | 0.0640 | 0.1988 | 0.2751 | 0.0306 | 0.0283 | 0.0442 | 0.1727 | 6.3142 | 1.2398 |
| | | Ours | 0.1295 | 0.0634 | 0.1988 | 0.2746 | 0.0301 | 0.0303 | 0.0432 | 0.1675 | 5.7733 | 1.2052 |
| | 1% | DeltaBoost | 0.2012 | 0.1814 | 0.2519 | OOM | 0.0861 | 0.0830 | 0.1761 | 0.3135 | 8.7275 | 1.3590 |
| | | MU in GBDT | 0.1311 | 0.0639 | 0.1988 | 0.2745 | 0.0334 | 0.0312 | 0.0460 | 0.1766 | 6.3558 | 1.2925 |
| | | Ours | 0.1295 | 0.0632 | 0.1987 | 0.2747 | 0.0273 | 0.0303 | 0.0424 | 0.1695 | 5.7620 | 1.2111 |

Table 9: Error rate after every online learning step.

| Online Learning Step | GlobalTemperatures (×10⁻³) | WebTraffic (×10⁻³) |
|---|---|---|
| Initial Train 10% | 4.1934 | 4.0984 |
| Add 10%, Total 20% | 2.5431 | 3.8383 |
| Add 10%, Total 30% | 2.1156 | 3.0296 |
| Add 10%, Total 40% | 2.0351 | 3.1297 |
| Add 10%, Total 50% | 1.9593 | 2.9149 |
| Add 10%, Total 60% | 1.8940 | 2.9525 |
| Add 10%, Total 70% | 1.8973 | 2.8682 |
| Add 10%, Total 80% | 1.8532 | 2.9024 |
| Add 10%, Total 90% | 1.8200 | 2.9141 |
| Add 10%, Total 100% | 1.7850 | 2.9049 |
| Del 10%, Total 90% | 1.8127 | 2.8432 |
| Del 10%, Total 80% | 1.9902 | 3.3453 |
| Del 10%, Total 70% | 2.0115 | 2.9007 |
| Del 10%, Total 60% | 2.1137 | 3.1288 |
| Del 10%, Total 50% | 2.0756 | 3.1187 |
| Del 10%, Total 40% | 2.1654 | 2.9539 |
| Del 10%, Total 30% | 2.1349 | 3.0132 |
| Del 10%, Total 20% | 2.4975 | 3.8429 |
| Del 10%, Total 10% | 3.6064 | 4.4339 |

with the removed data and the model's adjustment to the remaining subsets. As shown in Table 9, these results confirm the effectiveness of our method in adapting to changing data distributions.

Table 10: Model functionality change after online learning.

| Dataset | Metric | iGBDT (Incr.) | | Ours (Incr.) | | DeltaBoost (Decr.) | | MUinGBDT (Decr.) | | Ours (Decr.) | |
|---|---|---|---|---|---|---|---|---|---|---|---|
| | | Add 1 | Add 0.1% | Add 1 | Add 0.1% | Del 1 | Del 0.1% | Del 1 | Del 0.1% | Del 1 | Del 0.1% |
| Adult | C2W ↓ | 0.40% | 0.93% | 0.17% | 0.61% | 1.17% | 1.87% | 0.63% | 0.51% | 0.55% | 0.51% |
| | W2C ↓ | 0.27% | 0.80% | 0.18% | 0.56% | 0.72% | 1.28% | 0.60% | 0.73% | 0.56% | 0.68% |
| | $\phi$ ↑ | 99.34% | 98.27% | 99.66% | 98.83% | 98.11% | 96.85% | 98.77% | 98.76% | 98.88% | 98.82% |
| CreditInfo | C2W ↓ | 0.21% | 0.40% | 0.16% | 0.30% | 0.58% | 0.92% | 0.10% | 0.21% | 0.10% | 0.18% |
| | W2C ↓ | 0.18% | 0.40% | 0.15% | 0.29% | 0.08% | 0.13% | 0.08% | 0.23% | 0.08% | 0.19% |
| | $\phi$ ↑ | 99.60% | 99.20% | 99.70% | 99.41% | 99.34% | 98.96% | 99.82% | 99.56% | 99.82% | 99.63% |
| SUSY | C2W ↓ | 0.25% | 0.82% | 0.22% | 0.74% | 3.50% | 3.40% | 0% | 0.78% | 0% | 0.73% |
| | W2C ↓ | 0.24% | 0.78% | 0.21% | 0.73% | 1.34% | 1.14% | 0% | 0.79% | 0% | 0.76% |
| | $\phi$ ↑ | 99.51% | 98.40% | 99.58% | 98.53% | 95.16% | 95.46% | 100% | 98.43% | 100% | 98.51% |
| HIGGS | C2W ↓ | 0.00% | 2.52% | 0% | 2.64% | OOM | | 0% | 1.92% | 0% | 1.92% |
| | W2C ↓ | 0.00% | 2.56% | 0% | 2.63% | | | 0% | 1.93% | 0% | 1.92% |
| | $\phi$ ↑ | 100.00% | 94.92% | 100% | 94.73% | | | 100% | 96.14% | 100% | 96.17% |
| Optdigits | C2W ↓ | 0.33% | 0.56% | 0.17% | 0.28% | 0.22% | 0.56% | 0.61% | 0.45% | 0.45% | 0.61% |
| | W2C ↓ | 0.56% | 0.61% | 0.28% | 0.50% | 0.28% | 0.22% | 0.22% | 0.33% | 0.28% | 0.39% |
| | W2W ↓ | 0.06% | 0.11% | 0.06% | 0% | 0.17% | 0.11% | 0.06% | 0.11% | 0.06% | 0.06% |
| | $\phi$ ↑ | 99.05% | 98.72% | 99.50% | 99.22% | 99.33% | 99.11% | 99.11% | 99.11% | 99.22% | 98.94% |
| Pendigits | C2W ↓ | 0.26% | 0.83% | 0.14% | 0.17% | 0.17% | 0.09% | 0.29% | 0.26% | 0.26% | 0.23% |
| | W2C ↓ | 0.14% | 0.43% | 0.11% | 0.17% | 0.26% | 0.37% | 0.17% | 0.20% | 0.23% | 0.20% |
| | W2W ↓ | 0.06% | 0.20% | 0.06% | 0.03% | 0.03% | 0.09% | 0.06% | 0.09% | 0.03% | 0.09% |
| | $\phi$ ↑ | 99.54% | 98.54% | 99.69% | 99.63% | 99.54% | 99.46% | 99.49% | 99.46% | 99.49% | 99.49% |
| Letter | C2W ↓ | 0.74% | 1.62% | 0.64% | 0.68% | 0.52% | 0.80% | 1.24% | 1.36% | 1.26% | 1.40% |
| | W2C ↓ | 0.82% | 0.88% | 0.78% | 0.80% | 0.58% | 0.62% | 1.06% | 1.42% | 1.06% | 1.38% |
| | W2W ↓ | 0.28% | 0.44% | 0.30% | 0.30% | 0.20% | 0.40% | 0.44% | 0.24% | 0.42% | 0.28% |
| | $\phi$ ↑ | 98.16% | 97.06% | 98.28% | 98.22% | 98.70% | 98.18% | 97.26% | 96.98% | 97.26% | 96.94% |
| Covtype | C2W ↓ | 0.98% | 2.37% | 1.78% | 1.78% | 0.11% | 0.61% | 1.94% | 2.04% | 1.94% | 1.96% |
| | W2C ↓ | 1.15% | 2.10% | 1.77% | 1.77% | 0.14% | 0.70% | 1.80% | 1.76% | 1.80% | 1.71% |
| | W2W ↓ | 0.04% | 0.09% | 0.07% | 0.07% | 0.02% | 0.03% | 0.06% | 0.07% | 0.06% | 0.07% |
| | $\phi$ ↑ | 97.83% | 95.44% | 96.38% | 96.38% | 99.74% | 98.66% | 96.19% | 96.13% | 96.20% | 96.26% |

## K  MODEL FUNCTIONAL SIMILARITY

As mentioned in Section 2.2, the goal of the framework is to find a model close to the model retrained from scratch. The model functional similarity is a metric to evaluate how close the model learned by dynamic learning and the one retrained from scratch. We show the model functional similarity for incremental learning and decremental learning in Table 10. C2W refers to the ratio of testing instances that are correctly predicted during retraining but are wrongly predicted after decremental learning. Similarly, W2C represents the testing instances that are wrongly predicted during retraining but are correctly predicted after decremental learning. The W2W column indicates the cases where the two models have different wrong predictions. For binary labels, W2W is not applicable. In the $|D'|$ column, 1 indicates that only add/remove one instance, while 0.1% corresponds to $|D'| = 0.1\% \times |D_{tr}|$. We present $\phi$ to evaluate the model functional similarity (adapted from the model functionality (Adi et al., 2018)), indicating the leakage of dynamic learning:

**Definition 3** (Functional Similarity) *Given an input space $\mathcal{X}$, a model $T$, a model $\hat{T}$ dynamic learned from $T$, and a dataset $D = \{y_i, \mathbf{a}_i\} \in \mathcal{X}$, the functional similarity $\phi$ between model $T$ and $\hat{T}$ is:* $\phi = 1 - (r_{w2w} + r_{w2c} + r_{c2w})$ *,where $\phi$ is the leakage of learning.*

Due to the size limitations of the table, we have omitted OnlineGB from this table because its learning duration is excessively long, making it relatively meaningless compared to other methods. We compared iGBDT in adding 1 and 0.1% data instances, and DeltaBoost and MUinGBDT in deleting data. As shown in Table 10, we have a comparable model functionality in adding/deleting both 1 and 0.1%. In most cases, DyGB reaches 98% similarity in both incremental and decremental learning.

## L  BACKDOOR ATTACKING

**Experimental Setup.** In this evaluation, we randomly select a subset of the training dataset, and set first a few features to a specific value (trigger, e.g. 0 or greatest feature value) on these data instances, and then set the label to a target label (e.g., 0). In the testing dataset, we set all labels to the target label to compose a backdoor test dataset. In this setting, if the model has correctly learned the trigger and target label, it should achieve a high accuracy on backdoor test dataset.

## M  MEMBERSHIP INFERENCE ATTACK

The membership inference attack (MIA) aims to predict whether a data sample is part of the training dataset (Shokri et al., 2017; Hu et al., 2022; Choquette-Choo et al., 2021). Therefore, the goal of this experiments is to determine if "deleted" data can still be identified as training data after decremental learning. However, in our experiment with default hyper-parameter setting, the predictions made by MIA are nearly random guesses.

**Experimental Setup.** Previous studies demonstrate that overfitting can make machine learning models more vulnerable to MIA (Yeom et al., 2018; van Breugel et al., 2023; Hu et al., 2022). To further validate our approach, we apply a smaller model with the number of iterations $M = 5$, which can be easily overfitted. For overfitting the model, we split each dataset into three subsets: base dataset $D_{\text{base}}$ (49.9%), dynamic dataset $D'$ (0.1%), and test dataset $D_{\text{test}}$ (50%). We first train a base model on $D_{\text{base}} + D'$. For this base model, the MIA should identify the data in $D'$ as part of the training dataset. Next, we perform decremental learning to delete $D'$ from the base model. After this process, the MIA should no longer identify the data in $D'$ as part of the training dataset, confirming that our approach effectively deletes the data from the model. Finally, we add $D'$ back to the model by incremental learning. Following this, the MIA should once again identify the data in $D'$ as part of the training dataset. These experiments are conducted on multi-class datasets: Optdigits, Pendigits, Letter, and Covtype.

**MIA Model.** By following the existing MIA methods (Yan et al., 2023; Li et al., 2022; Carlini et al., 2022), we train an MIA model (binary classification) on the prediction probabilities of each class. Since the GBDT model is overfitted, the probability distributions of the training data should substantially differ from those of the unseen data (test data). Therefore, the MIA model can predict whether a data sample is part of the training dataset based on its probability distribution. We sample 50% of $D_{\text{base}}$ and 50% of $D_{\text{test}}$ to train the MIA model. Then remaining 50% of $D_{\text{base}}$, the entire $D'$ and 50% of $D_{\text{test}}$ are used for evaluation.

Table 11: Membership Inference Attack.

| Dataset | Base Model | | | After decremetal learning | | | After incremetal learning | | |
|---|---|---|---|---|---|---|---|---|---|
| | $D_{\text{base}}$ | $D'$ | $D_{\text{test}}$ | $D_{\text{base}}$ | $D'$ | $D_{\text{test}}$ | $D_{\text{base}}$ | $D'$ | $D_{\text{test}}$ |
| Optdigits | 100% | 100% | 43.59% | 100% | 33.93% | 42.19% | 100% | 100% | 43.82% |
| Pendigits | 100% | 100% | 56.09% | 100% | 55.04% | 46.15% | 100% | 100% | 56.63% |
| Letter | 100% | 100% | 26.31% | 100% | 13.33% | 47.37% | 100% | 100% | 36.84% |
| Covtype | 100% | 100% | 38.89% | 100% | 15.2% | 38.89% | 100% | 100% | 44.31% |

**Results.** Table 11 presents the average probability of data samples being identified as part of the training dataset at different stages. For the base model, MIA identifies 100% of the data in $D_{\text{base}}$ and $D'$ as part of the training dataset, while the data in $D_{\text{test}}$ has a low probability of being identified as part of the training dataset. After decremental learning, the probability for $D_{\text{base}}$ remains unchanged, while the probability for $D'$ drops to a level almost identical to $D_{\text{test}}$. This confirms that $D'$ has been effectively deleted from the base model. After incremental learning, the probability for $D'$ increases to 100% again, indicating that the model has successfully relearned $D'$. The probability for $D_{\text{test}}$ in the incremental model remains almost the same as in the base model. This result confirms that our decremental/incremental learning approach can indeed delete/add data from/to the model.

## N  DIFFERENT BASE LEARNER

Since the proposed method is designed for decision trees, we conducted an experiment to compare it with the boosted linear regression (linear model). For the linear model, we set the maximum number of iterations to 1,000 and enabled early stopping. As shown in Table 12, our method consistently demonstrates superior accuracy, achieving lower error rates across all datasets. Although our method requires more memory and longer training time than the linear model, its incremental and decremental learning on a single data point is substantially faster than retraining from scratch.

Table 12: Comparison with linear model as base learner (max_iteration = 1,000, early_stop = True).

| Metrics | Method | Adult | CreditInfo | SUSY | HIGGS | Optdigits | Pendigits | Letter | Covtype |
|---|---|---|---|---|---|---|---|---|---|
| Memory (MB) | Linear Model | 167.78 | 22.44 | 3,671.12 | 12,144.70 | 162.11 | 160.03 | 161.97 | 1,192.70 |
| | Ours | 577.18 | 1,096.71 | 16,576.40 | 24,333.30 | 1,081.15 | 1,959.49 | 1,805.76 | 9,665.21 |
| Error Rate | Linear Model | 0.1877 | 0.0657 | 0.2119 | 0.358 | 0.0557 | 0.1075 | 0.3582 | 0.2876 |
| | Ours | 0.1276 | 0.0629 | 0.1987 | 0.2742 | 0.0307 | 0.0294 | 0.0418 | 0.1702 |
| Time (s) | Linear Model | 0.163 | 0.203 | 7.94 | 13.314 | 0.091 | 0.088 | 0.421 | 6.174 |
| | Ours (Training) | 2.673 | 1.818 | 64.935 | 177.1 | 0.276 | 0.368 | 0.352 | 9.336 |
| | Add 1 | 0.035 | 0.114 | 1.678 | 5.488 | 0.011 | 0.014 | 0.016 | 0.29 |
| | Del 1 | 0.034 | 0.055 | 1.303 | 3.367 | 0.01 | 0.015 | 0.014 | 0.161 |

## O  EXTREMELY HIGH-DIMENSIONAL DATASETS

We include two dataset with more features / high dimensional: RCV1 and News20, which have 47,236 and 1,355,191 features respectively. For News20 dataset, the substantial high dimension causes segmentation fault on CatBoost and GPU out of memory (OOM) on thunderGBM. We omit the results from the other incremental/decremental method because infeasible running time and massive occupied memory. Table 14 shows the comparison of the training time and memory usage for our methods and other popular methods. Table 15 illustrates the incremental and decremental learning time of our method for two high dimensional dataset.

Table 13: Dataset specifications.

| Dataset | # Train | # Test | # Dim | # Class |
|---|---|---|---|---|
| News20 | 5,000 | 14,996 | 1,355,191 | 2 |
| RCV1 | 20,242 | 677,399 | 47,236 | 2 |

Table 14: Comparison of the training time consumption and memory usage for RCV1 and News20.

| | Dataset | XGBoost | LightGBM | CatBoost | ThunderGMB (GPU) | Ours |
|---|---|---|---|---|---|---|
| Training Time (s) | RCV1 | 459.75 | 59.63 | 335.70 | 49.44 | 295.43 |
| | News20 | 637.02 | 28.42 | Seg. Fault | OOM | 225.73 |
| Memory ($MB$) | RCV1 | 3,008.28 | 2,922.32 | 263.63 | 1,913.05 | 185,851.72 |
| | News20 | 3,061.99 | 2,509.29 | Seg. Fault | OOM | 128,131.43 |

Table 15: The incremental/decremental learning time of the proposed method for RCV1 and News20. (ms, per tree, incre./decre.)

| Dataset | $|D'|$ | Learning Time (Ours) | Incremental Learning Speedup v.s. | | | | Learning Time (Ours) | Decremental Learning Speedup v.s. | | | |
|---|---|---|---|---|---|---|---|---|---|---|---|
| | | | XGBoost | LightGBM | CatBoost | ThunderGBM (GPU) | | XGBoost | LightGBM | CatBoost | ThunderGBM (GPU) |
| RCV1 | 1 | 21.431 | 214.5x | 27.8x | 156.6x | 23.1x | 19.268 | 238.6x | 30.9x | 174.2x | 25.7x |
| | 0.1% | 37.707 | 121.9x | 15.8x | 89.0x | 13.1x | 29.232 | 157.3x | 20.4x | 114.8x | 16.9x |
| | 0.5% | 39.428 | 116.6x | 15.1x | 85.1x | 12.5x | 48.218 | 95.3x | 12.4x | 69.6x | 10.3x |
| | 1% | 43.901 | 104.7x | 13.6x | 76.5x | 11.3x | 70.666 | 65.1x | 8.4x | 47.5x | 7.0x |
| News20 | 1 | 11.76 | 541.7x | 24.2x | - | - | 7.718 | 825.4x | 36.8x | - | - |
| | 0.1% | 17.113 | 372.2x | 16.6x | - | - | 12.363 | 515.3x | 23.0x | - | - |
| | 0.5% | 22.261 | 286.2x | 12.8x | - | - | 30.076 | 211.8x | 9.5x | - | - |
| | 1% | 23.469 | 271.4x | 12.1x | - | - | 37.825 | 168.4x | 7.5x | - | - |

## P  DATA ADDITION WITH MORE CLASSES

DyGB can update data with unseen classes. We divide the dataset into sub-datasets based on labels (e.g., Optdigits has 10 labels, so we divide it into 10 sub-datasets). We train a model on the first sub-dataset and test it on two test datasets: 1) the original full test dataset with all labels, and 2) the partial test dataset with only the learned labels. We fine-tune the model with a new sub-dataset through incremental learning until learning the full dataset, testing the model on both test datasets after each training. Figure 5 shows that the accuracy of incremental learning and retraining is nearly identical on both the full and partial datasets. Note that the decrease in accuracy on the partial dataset is likely due to the increasing complexity of the learned data, which leads to a decrease in accuracy.

Table 16: The approximation error of leave's score between the model after addition/delection and the model retrained from scratch. Appr. Error $= \frac{\sum_{\text{all trees}} \sum_{\text{all leaves}} \text{abs}(p_{\text{add/del}} - p_{\text{retrain}})}{\sum_{\text{all trees}} \sum_{\text{all leaves}} \text{abs}(p_{\text{retrain}})}$, where $p_{\text{add/del}}$ is the leave's score after adding/deleting, $p_{\text{retrain}}$ is the leave's score of the model retraining from scratch.

|          | Adult  | CreditInfo | SUSY  | HIGGS | Optdigits | Pendigits | Letter | Covtype |
|----------|--------|------------|-------|-------|-----------|-----------|--------|---------|
| Add 1    | 2.42%  | 1.18%      | 0.24% | 0.00% | 2.69%     | 2.23%     | 1.31%  | 0.17%   |
| Add 0.1% | 4.59%  | 6.57%      | 2.73% | 1.63% | 3.48%     | 4.12%     | 5.78%  | 9.47%   |
| Add 0.5% | 5.10%  | 7.44%      | 2.27% | 3.05% | 5.12%     | 4.50%     | 10.45% | 11.68%  |
| Add 1%   | 5.30%  | 7.43%      | 3.07% | 3.89% | 5.92%     | 4.70%     | 11.75% | 10.01%  |
| Add 10%  | 4.25%  | 8.33%      | 1.07% | 1.73% | 4.64%     | 4.42%     | 13.34% | 4.96%   |
| Add 50%  | 3.55%  | 0.00%      | 0.00% | 1.51% | 0.00%     | 0.00%     | 6.26%  | 0.01%   |
| Add 80%  | 0.00%  | 0.00%      | 0.00% | 0.00% | 0.00%     | 0.00%     | 0.00%  | 0.00%   |
| Del 1    | 1.21%  | 0.00%      | 0.00% | 0.00% | 0.01%     | 0.19%     | 0.57%  | 0.28%   |
| Del 0.1% | 3.63%  | 3.80%      | 0.79% | 0.72% | 1.40%     | 0.50%     | 1.88%  | 4.31%   |
| Del 0.5% | 3.58%  | 3.76%      | 0.18% | 0.56% | 2.52%     | 1.15%     | 3.49%  | 6.04%   |
| Del 1%   | 3.40%  | 3.16%      | 0.15% | 0.65% | 3.07%     | 1.73%     | 3.74%  | 4.48%   |
| Del 10%  | 0.27%  | 0.39%      | 0.00% | 0.16% | 1.67%     | 0.97%     | 1.35%  | 0.46%   |
| Del 50%  | 0.00%  | 0.00%      | 0.00% | 0.00% | 0.00%     | 0.00%     | 0.00%  | 0.00%   |
| Del 80%  | 0.00%  | 0.00%      | 0.00% | 0.00% | 0.00%     | 0.00%     | 0.00%  | 0.00%   |

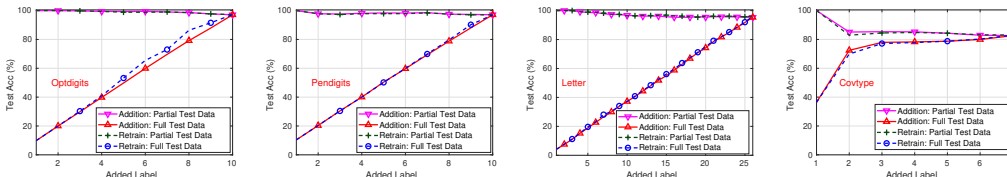

Figure 5: The impact of tuning data size on the number of retrained nodes for each iteration in incremental learning.

## Q APPROXIMATION ERROR OF LEAF SCORES

As mentioned in Section 3.2, outdated derivatives are used in gain computation to reduce the cost of updating derivatives. However, these outdated derivatives are only applied to nodes where the best split remains unchanged. When a sub-tree requires retraining, the derivatives are updated. Therefore, using outdated derivatives typically occurs when $|D'|$ is small, as fewer data modifications result in fewer changes to the best splits. Conversely, when more data is added or deleted, $|D'|$ becomes larger, increasing the likelihood of changes to the best splits in some nodes. As a result, the sub-trees are retrained, and the derivatives for the data reaching those nodes are updated.

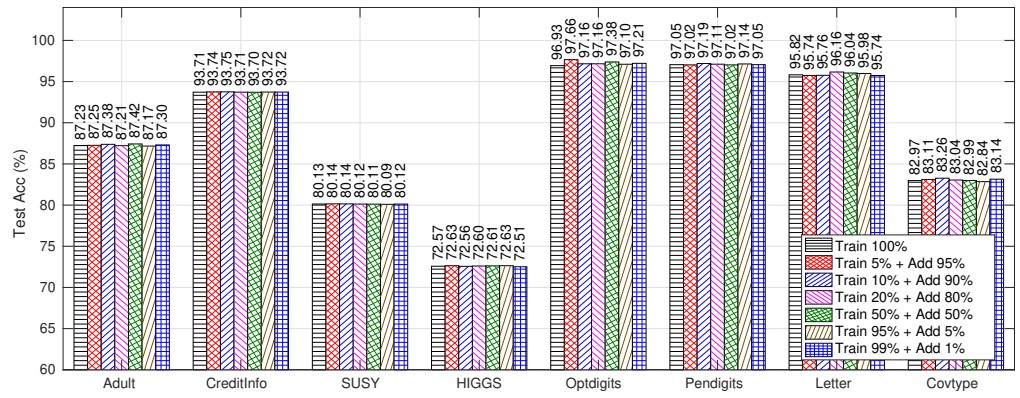

Figure 6: Different fine-tuning ratio.

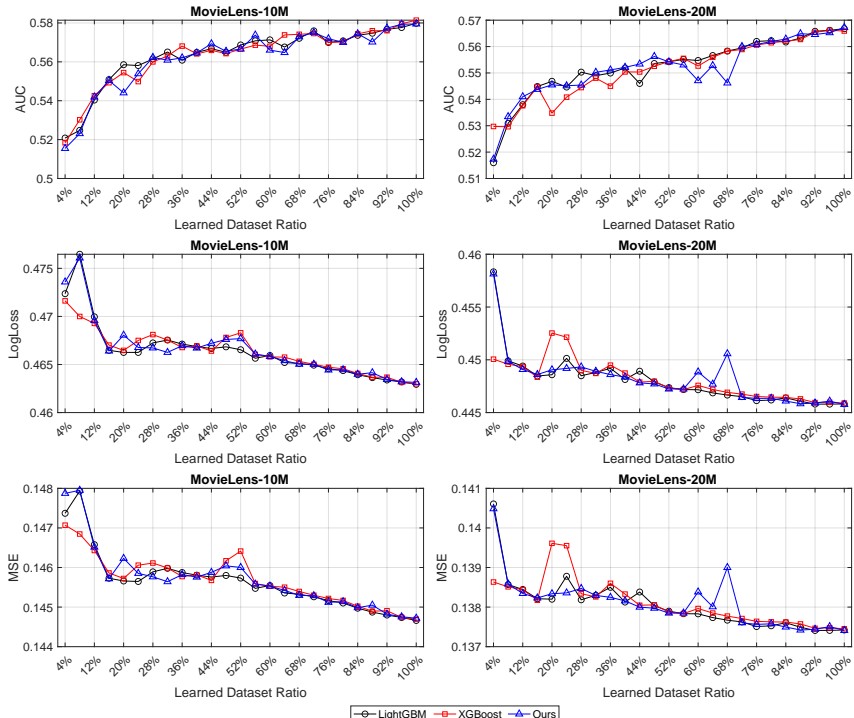

Figure 7: **(Incremental Learning)** Performance (AUC, Log Loss, and MSE) of the recommender system on different dataset ratios. We first sort the entire dataset by timestamp, from oldest to most recent. Then, we partition the oldest 80% as the training dataset and the most recent 20% as the testing dataset. To evaluate our proposed method, we initially train a model on the oldest 4% of the training data, then gradually learn every additional 4% via incremental learning until the full training dataset (100%) is used. These results illustrate that more recent data can positively impact the performance of the recommender system.

To confirm the effect of using outdated derivatives during dynamic learning, we report the result for the approximation error of leaf scores in Table 16. Appr. Error $= \frac{\sum_{\text{all trees}} \sum_{\text{all leaves}} \text{abs}(p_{\text{add/del}} - p_{\text{retrain}})}{\sum_{\text{all trees}} \sum_{\text{all leaves}} \text{abs}(p_{\text{retrain}})}$, where $p_{\text{add/del}}$ is the leaf score after adding/deleting, and $p_{\text{retrain}}$ is the leaf score of the model retraining from scratch. Please note that the retrained model has the same structure and split in all nodes of all trees as the model after adding/deleting, and we only update the latest residual and hessian to calculate the latest leaf score. When the number of added/deleted data increases, the error will increase because our method uses outdated derivatives if the best splits remain unchanged. When the number of add/delete is large enough, almost all nodes in the model will be retrained because their best splits have changed, so the error becomes 0.

## R    DYGB ON RECOMMENDER SYSTEMS

In the paper, we mention that a potential use case is recommendation systems. In this experiment, we show how the proposed method improves the performance of recommendation systems through incremental and decremental learning on GBDT.

**Interest-drift in Recommender Systems.** Interest drift refers to the evolution of a user's preferences over time. In recommendation systems, this means that past interactions may no longer accurately represent a user's current interests. As a result, relying on outdated data can degrade the performance of the system. To address this issue, previous studies have proposed time-weighted methods that gradually reduce the influence of older interactions (Yoon et al., 2008; Campos et al., 2014). However, instead of reducing their impact, completely removing outdated data can lead to better recommendation performance (Matuszyk et al., 2018; 2015; Tavakolian et al., 2012; Gordea & Zanker, 2007). Since the proposed GBDT supports both decremental learning and incremental

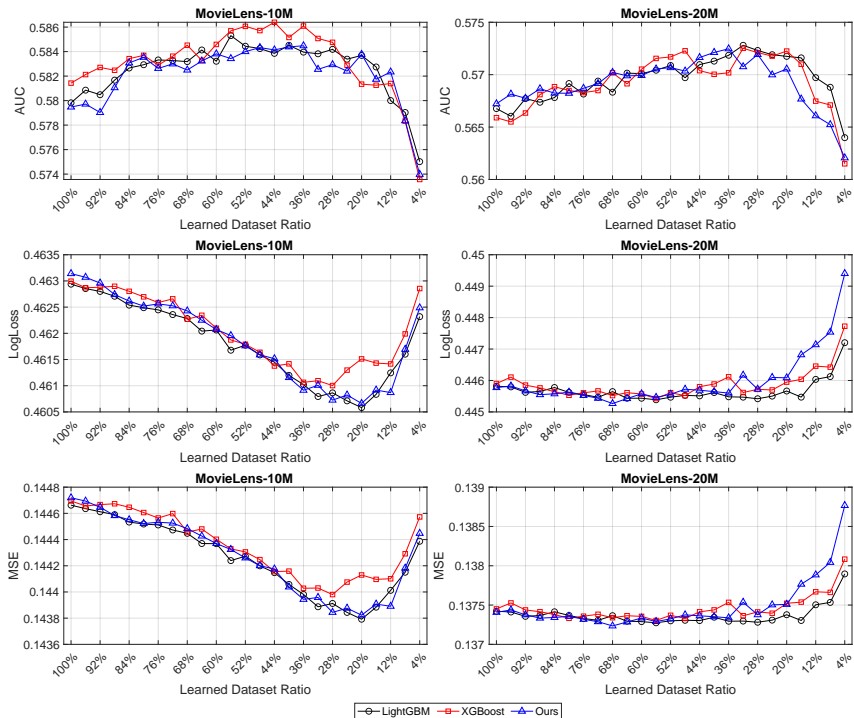

Figure 8: **(Decremental Learning)** Performance (AUC, Log Loss, and MSE) of the recommender system on different dataset ratios. We first sort the entire dataset by timestamp, from oldest to most recent. Then, we partition the oldest 80% as the training dataset and the remaining latest 20% as the testing dataset. To evaluate our proposed method, we initially train a model on the full training dataset (100%), then gradually remove 4% of the oldest data from the model via decremental learning until only 4% of the data remains. These results illustrate that outdated (oldest) data can negatively impact the performance of the recommender system.

learning, it naturally works on such recommender system, which can incrementally learn latest user behaviors and remove outdated behaviors without training from scratch.

**Datasets.** We use two large-scale datasets that include timestamps spanning long time periods: (1) **MovieLens-10M**: contains about 10 million ratings from 72,000 users on 10,000 movies from 1995 to 2009. (2) **MovieLens-20M**: contains about 20 million ratings from 138,000 users on 27,000 movies from 1995 to 2015. For each dataset, we sort the entire dataset by timestamps, from oldest to most recent. Then, we partition the oldest 80% as the training dataset and the remaining latest 20% as the testing dataset.

**Experimental Settings.** This experiment aims to answer the question: *Can the proposed method improve the performance of recommendation systems through incremental and decremental learning on GBDT?* To this end, we design two experiments to demonstrate the effectiveness of our approach through two key capabilities: (1) incrementally learning from the latest user behaviors, and (2) removing outdated behaviors without retraining the model from scratch.

**Incremental Learning.** This experiments is to confirm that incrementally learn the latest user behaviors improves the performance of the recommendation system. Our goal is to predict the Click-Through Rate (CTR) using LightGBM, XGBoost and our proposed GBDT. Recall the dataset processing, we partition the oldest 80% as the training dataset and the remaining latest 20% as the testing dataset. We further divide the training data into 25 segments, each accounting for 4% of the data. Our approach begins by training the model on the first (oldest) 4% of the data and incrementally incorporates each subsequent 4% partition in order. After each incremental update, we evaluate the model on the testing set using AUC, Log Loss, and MSE, as illustrated in Figure 7. For LightGBM and XGBoost, which do not support incremental learning natively, we retrain the models from scratch using the accumulated data up to the current partition at each step. Across

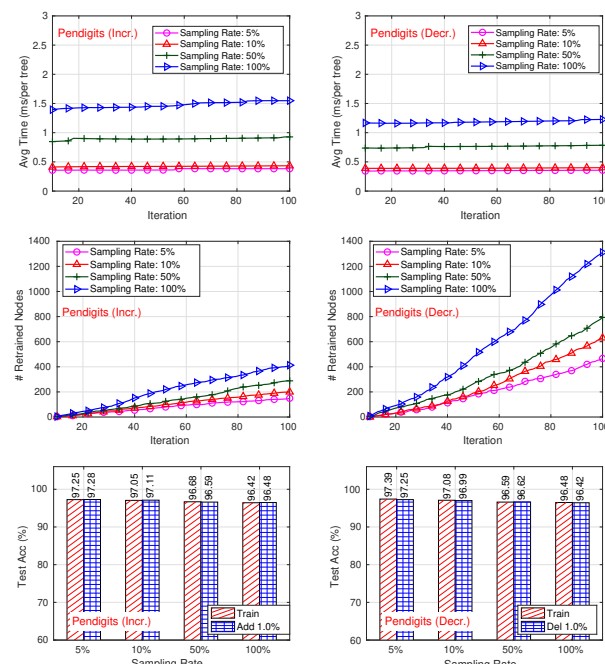

Figure 9: The impact of sampling rate on time, number of retrain nodes, and test accuracy during incremental/decremental learning.

both MovieLens-10M and MovieLens-20M datasets, all models improve as more recent data is incrementally learned. This demonstrates that learning from the latest user behaviors can improve recommendation effectiveness.

**Decremental Learning.** This experiment investigates whether removing outdated user behaviors can improve the performance of the recommendation system. Similar to the previous setup, we aim to predict the Click-Through Rate (CTR) using LightGBM, XGBoost, and our proposed GBDT model. We partition the dataset chronologically, using the oldest 80% as the training set and the latest 20% as the testing set. We start by training each model on the full training dataset (100%) and then gradually remove the oldest 4% of the data at each step. After each removal, we evaluate the model's performance on the fixed testing set using AUC, Log Loss, and MSE, as shown in Figure 8. For LightGBM and XGBoost, which lack native support for decremental updates, we retrain the models from scratch using the remaining data at each step. Across both MovieLens-10M and MovieLens-20M datasets, we observe a clear trend: model performance initially improves as stale (outdated) data is removed, but begins to degrade once too much data is discarded. This indicates that while removing outdated user behavior can help reduce noise and improve generalization, excessive data removal eventually harms performance due to loss of useful historical patterns.

**Conclusion.** The experimental results demonstrate that our proposed method can effectively improve recommender system performance by incrementally learning recent user behaviors and removing outdated data without the need to retrain from scratch. This highlights the model's adaptability and efficiency in capturing evolving user preferences over time.

### R.1 VARIANCE ANALYSIS AND STATISTICAL CONSISTENCY

To validate the reliability and consistency of our speedup claims, we conducted a variance analysis. The 5 independent runs were performed with different random seeds to measure the stability of the algorithm's complexity profile across diverse execution paths (e.g., initial splits and sampling choices). As shown in Table 17, the empirical results confirm the high statistical consistency of the DyGB algorithm. The standard deviation is consistently low (typically less than 3% of the mean runtime), demonstrating that the massive speedup margins reported are reliable across stochastic execution paths.

Table 17: The table reports the mean execution time (seconds) and standard deviation ($\pm\sigma$) over 5 independent runs for incremental and decremental learning on Adult and HIGGS datasets with varying update sizes ($|D'|$).

| Dataset | $\|D'\|$ | Incremental Learning (s) | Decremental Learning (s) |
|---|---|---|---|
| | 1 | $0.035 \pm 0.001$ | $0.034 \pm 0.001$ |
| Adult | 0.1% | $0.105 \pm 0.003$ | $0.104 \pm 0.003$ |
| | 0.5% | $0.215 \pm 0.003$ | $0.219 \pm 0.008$ |
| | 1% | $0.340 \pm 0.013$ | $0.381 \pm 0.014$ |
| | 1 | $5.495 \pm 0.235$ | $3.223 \pm 0.034$ |
| HIGGS | 0.1% | $26.651 \pm 0.779$ | $19.402 \pm 0.272$ |
| | 0.5% | $43.383 \pm 1.621$ | $49.419 \pm 1.451$ |
| | 1% | $66.961 \pm 1.463$ | $79.418 \pm 1.407$ |

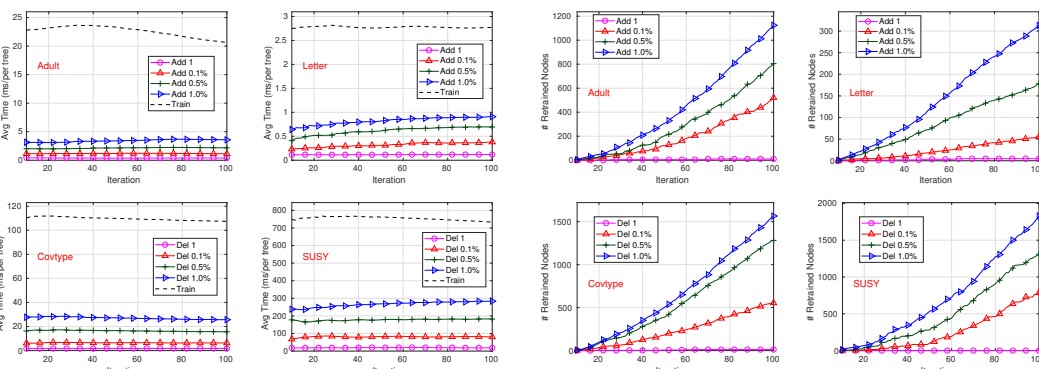

Figure 10: The impact of $|D'|$ on average learning time in incremental/decremental learning (top/bottom row).

Figure 11: The impact of $|D'|$ on the accumulated number of retrained nodes for each iteration in incr./decr. learning (top/bottom row).

## S  ABLATION STUDY

In this section, we discuss the impact of different hyper-parameter settings on the performance of DyGB, e.g., time and accuracy.

### S.1  SIZE OF DYNAMIC DATASET $|D'|$.

Different sizes of dynamic learning dataset $D'$ can have varying impacts on both the accuracy and time of the dynamic learning process. Figure 6 shows the impact of different data addition settings on test accuracy. Across all datasets, DyGB achieved nearly the same test accuracy, which validates the effectiveness of our dynamic learning framework. Decremental learning also has similar results.

Figure 10 shows the influence of $|D_{in}|$ on incremental/decremental learning time. We only present the experiment on 2 datasets each for incremental/decremental learning, due to the results on other datasets show a similar trend. These results show that the dynamic learning time increase when the size of $D_{in}$ increase. The reason is straightforward: as the size of $D_{in}$ increases, the model undergoes more significant changes, resulting in unstable splits. This leads to a greater number of sub-trees that require retraining, ultimately consuming more time. Figure 11 provides evidence to support this observation. It illustrates the accumulated number of retrained nodes – how many nodes need to be retrained. As the size of $D_{in}$ increases, the number of nodes that need to be retrained increases, leading to longer learning times.

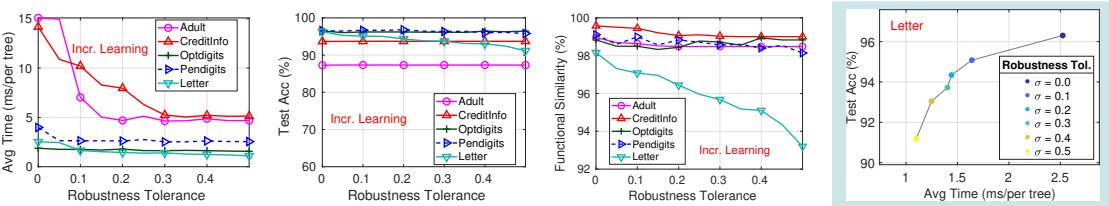

Figure 12: The impact of split robustness tolerance on the learning time, test accuracy, and model functional similarity $\phi$ in incremental learning.

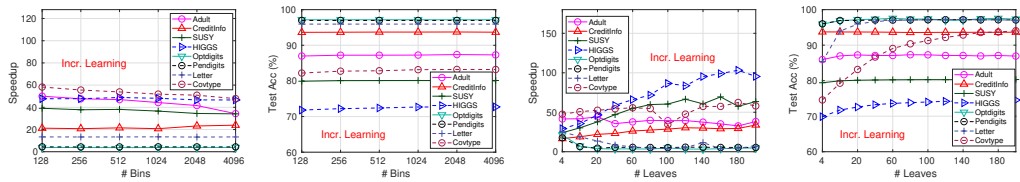

Figure 13: The impact of the # bins and # leaves on the acceleration factor of incremental learning (adding 1 data point).

## S.2 SPLIT RANDOM SAMPLING

Split random sampling is designed to reduce the frequency of retraining by limiting the number of splits. As mentioned in Section 3.3, a smaller sampling rate leads to more stable splits, resulting in fewer nodes that require retraining and shorter dynamic learning time. Figure 9 shows the impact of sampling rate $\alpha$ in split random sampling. The figures at the top demonstrate that when the sample rate is reduced, a smaller number of split candidates are taken into account, leading to an expected decrease in dynamic learning time. However, there is no significant difference between 5% and 10% in the Pendigits dataset. The figures in the second row show the accumulated number of retrained nodes. It also shows that as the sample rate decreases, the splits become more stable, resulting in fewer nodes that require retraining. In Pendigits, since the number of nodes that require retraining is similar for 5% and 10%, it results in a minimal difference in the dynamic learning time, as mentioned above. However, interestingly, for example in 100% sampling rate, although there are fewer retraining in incremental learning, it take more time during learning process, because incremental learning does not have derivatives of the data to be added. Therefore, more time is needed to calculate their derivatives. On the contrary, decremental learning can reuse the stored derivatives of the training process, resulting in less time. The bottom row shows the impact of the sampling rate on the test accuracy. The test accuracy remains almost identical across all sampling rates. Similar results can be observed in other datasets.

## S.3 SPLIT ROBUSTNESS TOLERANCE

Split robustness tolerance aims to enhance the robustness of a split in dynamic learning. As the observation in Figure 2, most best splits will be changed to second-best. Although the best split may change, we can avoid frequent retraining if we allow the split to vary within a certain range. For a node with $\lceil \alpha B \rceil$ potential splits, if the current split remains within the top $\lceil \sigma \alpha B \rceil$, we will continue using it. Here $\sigma$ ($0 \le \sigma \le 1$) is the robustness tolerance. Figure 12 illustrates the impact of split robustness tolerance $\sigma$ on learning time, test accuracy, and functional similarity $\phi$ in incremental learning. To obtain more pronounced experimental results, in this experiment, we set $|D'| = 1\% \times |D_{tr}|$.

The figure on the left shows that the learning time decreases as the tolerance level increases. Although test accuracy changes only slightly (middle figure), the functional similarity $\phi$ drops significantly (right figure). For example, in the Letter dataset, $\phi$ drops about 5% from $\sigma = 0$ to $\sigma = 0.5$. This demonstrates that higher tolerance levels result in faster learning by avoiding retraining, but with a trade-off of decreased functional similarity. Therefore, we suggest $\sigma$ should not be greater than 0.15. Similar results can be obtained on decremental learning.

Table 18: The test error rate after training, adding and deleting on GDBT with various iterations.

| | Method | Adult | | | | CreditInfo | | | | Optdigits | | | | Pendigits | | | | Letter | | | |
|---|---|---|---|---|---|---|---|---|---|---|---|---|---|---|---|---|---|---|---|---|---|
| | | 100 iter | 200 iter | 500 iter | 1000 iter | 100 iter | 200 iter | 500 iter | 1000 iter | 100 iter | 200 iter | 500 iter | 1000 iter | 100 iter | 200 iter | 500 iter | 1000 iter | 100 iter | 200 iter | 500 iter | 1000 iter |
| Training | XGBoost | **0.1270** | 0.1319 | 0.1379 | 0.1430 | 0.0630 | 0.0648 | 0.0663 | 0.0676 | 0.0418 | 0.0390 | 0.0412 | 0.0395 | 0.0397 | 0.0355 | 0.0352 | 0.0346 | 0.0524 | 0.0364 | 0.0356 | 0.0358 |
| | LightGBM | 0.1277 | 0.1293 | **0.1260** | **0.1318** | 0.0635 | 0.0636 | 0.0644 | 0.0654 | 0.0334 | 0.0317 | 0.0334 | 0.0329 | 0.0355 | 0.0343 | 0.0340 | 0.0340 | **0.0374** | **0.0310** | 0.0296 | 0.0298 |
| | CatBoost | 0.2928 | 0.2887 | 0.2854 | 0.2843 | 0.1772 | 0.1765 | 0.1765 | 0.1765 | 0.0618 | 0.0396 | 0.0293 | 0.0248 | 0.0440 | 0.0365 | 0.0281 | 0.0257 | 0.0655 | 0.0406 | **0.0252** | **0.0186** |
| | Ours | 0.1276 | **0.1265** | 0.1294 | 0.1325 | **0.0629** | **0.0632** | **0.0639** | **0.0648** | **0.0307** | **0.0251** | **0.0239** | **0.0239** | **0.0294** | **0.0280** | **0.0277** | **0.0277** | 0.0418 | 0.0318 | 0.0256 | 0.0246 |
| Ours (Incr. Learning) | Add 1 | 0.1275 | 0.1271 | 0.1287 | 0.1323 | 0.063 | 0.0635 | 0.0638 | 0.0644 | 0.0295 | 0.0262 | 0.0239 | 0.0239 | 0.0297 | 0.0275 | 0.0275 | 0.0275 | 0.0404 | 0.0330 | 0.0266 | 0.0260 |
| | Add 0.1% | 0.1269 | 0.1287 | 0.1313 | 0.1325 | 0.0626 | 0.0633 | 0.0631 | 0.0638 | 0.0295 | 0.0256 | 0.0256 | 0.0256 | 0.0297 | 0.0275 | 0.0277 | 0.0277 | 0.0406 | 0.0322 | 0.0250 | 0.0240 |
| | Add 0.5% | 0.1294 | 0.1276 | 0.1298 | 0.1316 | 0.0632 | 0.0629 | 0.0633 | 0.0648 | 0.029 | 0.0262 | 0.0256 | 0.0256 | 0.0295 | 0.0266 | 0.0283 | 0.0283 | 0.0394 | 0.0326 | 0.0270 | 0.0256 |
| | Add 1% | 0.1267 | 0.1279 | 0.1287 | 0.1337 | 0.0632 | 0.0630 | 0.0639 | 0.0646 | 0.0262 | 0.0228 | 0.0228 | 0.0228 | 0.0283 | 0.0272 | 0.0275 | 0.0277 | 0.044 | 0.0310 | 0.0246 | 0.0242 |
| Ours (Decr. Learning) | Del 1 | 0.1276 | 0.1266 | 0.1294 | 0.1324 | 0.0628 | 0.0632 | 0.0640 | 0.0647 | 0.0306 | 0.0251 | 0.0239 | 0.0239 | 0.0295 | 0.0283 | 0.0280 | 0.0280 | 0.0416 | 0.0318 | 0.0260 | 0.0242 |
| | Del 0.1% | 0.1284 | 0.1273 | 0.1288 | 0.1321 | 0.0633 | 0.0634 | 0.0640 | 0.0648 | 0.0295 | 0.0256 | 0.0245 | 0.0245 | 0.0283 | 0.0280 | 0.0280 | 0.0280 | 0.0432 | 0.0336 | 0.0272 | 0.0246 |
| | Del 0.5% | 0.1295 | 0.1266 | 0.1280 | 0.1327 | 0.0634 | 0.0631 | 0.0644 | 0.0646 | 0.0301 | 0.0245 | 0.0239 | 0.0239 | 0.0303 | 0.0289 | 0.0283 | 0.0283 | 0.0432 | 0.0320 | 0.0258 | 0.0244 |
| | Del 1% | 0.1295 | 0.1281 | 0.1290 | 0.1313 | 0.0632 | 0.0633 | 0.0638 | 0.0654 | 0.0273 | 0.0239 | 0.0234 | 0.0234 | 0.0303 | 0.0292 | 0.0280 | 0.0280 | 0.0424 | 0.0328 | 0.0270 | 0.0252 |

Table 19: The Total training, incremental or decremental learning time (in seconds).

| | Method | Adult | | | | CreditInfo | | | | Optdigits | | | | Pendigits | | | | Letter | | | |
|---|---|---|---|---|---|---|---|---|---|---|---|---|---|---|---|---|---|---|---|---|---|
| | | 100 iter | 200 iter | 500 iter | 1000 iter | 100 iter | 200 iter | 500 iter | 1000 iter | 100 iter | 200 iter | 500 iter | 1000 iter | 100 iter | 200 iter | 500 iter | 1000 iter | 100 iter | 200 iter | 500 iter | 1000 iter |
| Training | XGBoost | 9.467 | 19.128 | 43.064 | 103.767 | 13.314 | 34.619 | 77.706 | 78.845 | 0.752 | 1.385 | 2.598 | 5.271 | 0.574 | 1.743 | 3.225 | 5.976 | 1.171 | 3.647 | 8.097 | 14.597 |
| | LightGBM | 0.516 | 0.926 | 1.859 | 3.775 | 1.836 | 2.081 | 4.737 | 8.504 | 0.106 | 0.164 | 0.248 | 0.462 | 0.131 | 0.196 | 0.351 | 0.516 | 0.203 | 0.376 | 0.758 | 1.342 |
| | CatBoost | 1.532 | 2.646 | 5.805 | 10.974 | 3.447 | 5.467 | 12.002 | 13.339 | 0.177 | 0.262 | 1.160 | 2.360 | 0.183 | 0.399 | 1.104 | 1.986 | 0.232 | 0.524 | 1.475 | 3.196 |
| | Ours | 2.673 | 3.289 | 7.466 | 14.509 | 1.818 | 3.005 | 5.391 | 14.122 | 0.276 | 0.573 | 1.444 | 2.874 | 0.368 | 0.592 | 1.978 | 3.990 | 0.352 | 0.357 | 1.284 | 1.798 |
| Ours (Incr. Learning) | Add 1 | 0.035 | 0.071 | 0.167 | 0.328 | 0.114 | 0.125 | 0.244 | 0.616 | 0.011 | 0.031 | 0.118 | 0.285 | 0.014 | 0.045 | 0.142 | 0.227 | 0.016 | 0.018 | 0.206 | 0.464 |
| | Add 0.1% | 0.105 | 0.167 | 0.402 | 0.859 | 0.249 | 0.307 | 0.661 | 2.402 | 0.015 | 0.031 | 0.106 | 0.311 | 0.026 | 0.059 | 0.187 | 0.347 | 0.040 | 0.070 | 0.483 | 0.807 |
| | Add 0.5% | 0.212 | 0.383 | 0.937 | 2.463 | 0.321 | 0.593 | 1.502 | 4.670 | 0.029 | 0.039 | 0.137 | 0.335 | 0.042 | 0.062 | 0.194 | 0.411 | 0.067 | 0.127 | 0.537 | 0.979 |
| | Add 1% | 0.344 | 0.670 | 1.747 | 3.904 | 0.383 | 0.789 | 2.255 | 6.369 | 0.043 | 0.042 | 0.146 | 0.344 | 0.053 | 0.067 | 0.202 | 0.435 | 0.128 | 0.176 | 0.657 | 1.207 |
| Ours (Decr. Learning) | Del 1 | 0.034 | 0.128 | 0.177 | 0.179 | 0.055 | 0.265 | 0.359 | 0.342 | 0.010 | 0.007 | 0.037 | 0.092 | 0.015 | 0.012 | 0.067 | 0.165 | 0.014 | 0.007 | 0.007 | 0.011 |
| | Del 0.1% | 0.103 | 0.305 | 0.541 | 0.549 | 0.153 | 0.595 | 0.729 | 0.665 | 0.014 | 0.011 | 0.045 | 0.115 | 0.025 | 0.020 | 0.089 | 0.185 | 0.058 | 0.017 | 0.021 | 0.021 |
| | Del 0.5% | 0.222 | 0.753 | 1.481 | 1.467 | 0.251 | 0.941 | 1.217 | 1.220 | 0.029 | 0.024 | 0.065 | 0.123 | 0.041 | 0.038 | 0.106 | 0.198 | 0.103 | 0.035 | 0.041 | 0.038 |
| | Del 1% | 0.379 | 1.297 | 2.033 | 2.464 | 0.355 | 1.375 | 2.556 | 2.694 | 0.046 | 0.035 | 0.075 | 0.132 | 0.057 | 0.050 | 0.119 | 0.209 | 0.134 | 0.051 | 0.060 | 0.056 |

To further illustrate this tradeoff, the fourth subfigure presents the detailed time–accuracy relationship on the Letter dataset under six representative tolerance values ($\sigma = \{0, 0.1, 0.2, 0.3, 0.4, 0.5\}$). As the tolerance increases, the learning time decreases consistently due to reduced retraining, accompanied by a gradual decline in test accuracy. This dataset-level visualization highlights how tolerance directly shifts the balance between efficiency and accuracy.

Although the fourth subfigure shows the results for the Letter dataset, the other datasets exhibit the same qualitative pattern: larger tolerance values consistently reduce learning time while slightly compromising model accuracy. This confirms that the observed tradeoff is a general behavior across datasets, not an artifact of a particular domain.

Table 20: Accuracy for clean test dataset and attack successful rate for backdoor test dataset.

| # Iteration | Dataset | Train Clean | | Train Backdoor | | Add Backdoor | | Remove Backdoor | |
|---|---|---|---|---|---|---|---|---|---|
| | | Clean | Backdoor | Clean | Backdoor | Clean | Backdoor | Clean | Backdoor |
| 200 | Optdigits | 97.49% | 8.85% | 97.55% | 100.00% | 97.27% | 100.00% | 97.49% | 8.80% |
| | Pendigits | 97.28% | 5.06% | 97.25% | 100.00% | 97.25% | 100.00% | 100.00% | 11.67% |
| | Letter | 96.82% | 2.90% | 96.64% | 100.00% | 96.56% | 100.00% | 96.74% | 2.56% |
| 500 | Optdigits | 97.61% | 8.63% | 97.49% | 100.00% | 97.72% | 100.00% | 97.66% | 8.57% |
| | Pendigits | 97.23% | 5.06% | 97.14% | 100.00% | 97.28% | 100.00% | 97.25% | 5.63% |
| | Letter | 97.44% | 5.18% | 97.36% | 100.00% | 97.14% | 100.00% | 97.14% | 3.56% |
| 1000 | Optdigits | 97.61% | 8.63% | 97.77% | 100.00% | 97.72% | 100.00% | 97.83% | 10.30% |
| | Pendigits | 97.23% | 5.00% | 97.11% | 100.00% | 97.28% | 100.00% | 97.25% | 4.46% |
| | Letter | 97.66% | 5.18% | 97.38% | 100.00% | 97.52% | 100.00% | 97.42% | 11.18% |

## S.4 NUMBER OF BINS AND LEAVES

In dynamic learning procedure, the number of bins and leaves also affects the dynamic learning time. We report the impact of varying the number of bins $(128, 256, \cdots, 4096)$ and leaves $(4, 10, 20, 40, 60, \cdots, 200)$ on the acceleration factor of incremental learning (adding 1 data point) in Figure 13. The number of bins has few effect on both accuracy and the speed of dynamic learning as shown in the top row of the figures. In terms of the number of leaves, when it exceeds 20, the accuracy tends to stabilize, except for Covtype, as shown in the bottom row of the figures. For smaller datasets (Adult, Optdigits, Pendigits, Letter), the more the number of leaves, the lower the acceleration factor for incremental learning. However, for larger datasets (CreditInfo, SUSY, HIGGS, Covtype), the more the number of leaves, the greater the acceleration is. Especially for HIGGS, the largest dataset in our experiments, the acceleration can be more than 100x.

## S.5 Number of Iterations

The number of base learners is important in practical applications. We provide additional results for different numbers of base learners in Tables 18 and 19. Table 18 reports the test error rate after training, adding, and deleting base learners in GBDT models with varying iterations, demonstrating that DyGB achieves a comparable error rate across different iterations. Table 19 shows the time consumption for incremental and decremental learning, illustrating that DyGB are substantially faster than retraining a model from scratch, particularly in cases where a single data sample is added/deleted.

Additionally, to confirm that our method can effectively add and delete data samples across various iterations, we report results on backdoor attacks for different iterations, as shown in Table 20. These results confirm that our method successfully adds and removes data samples from the model across different numbers of iterations.

## S.6 Ablation Study Summary

The efficiency and accuracy of DyGB result from the synergistic relationship among our key optimizations. This ablation study clarifies the contribution of each component and establishes practical parameter guidelines.

**Primary Efficiency Lever ($\sigma$):** The Adaptive Split Robustness Tolerance ($\sigma$) is the critical mechanism for balancing speed and accuracy, as it directly controls the frequency of costly subtree retraining. Our analysis shows that this mechanism achieves massive time reduction, provided the functional similarity ($\phi$) remains within acceptable bounds.

**Guideline for $\sigma$:** For optimal trade-off, we recommend setting $\sigma \approx 0.1$. Higher tolerance levels (e.g., $\sigma > 0.15$) result in faster learning but with a significant risk of decreased functional similarity.

**Foundational Speedup ($\alpha$):** The Split Candidate Sampling ($\alpha$) provides the foundational speedup by drastically limiting the search space during both initial training and subsequent dynamic checks. Empirically, reducing $\alpha$ to $0.1$ is sufficient to achieve near-optimal time reductions without compromising test accuracy.

**Structural Scalability ($J$):** The Number of Leaves ($J$) is crucial for handling high-volume data. While accuracy stabilizes around $J = 20$, optimizing the number of leaves is essential for maximizing the acceleration factor on large datasets (CreditInfo, SUSY, HIGGS), where speedups can exceed 100x.

The comprehensive analysis of DyGB's hyperparameters confirms that the framework successfully achieves a robust balance between time-efficiency and structural stability. The core mechanism lies in the synergy between the Adaptive Split Robustness Tolerance ($\sigma$), which serves as the primary lever for minimizing costly retraining events, and the Split Candidate Sampling ($\alpha$), which reduces the baseline computational cost of each check. This combination validates DyGB's ability to perform continuous dynamic updates efficiently and reliably, ensuring that the necessary speedups are achieved without compromising the functional integrity of the trained model.

## T Impact of Sub-optimal Splits: Stochastic Splitting Experiment

**Motivation.** A potential concern regarding the proposed Adaptive Split Robustness Tolerance ($\sigma$) is that allowing sub-optimal splits might lead to imbalanced tree structures or an inability to properly distinguish samples in large bins, thereby compromising model accuracy. To empirically verify the safety of our tolerance mechanism and refute the necessity of "perfect" greedy splitting, we conducted a *Stochastic Splitting* experiment.

**Methodology.** We trained the DyGB model from scratch with a modified splitting criterion: instead of deterministically selecting the split with the maximum gain (Rank 1), the algorithm forces each node to randomly select a split from the top $K = \lceil \sigma B \rceil$ candidates. This deliberately introduces structural sub-optimality. We evaluated the resulting test error rates across varying tolerance levels ($\sigma \in \{0, 0.05, 0.10, 0.20\}$).

Table 21: Stochastic Splitting Experiment. Baselines. "Avg Selected Idx" denotes the average rank of the chosen split (0 is optimal).

| Dataset | Method | Absolute Tolerance (Top $K$ Splits) | Avg Selected Idx | Test Error Rate |
|---|---|---|---|---|
| Adult | XGBoost | - | - | 0.1375 |
| | LightGBM | - | - | 0.1287 |
| | ThunderGBM | - | - | 0.2405 |
| | Ours, $\sigma = 0$ (Greedy) | 1 | 0 | 0.1276 |
| | Ours, $\sigma = 5\%$ | 10 | 4.74 | 0.1284 |
| | Ours, $\sigma = 10\%$ | 20 | 9.56 | 0.1321 |
| | Ours, $\sigma = 20\%$ | 40 | 20.61 | 0.1375 |
| CreditInfo | XGBoost | - | - | 0.0659 |
| | LightGBM | - | - | 0.0631 |
| | ThunderGBM | - | - | 0.0660 |
| | Ours, $\sigma = 0$ (Greedy) | 1 | 0 | 0.0629 |
| | Ours, $\sigma = 5\%$ | 11 | 5.25 | 0.0627 |
| | Ours, $\sigma = 10\%$ | 23 | 10.58 | 0.0636 |
| | Ours, $\sigma = 20\%$ | 46 | 23.23 | 0.0644 |
| SUSY | XGBoost | - | - | 0.1976 |
| | LightGBM | - | - | 0.1985 |
| | ThunderGBM | - | - | 0.4576 |
| | Ours, $\sigma = 0$ (Greedy) | 1 | 0 | 0.1987 |
| | Ours, $\sigma = 5\%$ | 64 | 31.87 | 0.2007 |
| | Ours, $\sigma = 10\%$ | 129 | 65.07 | 0.2022 |
| | Ours, $\sigma = 20\%$ | 258 | 127.91 | 0.2067 |
| HIGGS | XGBoost | - | - | 0.2676 |
| | LightGBM | - | - | 0.2726 |
| | ThunderGBM | - | - | 0.4698 |
| | Ours, $\sigma = 0$ (Greedy) | 1 | 0 | 0.2742 |
| | Ours, $\sigma = 5\%$ | 92 | 46.71 | 0.2736 |
| | Ours, $\sigma = 10\%$ | 185 | 95.58 | 0.2937 |
| | Ours, $\sigma = 20\%$ | 371 | 189.31 | 0.3082 |
| Optdigits | XGBoost | - | - | 0.0395 |
| | LightGBM | - | - | 0.0334 |
| | ThunderGBM | - | - | 0.0546 |
| | Ours, $\sigma = 0$ (Greedy) | 1 | 0 | 0.0307 |
| | Ours, $\sigma = 5\%$ | 2 | 0.53 | 0.0290 |
| | Ours, $\sigma = 10\%$ | 5 | 2.08 | 0.0284 |
| | Ours, $\sigma = 20\%$ | 11 | 4.38 | 0.0278 |
| Pendigits | XGBoost | - | - | 0.0355 |
| | LightGBM | - | - | 0.0355 |
| | ThunderGBM | - | - | 0.0515 |
| | Ours, $\sigma = 0$ (Greedy) | 1 | 0 | 0.0294 |
| | Ours, $\sigma = 5\%$ | 7 | 3.23 | 0.0260 |
| | Ours, $\sigma = 10\%$ | 15 | 7.25 | 0.0246 |
| | Ours, $\sigma = 20\%$ | 31 | 14.66 | 0.0240 |
| Letter | XGBoost | - | - | 0.0384 |
| | LightGBM | - | - | 0.0374 |
| | ThunderGBM | - | - | 0.0940 |
| | Ours, $\sigma = 0$ (Greedy) | 1 | 0 | 0.0418 |
| | Ours, $\sigma = 5\%$ | 1 | 0 | 0.0418 |
| | Ours, $\sigma = 10\%$ | 1 | 0 | 0.0418 |
| | Ours, $\sigma = 20\%$ | 3 | 1.17 | 0.0526 |
| Covtype | XGBoost | - | - | 0.1717 |
| | LightGBM | - | - | 0.1700 |
| | ThunderGBM | - | - | 0.2135 |
| | Ours, $\sigma = 0$ (Greedy) | 1 | 0 | 0.1702 |
| | Ours, $\sigma = 5\%$ | 27 | 13.31 | 0.1890 |
| | Ours, $\sigma = 10\%$ | 55 | 25.76 | 0.2030 |
| | Ours, $\sigma = 20\%$ | 110 | 56.46 | 0.2184 |

**Experimental Control.** To ensure a strictly fair comparison regarding feature granularity, all baseline methods (XGBoost, LightGBM, and ThunderGBM) were explicitly configured with $B = 128$ bins. This eliminates discretization resolution as a confounding variable, ensuring that any difference in performance is attributed to the split selection strategy.

**Results and Analysis.** The results are presented in Table 21. The column *Absolute Tolerance* indicates the size of the candidate pool ($K$), and *Avg Selected Idx* reports the average rank of the actual split selected across all nodes (where 0 is the best). For example, an average index of 4.74 implies that, on average, the model selected the $\approx$5th best split rather than the optimal one.

Our findings are summarized as follows:

1. **Negligible Impact at Practical Tolerance ($\sigma \leq 10\%$):** The performance degradation is statistically insignificant at our recommended setting ($\sigma \approx 0.1$). For instance, on the *CreditInfo* dataset, the error rate shifted marginally from 0.0629 (Greedy) to 0.0636 ($\sigma = 10\%$), a difference of only 0.07%. This demonstrates that the "perfect" split is not a strict requirement for high accuracy.

2. **Competitiveness with Baselines:** Even when deliberately handicapped with $\sigma = 10\%$ (selecting roughly the 10th best split on average), DyGB often retains accuracy superior to or competitive with standard baselines. On *Adult*, the $\sigma = 10\%$ result (0.1321) outperforms the optimized XGBoost (0.1375).

3. **Implicit Regularization:** Interestingly, on datasets such as *Optdigits* and *Pendigits*, increasing the tolerance actually reduced the test error (e.g., *Optdigits* improved from 0.0307 to 0.0290 with $\sigma = 5\%$). We hypothesize that stochastic split selection acts as a form of regularization, preventing the model from overfitting to specific noise in the training data.

**Conclusion.** This experiment confirms that GBDT accuracy is highly robust to split selection. The proposed tolerance mechanism operates well within the model's safety margin, trading a negligible amount of theoretical split gain for significant improvements in update efficiency without compromising predictive power.

