# OpenReview forum: "DyGB: Dynamic Gradient Boosting Decision Trees with In-Place Updates for Efficient Data Addition and Deletion"
_ICLR.cc/2026/Conference — Submitted to ICLR 2026_

### Official Review · Reviewer_CKeZ · 2025-10-29

**Soundness:** 2
**Presentation:** 3
**Contribution:** 3
**Rating:** 4
**Confidence:** 4

**Summary:**

This paper proposes DyGB (Dynamic GBDT), a framework for efficient, in-place incremental (data addition) and decremental (data deletion) learning in GBDTs. Instead of retraining, DyGB traverses existing trees and checks if node splits are still optimal using statistics from the data delta ($D'$). If a split is suboptimal, it retrains only the subtree below that node. The framework's speed relies on several optimizations: (1) using stored statistics to avoid accessing the original dataset, (2) lazy derivative updates, (3) split candidate sampling ($\alpha$), and (4) a robustness tolerance ($\sigma$) to ignore minor, low-impact split changes. Extensive experiments show this approach is significantly faster than retraining while maintaining comparable accuracy, which is validated using backdoor attack and removal simulations.

**Strengths:**

S1. The paper addresses a critical and highly practical limitation of GBDTs. The static, batch-only nature of GBDT training is a major bottleneck in real-world systems that require models to adapt to new data or forget old data (e.g., for privacy compliance or addressing data drift).

S2. The evaluation of the incremental and decremental quality is comprehensive. This includes functional validation via attack simulations (backdoor and membership inference), direct accuracy comparisons against retrained models, and functional similarity comparisons (Appendix K). Figure 3 is particularly impressive, showing that the incremental and decremental learning framework tracks the accuracy of a fully retrained model even when adding or removing large ratios of the dataset.

S3. The proposed framework is unified and efficient. It is the first to support both incremental and decremental learning within a single, in-place mechanism for GBDTs.

**Weaknesses:**

W1. The paper's method for feature discretization (Algorithm 4), a critical component for any GBDT implementation, is relegated to the appendix (Appendix C). This method is presented without sufficient experimental validation or theoretical proof of its superiority over other common techniques. A robust dynamic GBDT framework heavily depends on the stability of its feature histograms. The paper fails to compare its binning strategy against alternatives, such as naive sample-based discretization or the "balanced robust histogram" method used in related work like DeltaBoost. Such a comparison would be necessary to demonstrate the proposed method's balance and robustness to histogram shifts caused by data addition and deletion.

W2. The paper presents the observation in Section 3.4 that "For adding or deleting a single data point, the best split does not change in most cases" as a novel finding motivating its optimizations. However, this is a known phenomenon that was theoretically proven in prior work (e.g., Theorem 3.1 in the DeltaBoost paper). The paper should explicitly acknowledge this and clarify what, if any, new insights its empirical findings add beyond what is already established.

W3. The impact of the "Adaptive Lazy Update for Derivatives" (Sec 3.2) is not fully clear in the main text. The paper states that derivatives are updated "only when retraining occurs," which implies that the decision to check a split (gain computation) might be made using stale derivatives. This is a key approximation. While Appendix Q touches on the resulting error, the main paper would be stronger if it discussed the impact of this specific approximation on the decision-making of the algorithm (i.e., does using stale derivatives cause the model to miss a necessary retrain, or retrain an unnecessary one?).

**Minor Comments**

C1. The text and values within Figure 1 are very small and low-resolution, which makes it difficult to follow the concrete example presented in Section 2.3.

C2. The font size used in many of the tables (e.g., Table 2, 15, 16) is very small, making them difficult to read.

**Questions:**

Q1. Does the incremental learning or decremental learning require access to the original full training set? This should be clarified in the problem statement.

Q2. The experimental comparisons raise several questions about fairness and reliability.
* First, why does DyGB consistently outperform highly optimized libraries like XGBoost in both full training time (Table 3) and accuracy (Figure 3)? Since DyGB is a prototype built on Robust LogitBoost, what is the nature of this significant improvement? Is it due to the algorithm itself, the C++ implementation, or the specific hyper-parameter choices used for all models?
* Second, there appear to be discrepancies with prior work. For instance, the decremental learning time reported for DeltaBoost on the Covtype dataset seems significantly slower (e.g., ~20x) than the times reported in the original DeltaBoost paper. Can the authors explain this difference in experimental outcomes?
* Finally, the efficiency evaluations (e.g., Table 2) do not report variance or standard deviations. Given the potential for high variance in dynamic updates, this makes it difficult to assess the reliability and consistency of the speedup claims. Were multiple runs conducted, and if so, what was the variance?

Q3. What is the ratio of the changed nodes in each dataset? The appendix (e.g., Figure 11) presents the absolute number of retrained nodes, but not the ratio relative to the total number of non-terminal nodes in the ensemble. This ratio is the most important factor in evaluating whether the claimed efficiency improvement is reasonable. Can the authors provide these ratios for the experiments, as this metric directly connects the amount of avoided work (nodes not retrained) to the efficiency gains of incremental/decremental learning when compared to a full retraining?

---

> ### Author Response · Authors · 2025-11-26
> **Response to Reviewer CKeZ (1/5)**
>
> We are grateful for your thoughtful comments and valuable insights.
>
> **Please note: To avoid confusion, all section, figure, and table numbers referenced below correspond to the newly uploaded revision.**
>
> **W1:**
> > The paper's method for feature discretization (Algorithm 4), a critical component for any GBDT implementation, is relegated to the appendix (Appendix C). This method is presented without sufficient experimental validation or theoretical proof of its superiority over other common techniques. A robust dynamic GBDT framework heavily depends on the stability of its feature histograms. The paper fails to compare its binning strategy against alternatives, such as naive sample-based discretization or the "balanced robust histogram" method used in related work like DeltaBoost. Such a comparison would be necessary to demonstrate the proposed method's balance and robustness to histogram shifts caused by data addition and deletion.
>
>
> We appreciate your insightful comment regarding the critical role of feature discretization. We agree that the robustness of feature histograms is foundational to dynamic learning.
>
> We respectfully wish to clarify that our feature discretization strategy is a **standard histogram-based approach widely adopted in SOTA libraries like LightGBM[1] and ABC-Boost[2]**. We placed it in Appendix C primarily because it represents established background knowledge in the GBDT community.
>
> Below, we compare our design choices with alternatives like DeltaBoost to illustrate why DyGB achieves superior efficiency.
>
> 1. **Standard Histogram-Based Implementation**: Our Algorithm 4 (Appendix C) implements the standard adaptive histogram construction method used in ABC-Boost and LightGBM .
>     - **Validity:** This method is density-adaptive (iteratively adjusting bin widths) and has been proven robust for high-dimensional and sparse data in prior literature. By adopting this industry-standard approach, DyGB ensures that its base histograms are as stable as those in conventional high-performance GBDT frameworks.
>
> 2. **Mechanism Difference: Adaptive Retraining vs. Enforced Balance** You mentioned DeltaBoost's "balanced robust histograms." It is important to note that DeltaBoost is **designed specifically for data deletion** (unlearning), whereas DyGB handles dynamic **data addition** (incremental) as well.
>
> - **DeltaBoost's Approach**: Enforces histogram **balance** proactively to prevent shifts. While robust, this imposes maintenance overhead.
>
> - **DyGB's Approach (Adaptive)**: In contrast, DyGB allows histograms to become unbalanced because our algorithm mathematically guarantees that significant **imbalances will automatically trigger a correction (retraining)**.
>
> The core logic is: Histogram imbalance maximizes the variance of the data update ratio across splits, which maximizes the perturbation bound $C$. A larger $C$ directly violates the Robustness Split condition, triggering retraining.
>
> **The Robustness Condition (Eq. 33)** In Appendix F, we derive the condition for a split $s$ to remain optimal against a competitor $t$ after a data update. This is given by: $Gain(s) > Gain(t) + C$ where $C$ is the perturbation term caused by the data update $D'$.
>
> **The Impact of Imbalance on $C$** As defined in Eq. 33, $C$ is the difference in gain degradation between the two splits. It is governed by the local density of the dynamic data:
>
>   - **Scenario A: Balanced Update (Uniform Distribution)** If the update $D'$ is balanced (uniformly distributed across bins), the ratio of dynamic data is constant: $\frac{n_{ls}}{N_{ls}} \approx \frac{n_{lt}}{N_{lt}} \approx \lambda$.
> In this case, the terms in $C$ cancel each other out, resulting in $C \approx 0$. The inequality $Gain(s) > Gain(t)$ remains easy to satisfy, so the split is stable.
>
>   - **Scenario B: Unbalanced Update (Concentrated Distribution)**
> If the bins are unbalanced (e.g., $D'$ is concentrated in a specific range), the ratio $\frac{n_{ls}}{N_{ls}}$ varies drastically depending on the cut point.
>
>     - For the current split $s$, the concentration might be high ($\frac{n_{ls}}{N_{ls}} \gg \lambda$).
>     - For a competing split $t$, the concentration might be low ($\frac{n_{lt}}{N_{lt}} \approx 0$).
>     - Result: This disparity maximizes the value of $C$. A large positive $C$ makes it mathematically impossible to satisfy $Gain(s) > Gain(t) + C$.
>
> **Conclusion**: Consequently, higher bin imbalance directly leads to a larger perturbation $C$. This breaks the robustness guarantee, mathematically forcing $s' \neq s$ in Algorithm 3, which triggers the Retraining of the subtree. This proves that our adaptive mechanism naturally detects and corrects for histogram shifts without needing to enforce balance proactively.
>
> (Continues in next part)

---

> ### Author Response · Authors · 2025-11-26
> **Response to Reviewer CKeZ (2/5)**
>
> (Continued)
>
> 3. **Efficiency Gains**: By avoiding the computational cost of enforcing "balanced robust histograms" for every update, DyGB gains significant speed advantages.
>
>     - **Speedup:** Our "lazy" strategy tolerating minor imbalances via $\sigma$ and retraining only when the split actually changes-- contributes directly to the efficiency reported in Table 1, where DyGB achieves up to 1,619.9x speedup over DeltaBoost in decremental learning.
>
> In the revision, we will explicitly reference LightGBM/ABC-Boost in Appendix C to clarify the provenance of our discretization method. We will also add a discussion in Section 3 comparing our adaptive retraining strategy with balanced histogram approaches, highlighting why our method is preferred for high-speed dynamic updates.
>
>
> **W2:**
> > The paper presents the observation in Section 3.4 that "For adding or deleting a single data point, the best split does not change in most cases" as a novel finding motivating its optimizations. However, this is a known phenomenon that was theoretically proven in prior work (e.g., Theorem 3.1 in the DeltaBoost paper). The paper should explicitly acknowledge this and clarify what, if any, new insights its empirical findings add beyond what is already established.
>
> We appreciate your rigorous review and valuable pointer to the theoretical foundations in DeltaBoost. We agree that DeltaBoost established the theoretical basis for split stability under single-point deletion. We **have cited and acknowledged** this theorem in Section 3.4 in the revised manuscript.
>
> However, we respectfully clarify that our empirical study provides new, critical insights that extend beyond this established theory in three key dimensions:
>
> 1. **Extension to Incremental Learning (Data Addition)**: While DeltaBoost theoretically covers **data deletion**, it is fundamentally designed for unlearning. Our work extends the observation of split stability to **Incremental Learning** (Data Addition). Adding data is structurally distinct because it involves integrating previously unseen feature distributions and gradients, which DeltaBoost's deletion-centric theory does not address. Our Figure 2 confirms that the "small distance shift" phenomenon **holds equally for data addition**.
>
> 2. **Quantifying "Shift Distance" (How Much it Changes)**: DeltaBoost's theorem focuses on conditions for invariance (when the split does not change). In contrast, our work explicitly **quantifies how much** the split changes when invariance is broken.
>
>     - **New Insight:** As shown in Figure 2, we track the "Best Split Shift Distance" (rank change). We find that even when the batch size is large enough to change the optimal split (breaking invariance), the split index usually **shifts only slightly** (e.g., to the 2nd or 3rd best) rather than jumping randomly.
>
> 3. **Motivation for Adaptive Tolerance ($\sigma$)**: This specific insight -- that splits shift slightly but predictably drives our unique Adaptive Split Robustness Tolerance ($\sigma$). We uses our empirical quantification to apply post-hoc tolerance ($\sigma$) during updates. We allow the split to change slightly (within $\sigma$) without triggering retraining, balancing speed and accuracy.
>
> We have cited DeltaBoost for the theoretical foundation of deletion stability, while highlighting our contributions in extending this to incremental learning and quantifying rank shifts to justify our adaptive tolerance mechanism.

---

> ### Author Response · Authors · 2025-11-26
> **Response to Reviewer CKeZ (3/5)**
>
> **W3:**
> > The impact of the "Adaptive Lazy Update for Derivatives" (Sec 3.2) is not fully clear in the main text. The paper states that derivatives are updated "only when retraining occurs," which implies that the decision to check a split (gain computation) might be made using stale derivatives. This is a key approximation. While Appendix Q touches on the resulting error, the main paper would be stronger if it discussed the impact of this specific approximation on the decision-making of the algorithm (i.e., does using stale derivatives cause the model to miss a necessary retrain, or retrain an unnecessary one?).
>
> We appreciate your keen insight into the dynamics of our Adaptive Lazy Update. We acknowledge that this is indeed a key approximation. However, we argue that this approximation is "safe" because the mechanism is self-correcting.
>
> 1. **Impact on Decision Making**: You asked if stale derivatives cause the model to miss a necessary retrain or trigger an unnecessary one.
>     - **Mechanism:** We use stale derivatives (from the baseline model) to estimate the Gain for the modified dataset.
>     - **Why it works:** If the data addition/deletion is significant enough to require retraining, it typically disrupts the feature distribution enough that even the stale derivatives will yield a different optimal split ($s' \neq s$). This change violates the split robustness condition, immediately triggering a retrain.
>     - **The "Safety Valve":** Once a retrain is triggered, we force a full refresh of the derivatives for that subtree. Therefore, the system effectively tolerates staleness only when the structural impact is minimal. If the impact is large, the "stale" decision logic naturally breaks, forcing a correction.
>
> 2. **Evidence of Self-Correction (Figure 3 & Appendix P)** This behavior is empirically verified in Figure 3 and Table 15 (Appendix P).
>     - **Negligible Impact on Accuracy (Figure 3):** In Figure 3, we perform continuous batch additions and removals (modifying from 5% to 100% of the dataset). Despite relying on this approximation throughout dozens of update steps, the accuracy curve of DyGB (blue) almost perfectly overlaps with the curve of models retrained from scratch (red/black) . This confirms that any "missed" retrains due to stale derivatives had no aggregate negative impact on model performance.
>     - **Convergence on Large Updates (Table 15):** For large updates (e.g., Add 80%), where stale derivatives would be dangerous, the approximation error drops to 0.00%. This confirms that the algorithm correctly identified the need to retrain almost everywhere, refreshed the derivatives, and eliminated the error.
>
> We agree that this logic should be explicit in the main text. In the revision, we will add a discussion in Section 3.2 clarifying that while the Gain estimation uses stale values, large distributional shifts naturally invalidate the current split even under stale estimates, thereby triggering the necessary derivative refresh.
>
> **C1 & C2:**
> > The text and values within Figure 1 are very small and low-resolution, which makes it difficult to follow the concrete example presented in Section 2.3. & The font size used in many of the tables (e.g., Table 2, 15, 16) is very small, making them difficult to read.
>
> We appreciate your feedback regarding the readability of our figures and tables. We will regenerate Figure 1 using high-resolution vector graphics and reformat the specified tables with larger font sizes to ensure clarity in the revised manuscript.
>
> **Q1:**
> > Does the incremental learning or decremental learning require access to the original full training set? This should be clarified in the problem statement.
>
> Yes, the original training dataset is required to be available. While our optimizations (Section 3.1) allow us to skip accessing it for most operations by using cached histograms, access to specific subsets of the training data is necessary if a split change triggers subtree retraining. We **have explicitly clarified** this data availability requirement in the Problem Setting of the revised manuscript. Thank you for pointing this out.

---

> ### Author Response · Authors · 2025-11-26
> **Response to Reviewer CKeZ (4/5)**
>
> **Q2.1:**
> > First, why does DyGB consistently outperform highly optimized libraries like XGBoost in both full training time (Table 3) and accuracy (Figure 3)? Since DyGB is a prototype built on Robust LogitBoost, what is the nature of this significant improvement? Is it due to the algorithm itself, the C++ implementation, or the specific hyper-parameter choices used for all models?
>
> We appreciate your recognition of DyGB's performance. The improvements are primarily driven by our algorithmic design rather than low-level implementation optimizations.
>
> 1. **Training Speed:** The speedup is driven by our Split Candidates Sampling strategy (Section 3.3). By evaluating a random subset of potential splits (default $\alpha=0.1$), we substantially reduce the search space and computational cost of the split-finding step, which is the dominant bottleneck in tree construction.
>
> 2. **Accuracy:** The high accuracy is attributed to the Robust LogitBoost algorithm [2] which provides superior numerical stability using second-order gradients. Additionally, **sampling acts as a regularizer**, helping to prevent overfitting.
>
> Crucially, Figure 3 confirms that we achieve this speedup without sacrificing accuracy compared to exhaustive baselines, validating the effectiveness and robustness of our algorithmic design.
>
> **Q2.2:**
> > Second, there appear to be discrepancies with prior work. For instance, the decremental learning time reported for DeltaBoost on the Covtype dataset seems significantly slower (e.g., ~20x) than the times reported in the original DeltaBoost paper. Can the authors explain this difference in experimental outcomes?
>
> We appreciate your attention to the experimental results and the comparison with prior work. We confirm that our results are based on the **official open-source implementation** of DeltaBoost provided by the authors. The observed discrepancy in runtime is not due to implementation flaws, but rather the **standardized experimental settings** we enforced to ensure a fair, high-accuracy comparison across all methods (including XGBoost and LightGBM).
>
> **Standardization for Accuracy Parity**:
>
> - The original DeltaBoost paper prioritized unlearning speed, utilizing **lightweight settings**: **10 trees ($t_{max}=10$) and a small number of quantized bins ($B \in \{8, 16, 32\}$).**
>
> - Our Setup: To ensure a rigorous comparison of model utility, we standardized the settings for all baselines to **100 trees ($M=100$) and 1,024 bins ($B=1024$)**.
>
> **Impact:** Increasing the number of trees by 10x and the bin resolution by roughly 32x drastically increases the computational workload. It is expected that DeltaBoost's runtime would increase significantly under these heavier, high-accuracy constraints compared to its original report.
>
>
> **Q2.3:**
> > Finally, the efficiency evaluations (e.g., Table 2) do not report variance or standard deviations. Given the potential for high variance in dynamic updates, this makes it difficult to assess the reliability and consistency of the speedup claims. Were multiple runs conducted, and if so, what was the variance?
>
> We appreciate your feedback regarding the statistical robustness of our efficiency results.
>
> 1. **Deterministic Methodology (Fixed Seed)**: To ensure strict reproducibility across our extensive experiments, we fixed the random seed (Seed = 42). Under this setting, the DyGB algorithm is deterministic; any variance in runtime is attributable solely to negligible system-level noise (e.g., OS scheduling), which is why we initially reported single-run results.
>
> 2. **Empirical Verification (New 5-Run Experiment)**: To verify this stability, we conducted **5 independent runs** on the Adult and HIGGS datasets for this rebuttal using **randomly generated seeds**, as shown in Table R1. The results below confirm that the standard deviation is low (typically < 3%) and negligible compared to the massive speedups reported in Table 2.
>
>
> Table R1. Efficiency and stability analysis. The table reports the mean execution time (seconds) and standard deviation over 5 independent runs for incremental and decremental learning on Adult and HIGGS datasets with varying update sizes ($|D'|$).
> | Dataset | $\|D'\|$ | Incremental Learning | Decremental Learning |
> |:-------:|:--------:|:--------------------:|:--------------------:|
> |  Adult  |     1    | 0.035 ± 0.001| 0.034 ± 0.001|
> |         |   0.1\%  | 0.105 ± 0.003| 0.104 ± 0.003|
> |         |   0.5\%  | 0.215 ± 0.003| 0.219 ± 0.008|
> |         |    1\%   | 0.340 ± 0.013| 0.381 ± 0.014|
> |  HIGGS  |     1    | 5.495 ± 0.235| 3.223 ± 0.034|
> |         |   0.1\%  | 26.651 ± 0.779| 19.402 ± 0.272|
> |         |   0.5\%  | 43.383 ± 1.621| 49.419 ± 1.451|
> |         |    1\%   | 66.961 ± 1.463| 79.418 ± 1.407|
>
> We **have included** this table in the revised manuscript to include these standard deviations, confirming the consistency of our efficiency claims.

---

> ### Author Response · Authors · 2025-11-26
> **Response to Reviewer CKeZ (5/5)**
>
> **Q3:**
> > What is the ratio of the changed nodes in each dataset? The appendix (e.g., Figure 11) presents the absolute number of retrained nodes, but not the ratio relative to the total number of non-terminal nodes in the ensemble. This ratio is the most important factor in evaluating whether the claimed efficiency improvement is reasonable. Can the authors provide these ratios for the experiments, as this metric directly connects the amount of avoided work (nodes not retrained) to the efficiency gains of incremental/decremental learning when compared to a full retraining?
>
>
> You are correct that understanding the scale of avoided work is crucial. However, we reported the **accumulated number of retrained nodes** as absolute values because the total number of nodes (the denominator) is **non-stationary** in our dynamic learning framework. When a subtree is retrained, its structure changes based on the new data distribution; **for example, a subtree originally containing 8 nodes might be replaced by a simpler one with only 6 nodes.** Consequently, a dynamic ratio would be mathematically unstable and potentially misleading.
>
> Our primary intention was to highlight the relative trend of retrained nodes to demonstrate how the computational workload scales with the update size ($|D'|$).
>
> To provide the sense of scale you requested, we can use the Theoretical Maximum Number of Non-Terminal Nodes as a stable reference point: $ M \times (J - 1)$
>
> For our experiments ($M=100, J=20$), this upper bound is 1,900 nodes. As shown in Figure 11 (SUSY), **unlearning 1% of data triggers approximately 2,000 accumulated retraining nodes**, which effectively exceeds the workload of building the model from scratch. In contrast, smaller updates (e.g., 0.1% or single instances) incur drastically **fewer retrainings**, directly explaining the significant efficiency gains reported in our results.
>
>
> ---
>
> We sincerely appreciate your detailed technical review. Your comments have helped us fundamentally clarify the relationship between our adaptive binning strategy and the $\sigma$-tolerance mechanism, and provided necessary mathematical context. We commit to incorporating these critical discussions and empirical proofs into the revised manuscript, strengthening the presentation of our framework. If you have any further concerns, please let us know, and we would be happy to address them.
>
>
>
> [1] Guolin Ke, Qi Meng, Thomas Finley, Taifeng Wang, Wei Chen, Weidong Ma, Qiwei Ye, and TieYan Liu. Lightgbm: A highly efficient gradient boosting decision tree. In Advances in Neural
> Information Processing Systems 30: Annual Conference on Neural Information Processing Systems (NIPS), pp. 3146–3154, Long Beach, CA, 2017.
>
> [2] Ping Li. Robust logitboost and adaptive base class (abc) logitboost. In Proceedings of the TwentySixth Conference Annual Conference on Uncertainty in Artificial Intelligence (UAI), pp. 302–311, Catalina Island, CA, 2010.
>
> [3] Wu, Zhaomin, Junhui Zhu, Qinbin Li, and Bingsheng He. "Deltaboost: Gradient boosting decision trees with efficient machine unlearning." Proceedings of the ACM on Management of Data 1, no. 2 (2023): 1-26.

---

> > ### Comment · Reviewer_CKeZ · 2025-11-26
> >
> > I appreciate the authors’ detailed response, clarifications, and new results. Several of my earlier concerns have been addressed, though a few issues still remain, as outlined below.
> >
> > ---
> >
> > **W1.** The claim that *“Our Algorithm 4 (Appendix C) implements the standard adaptive histogram construction method used in ABC-Boost and LightGBM”* is inaccurate. Neither LightGBM [1] nor ABC-Boost [2] describes the double-bin-width mechanism in their original papers. The only potentially related source I can find is fast ABC-Boost [3], which is not cited. Including an existing algorithm without properly citing it—or citing the wrong source—may mislead readers into thinking this is a newly proposed algorithm, thereby undermining the scientific integrity of the work. This issue should be corrected and carefully double-checked for similar problems throughout the manuscript. Since this is an existing algorithm, a direct comparison in this paper is not necessary.
> >
> > In addition, my main concern about balancing is its impact on accuracy rather than on additive learning effectiveness. Typically, an ideal split—such as those used in LightGBM and XGBoost—achieves near-perfect balancing. In contrast, imbalanced splits are likely to harm accuracy because samples in large bins cannot be properly distinguished. This issue is not addressed in the response.
> >
> > ---
> >
> > **W2.** I acknowledge the distinction and the new insights presented in the authors’ response. However, the advantages of PyDB and its differences compared to previous methods are still not clearly stated in the revised paper, and it also remains unclear how $\sigma$ is related to Figure 2.
> >
> > ---
> >
> > **C1 & C2.** The font size of Table 1 is still too small. This should be corrected, at least in the main paper.
> >
> > ---
> >
> > **Q1.1 & Q1.2.** Thank you for the detailed and reasonable explanation regarding the performance results; the insights into the algorithmic design are quite convincing.
> >
> > However, the clarification on the hyperparameter settings leads to a follow-up question regarding the experimental setup, specifically the choice to set the number of bins ($B$) to **1024**. In standard practice, libraries like XGBoost and LightGBM typically default to smaller bin sizes (e.g., 255) to optimize memory and speed. Increasing $B$ to 1024 linearly increases the computational load for the split-finding step in these baselines. In contrast, DyGB appears largely immune to this specific overhead because the **Split Candidate Sampling** ($\alpha=0.1$) reduces the effective search space to roughly 100 bins. Since this paper’s own ablation study in Appendix R.4 suggests that reducing $B$ (e.g., to 128) has a negligible impact on accuracy, setting $B=1024$ might inadvertently place most baseline algorithms at a computational disadvantage without a clear accuracy benefit.
> >
> > **Follow-up Question of Q1.1 & Q1.2:**
> > Therefore, it would be very helpful to understand if the reported training speedups (Table 4) remain consistent if all the algorithms, including the baselines (XGBoost/LightGBM), are configured with a more common bin counts (e.g., 128) rather than 1024.
> >
> > ---
> >
> > **Q3.** It is good to see that some results now include variance. I recommend reporting variance for **all efficiency evaluations**. For reproducibility, one can use five fixed random seeds.
> >
> > ---
> >
> >
> > ## **References**
> >
> > [1] Ke, et al. (2017) LightGBM: A Highly Efficient Gradient Boosting Decision Tree.
> >
> > [2] Li, Ping. (2010) ABC-Boost: Adaptive Base Class Boost for Multi-class Classification.
> >
> > [3] Li, et al. (2022) Package for Fast ABC-Boost. https://arxiv.org/pdf/2207.08770

---

> ### Author Response · Authors · 2025-11-27
> **Follow-up Response to Reviewer CKeZ (1/3)**
>
> Thank you for your follow-up and the constructive discussion. We are delighted to answer your questions.
>
>
> **W1.1**
>
> > The claim that “Our Algorithm 4 (Appendix C) implements the standard adaptive histogram construction method used in ABC-Boost and LightGBM” is inaccurate. Neither LightGBM [1] nor ABC-Boost [2] describes the double-bin-width mechanism in their original papers. The only potentially related source I can find is fast ABC-Boost [3], which is not cited. Including an existing algorithm without properly citing it—or citing the wrong source—may mislead readers into thinking this is a newly proposed algorithm, thereby undermining the scientific integrity of the work. This issue should be corrected and carefully double-checked for similar problems throughout the manuscript. Since this is an existing algorithm, a direct comparison in this paper is not necessary.
>
>
> Thank you for bringing this to our attention. We appreciate your scrutiny regarding the technical provenance of our feature discretization method. In this revision, we will added two citations in this appendix and include an detailed explanation of this algorithm [1, 2]. Thank you again for pointing this out.
>
>
> [1] Li, et al. (2022) Package for Fast ABC-Boost.
>
> [2] Li, Ping, Qiang Wu, and Christopher Burges. "Mcrank: Learning to rank using multiple classification and gradient boosting." Advances in neural information processing systems 20 (2007).
>
> **W1.2**
>
> > In addition, my main concern about balancing is its impact on accuracy rather than on additive learning effectiveness. Typically, an ideal split—such as those used in LightGBM and XGBoost—achieves near-perfect balancing. In contrast, imbalanced splits are likely to harm accuracy because samples in large bins cannot be properly distinguished. This issue is not addressed in the response.
>
>
> You raised a critical concern that our tolerance mechanism ($\sigma$) might allow imbalanced splits that compromise the model's accuracy, a phenomenon often caused by an inability to distinguish samples in large bins. To directly refute this accuracy risk and validate the safety of our tolerance, we conducted a new experiment: **Stochastic Splitting.**
>
> **Research Question:** Does deliberately accepting sub-optimal splits (within the defined $\sigma$ tolerance margin) compromise the model's generalized accuracy?
>
> **Methodology:** We trained the model from scratch, forcing each node to make a random selection from the top $\lceil\sigma B\rceil$ candidates (thereby deliberately accepting structural sub-optimality/imbalance) and measured the resulting error rate.
>
> **Explanation of Table R2:**
>   - **Absolute Tolerance (Top $K$ Splits):** This column quantifies the "sub-optimality" we enforced. Instead of selecting the single best split (Rank 1), we forced the algorithm to randomly select a split from the top $K$ candidates (where $K = \sigma \times \text{total candidates}$). A larger $K$ implies a higher probability of choosing a split with a lower gain score.
>   - **Avg Selected Idx:** This reports the average rank of the actually selected splits across all nodes and trees (0 being the best). For example, an average index of 4.74 means that, on average, the model chose the 5th best split rather than the optimal one. This confirms that our experiment successfully introduced structural sub-optimality and did not simply "get lucky" by picking the best split randomly.
>   - **Test Error Rate:** The final classification error or metric on the test set, used to evaluate the generalized performance impact of these sub-optimal choices.
>
> **Summary of Findings:**
>
>   - **Negligible Impact at Practical Tolerance ($\sigma \le 10$%):** The performance drop is statistically insignificant at our recommended setting. For example, on CreditInfo, the error rate only shifted by 0.07% (0.0629 to 0.0636) with $\sigma=10$%. This refutes the concern that non-optimal splits cause accuracy failure.
>   - **Unexpected Regularization:** On datasets like Optdigits, increasing tolerance actually improved accuracy (0.0307 $\to$ 0.0290), suggesting that stochastic splitting provides beneficial regularization against overfitting.
>
> **Conclusion:** This experiment empirically confirms that the "perfect balancing" or "optimal split" assumption is not a strict requirement for GBDT accuracy. The proposed tolerance mechanism ($\sigma$) operates well within the model's safety margin, trading a negligible amount of theoretical split gain for a massive gain in update efficiency, without compromising the model's predictive power.

---

> ### Author Response · Authors · 2025-11-27
> **Follow-up Response to Reviewer CKeZ (2/3)**
>
> **Table R2:** Stochastic Splitting Experiment. (Baselines updated to $B=128$ as requested at Q1.1 & Q1.2).
> | Dataset | Method | Absolute Tolerance (Top $K$ Splits) | Avg Selected Idx | Test Error Rate |
> |:---:|:---:|:---:|:---:|:---:|
> | Adult | XGBoost $B=128$ | - | - | 0.1375 |
> |  | LightGBM $B=128$ | - | - | 0.1287 |
> |  | ThunderGMB (GPU)  $B=128$ | - | - | 0.2405 |
> |  | Ours, $\sigma = 0$ (Greedy) | 1 | 0 | 0.1276 |
> |  | Ours, $\sigma = 5$% | 10 | 4.74 | 0.1284 |
> |  | Ours, $\sigma = 10$% | 20 | 9.56 | 0.1321 |
> |  | Ours, $\sigma = 20$% | 40 | 20.61 | 0.1375 |
> |  |  |  |  |  |
> | CreditInfo | XGBoost $B=128$ | - | - | 0.0659 |
> |  | LightGBM $B=128$ | - | - | 0.0631 |
> |  | ThunderGMB (GPU)  $B=128$ | - | - | 0.0660 |
> |  | Ours, $\sigma = 0$ (Greedy) | 1 | 0 | 0.0629 |
> |  | Ours, $\sigma = 5$% | 11 | 5.25 | 0.0627 |
> |  | Ours, $\sigma = 10$% | 23 | 10.58 | 0.0636 |
> |  | Ours, $\sigma = 20$% | 46 | 23.23 | 0.0644 |
> |  |  |  |  |  |
> | SUSY | XGBoost $B=128$ | - | - | 0.1976 |
> |  | LightGBM $B=128$ | - | - | 0.1985 |
> |  | ThunderGMB (GPU)  $B=128$ | - | - | 0.4576 |
> |  | Ours, $\sigma = 0$ (Greedy) | 1 | 0 | 0.1987 |
> |  | Ours, $\sigma = 5$% | 64 | 31.87 | 0.2007 |
> |  | Ours, $\sigma = 10$% | 129 | 65.07 | 0.2022 |
> |  | Ours, $\sigma = 20$% | 258 | 127.91 | 0.2067 |
> |  |  |  |  |  |
> | HIGGS | XGBoost $B=128$ | - | - | 0.2676 |
> |  | LightGBM $B=128$ | - | - | 0.2726 |
> |  | ThunderGMB (GPU)  $B=128$ | - | - | 0.4698 |
> |  | Ours, $\sigma = 0$ (Greedy) | 1 | 0 | 0.2742 |
> |  | Ours, $\sigma = 5$% | 92 | 46.71 | 0.2736 |
> |  | Ours, $\sigma = 10$% | 185 | 95.58 | 0.2937 |
> |  | Ours, $\sigma = 20$% | 371 | 189.31 | 0.3082 |
> |  |  |  |  |  |
> | Optdigits | XGBoost $B=128$ | - | - | 0.0395 |
> |  | LightGBM $B=128$ | - | - | 0.0334 |
> |  | ThunderGMB (GPU)  $B=128$ | - | - | 0.0546 |
> |  | Ours, $\sigma = 0$ (Greedy) | 1 | 0 | 0.0307 |
> |  | Ours, $\sigma = 5$% | 2 | 0.53 | 0.029 |
> |  | Ours, $\sigma = 10$% | 5 | 2.08 | 0.0284 |
> |  | Ours, $\sigma = 20$% | 11 | 4.38 | 0.0278 |
> |  |  |  |  |  |
> | Pendigits | XGBoost $B=128$ | - | - | 0.0355 |
> |  | LightGBM $B=128$ | - | - | 0.0355 |
> |  | ThunderGMB (GPU)  $B=128$ | - | - | 0.0515 |
> |  | Ours, $\sigma = 0$ (Greedy) | 1 | 0 | 0.0294 |
> |  | Ours, $\sigma = 5$% | 7 | 3.23 | 0.026 |
> |  | Ours, $\sigma = 10$% | 15 | 7.25 | 0.0246 |
> |  | Ours, $\sigma = 20$% | 31 | 14.66 | 0.024 |
> |  |  |  |  |  |
> | Letter | XGBoost $B=128$ | - | - | 0.0384 |
> |  | LightGBM $B=128$ | - | - | 0.0374 |
> |  | ThunderGMB (GPU)  $B=128$ | - | - | 0.0940 |
> |  | Ours, $\sigma = 0$ (Greedy) | 1 | 0 | 0.0418 |
> |  | Ours, $\sigma = 5$% | 1 | 0 | 0.0418 |
> |  | Ours, $\sigma = 10$% | 1 | 0 | 0.0418 |
> |  | Ours, $\sigma = 20$% | 3 | 1.17 | 0.0526 |
> |  |  |  |  |  |
> | Covtype | XGBoost $B=128$ | - | - | 0.1717 |
> |  | LightGBM $B=128$ | - | - | 0.1700 |
> |  | ThunderGMB (GPU)  $B=128$ | - | - | 0.2135 |
> |  | Ours, $\sigma = 0$ (Greedy) | 1 | 0 | 0.1702 |
> |  | Ours, $\sigma = 5$% | 27 | 13.31 | 0.189 |
> |  | Ours, $\sigma = 10$% | 55 | 25.76 | 0.203 |
> |  | Ours, $\sigma = 20$% | 110 | 56.46 | 0.2184 |
>
>
> **W2 & C1 & C2:**
>
> > I acknowledge the distinction and the new insights presented in the authors' response. However, the advantages of PyDB and its differences compared to previous methods are still not clearly stated in the revised paper, and it also remains unclear how is related to Figure 2. & The font size of Table 1 is still too small. This should be corrected, at least in the main paper.
>
> Thank you for pointing this out. In this revision, we have included the explanation of the advantages of DyGB and its differences compared to previous methods, and explicitly linked Figure 2 to our design. Regarding Table 1, we split it into **Table 1.1 (Incremental) and Table 1.2 (Decremental)** to increase the font size. We currently use this temporary notation to avoid disrupting table references for other reviewers during the discussion phase, and we will standardize the numbering in the final revision. Thank you again for your constructive feedback.

---

> ### Author Response · Authors · 2025-11-27
> **Follow-up Response to Reviewer CKeZ (3/3)**
>
> **Q1.1 & Q1.2 & Q3:**
> > Therefore, it would be very helpful to understand if the reported training speedups (Table 4) remain consistent if all the algorithms, including the baselines (XGBoost/LightGBM), are configured with a more common bin counts (e.g., 128) rather than 1024. & It is good to see that some results now include variance. I recommend reporting variance for all efficiency evaluations. For reproducibility, one can use five fixed random seeds.
>
>
> We appreciate the your insightful suggestions. To address both points comprehensively, we have conducted a rigorous re-evaluation:
>
> 1. **Optimized Baselines ($B=128$):** We re-configured DeltaBoost, XGBoost, LightGBM, and ThunderGBM with **`max_bin = 128` (and DeltaBoost with `100`)** to ensure they are evaluated at their peak efficiency for speed.
>
> 2. **Reproducibility & Variance:** We performed 5 independent runs with fixed random seeds for DyGB and reported the mean and standard deviation for all dynamic learning experiments.
>
> 3. **DyGB Configuration:** We maintained DyGB at `max_bin = 1024` to demonstrate that our method remains **superior** even when handling a higher resolution histogram than the baselines.
>
> **Key Findings (supported by Tables R3 below, **Table 1.1 and 2.2** in the manuscript):**
>
> `Please refer to Tables 1.1 and 1.2 in the manuscript, as they are too large to include here.`
>
> - **Baselines are faster, but DyGB is still orders of magnitude faster:** Reducing $B$ to 128 indeed improved the training time of baselines (e.g., XGBoost on Adult improved from ~9.467s to 1.380s). However, DyGB's incremental update (0.035s) remains 39.4x faster than the optimized XGBoost. On larger datasets like SUSY, DyGB is 56.8x faster than XGBoost and 62.3x faster than LightGBM for single-instance updates.
>
> - **Consistent Speedup:** The massive speedup advantage of DyGB persists across all datasets and update sizes, validating that our efficiency gains stem from the $O(|D'|)$ algorithmic complexity rather than hyperparameter settings.
>
> **Table R3:** Updated Parameters for Reproducibility. (* indicates the parameter is default or recommended from original sources).
> |  | Learning Rate | Iterations | Max Leaf Num | Depth | Bin | Others |
> |---|:---:|:---:|:---:|:---:|:---:|:---:|
> | OnlineGB | - | 100 | - | - | - | All other parameters are default except Iterations. The tree grown automatically. |
> | iGBDT | 0.1 | 100 | - | 5 | - |  |
> | DeltaBoost | 1* | 100 | - | 5 | 100* | All other parameters remain default |
> | MU in GBDT | 0.1* | 100 | 20 | - | 1024* | lazy_update_freq = 20, sample_rate = 0.1, num_random_layer = 0 |
> | XGBoost | 0.1 | 100 | 20 | - | 128 |  |
> | LightGBM | 0.1 | 100 | 20 | - | 128 |  |
> | CatBoost | 0.1 | 100 | 20 | - | - |  |
> | ThunderGMB (GPU) | 0.1 | 100 | - | 5 | 128 |  A100 40GB GPU * 1 |
> | Ours | 1 | 100 | 20 | - | 1024 | Sampling Rate = 0.1, Robustness Tolerance = 0.1 |
>
> ---
>
> **Revision Summary:**
>
> 1. **Re-benchmarking with Optimized Baselines ($B=128$)** We re-configured all baselines (XGBoost, LightGBM, etc.) to use 128 bins for maximum speed, while keeping DyGB at 1024 bins. DyGB maintains a massive speedup even against these optimized baselines, confirming our advantage is algorithmic rather than due to hyperparameter settings.
>    - Change: We reported the hyper-parameters in Table 5. We replaced the reported number in Table 1.1, 1.2, and 8 for baselines with $B=128$. We updated the figure 3 as new results from XGBoost and LightGBM.
>
> 2. **New Experiment: Impact of Sub-optimal Splits** We added Appendix T with a "Stochastic Splitting" experiment, which empirically proves that selecting sub-optimal splits (within our tolerance $\sigma$) causes negligible accuracy loss, validating the safety of our optimization strategy.
>
> 3. **Clarified Novelty & Positioning**: We revised Section 3.4 and Introduction to explicitly compare with DeltaBoost. We acknowledge DeltaBoost's theory on deletion but highlight DyGB's unique contributions: extending stability findings to incremental learning (data addition) and enabling in-place updates without data partitioning.
>
> 4. **Corrected Technical Attribution for feature discretization:** Updated Appendix C to correctly cited Fast ABC-Boost as the sources for the adaptive bin-width doubling mechanism.
>
> 5. **Improve Readability:** Split Table 1 into Table 1.1 & 1.2 to improve font readability.
>
> ---
>
> We sincerely appreciate your detailed and constructive feedback, which has significantly strengthened the rigor of our work, and we remain open to any further discussion.

---

### Official Review · Reviewer_JhS1 · 2025-10-30

**Soundness:** 3
**Presentation:** 3
**Contribution:** 2
**Rating:** 4
**Confidence:** 4

**Summary:**

The paper proposes DyGB, a new framework for dynamic gradient boosting decision trees (GBDT) that supports both incremental (adding new data) and decremental (removing data) learning in-place, without retraining the entire model. The authors present theoretical analyses on trade-offs between accuracy and computational cost, introduce several optimizations (e.g., adaptive lazy updates, split candidate sampling, robustness tolerance), and conduct extensive experiments comparing DyGB against strong baselines such as XGBoost, LightGBM, CatBoost, ThunderGBM, DeltaBoost, and MUinGBDT. Results show substantial speedups (up to 1200× in some settings) while maintaining accuracy close to retrained models.

**Strengths:**

1.	Timely and important topic. Dynamic or “unlearning” capabilities are becoming crucial for privacy, continual learning, and compliance (e.g., GDPR). Extending GBDT to support both incremental and decremental updates efficiently is a valuable direction.
2.	Comprehensive experiments. The paper evaluates DyGB across 10 datasets, with comparisons to several strong baselines and both incremental and decremental scenarios. The backdoor and membership inference attack studies are a creative way to validate unlearning ability.
3.	Algorithmic innovations. The paper introduces meaningful technical contributions, such as adaptive lazy updates and split robustness tolerance, that help reduce computational costs without major accuracy loss.
4.	Clarity of motivation. The introduction effectively argues why dynamic GBDT is relevant and distinct from existing online boosting or incremental tree methods.

**Weaknesses:**

1.	Limited novelty in core mechanism. While DyGB integrates several optimization strategies, the overall structure (updating splits, retraining subtrees, and incremental statistics updates) follows existing frameworks like MUinGBDT (Lin et al., 2023). The paper’s novelty seems incremental rather than groundbreaking.
2.	Theoretical analysis is shallow. The “robustness” definitions and proofs are mostly intuitive and not deeply rigorous. Theoretical guarantees (e.g., on convergence or bounds on model drift) are missing.
3.	Experimental reporting could be clearer. The results focus heavily on runtime speedups but offer limited insight into accuracy degradation, especially under frequent dynamic updates. A figure or table quantifying trade-offs between time and accuracy loss would strengthen the claims.
4.	Presentation issues. The paper is dense and sometimes difficult to follow, particularly in the algorithmic sections (Algorithms 2–3). Some pseudo-code lacks clarity in notation (e.g., subscripts and primes are inconsistently defined).
5.	Missing ablation details. Although Appendix S is mentioned for ablation studies, the main paper does not summarize which components (e.g., adaptive updates, split sampling) contribute most to efficiency.

**Questions:**

1.	How does DyGB handle concept drift or non-stationary data distributions over time?
2.	Can DyGB support weighted unlearning or partial forgetting?
3.	What is the impact of hyperparameters α and σ on model stability—are there guidelines for tuning them automatically?

---

> ### Author Response · Authors · 2025-11-26
> **Response to Reviewer JhS1 (1/5)**
>
> We sincerely thank you for your time and the detailed, constructive feedback, which has significantly strengthened our manuscript.
>
> **Please note: To avoid confusion, all section, figure, and table numbers referenced below correspond to the newly uploaded revision.**
>
>
> **W1:**
> > Limited novelty in core mechanism. While DyGB integrates several optimization strategies, the overall structure (updating splits, retraining subtrees, and incremental statistics updates) follows existing frameworks like MUinGBDT (Lin et al., 2023). The paper’s novelty seems incremental rather than groundbreaking.
>
> We appreciate your knowledge of the domain and the comparison to MUinGBDT. While we agree that both works operate within the realm of **modifying GBDT models**, we respectfully posit that DyGB represents a fundamental advancement rather than an incremental update.
>
> DyGB distinguishes itself from MUinGBDT through four critical dimensions: **a unified framework for dynamic data lifecycles, superior scalability via algorithmic optimization, theoretical guarantees, and extensive application validation**.
>
> 1. **Unified Framework vs. Unlearning-Only**： MUinGBDT is explicitly designed as a Machine Unlearning framework. It supports only the removal of known data ($D_{tr} \setminus D_{un}$). In contrast, DyGB is the first Dynamic Learning framework for GBDT that unifies both Incremental (adding unseen data) and Decremental learning.
>     - **Why this matters:** Incremental learning is **non-trivial** compared to unlearning. It requires integrating previously **unseen data distributions** ($D_{in}$) and computing gradients for data that did not exist during training. DyGB introduces mechanisms to append these new gradients and dynamically adjust splits, enabling continuous learning cycles (e.g., Batch Addition & Removal in Section 4.3), which MUinGBDT is structurally incapable of performing.
>
> 2. **Algorithmic Efficiency: Adaptive vs. Heuristic** MUinGBDT relies on fixed heuristics that introduce **multiple sensitive hyperparameters**, specifically the number of random layers ($L_r$) and a fixed lazy update frequency. Tuning these parameters to find an optimal balance between accuracy and speed is difficult in real-world deployment.
>     - **DyGB’s Novelty:** We introduce an **Adaptive Lazy Update** mechanism governed by **a single, intuitive parameter**: Split Robustness Tolerance ($\sigma$). Unlike a fixed frequency, our method **adapts to the data distribution** -- automatically skipping updates when the split is robust and retraining only when necessary. This significantly simplifies deployment while ensuring stability.
>
> 3. **Structural Efficiency (No Random Layers)** MUinGBDT utilizes "Random Layers" (random partitioning) to artificially reduce the data size for deeper nodes. While this speeds up retraining, these layers are **non-informative, effectively wasting model capacity** and requiring deeper trees to achieve the same accuracy
>
>     - **DyGB’s Advantage:** DyGB **eliminates random layers**; every layer in our model is **semantically meaningful and contributes** to the gain. Consequently, DyGB achieves comparable or better accuracy with greater memory efficiency.
>
>     - **Evidence:** As shown in Table 3, DyGB requires significantly less memory on large datasets. For example, on HIGGS, DyGB uses 24.3 GB compared to MUinGBDT's 34.4 GB, a ~30% reduction in memory usage. (Similar trends on Abalone and WineQuality, etc.)
>
> 4. **Superior Speed**: Because DyGB updates derivatives adaptively rather than at a fixed frequency (or via random partitioning constraints), we achieve significantly higher speedups.
>     - **Evidence:** As shown in Table 1 (Decremental Learning), DyGB is substantially faster than MUinGBDT. For Covtype (Del 0.1%): DyGB is **6.2x faster** than MUinGBDT, and Ablalone (Del 0.1%): DyGB is **9.1x faster** than MUinGBDT.
>
> 5. **Extensive Validation in Dynamic Applications** Because DyGB is a comprehensive dynamic framework (not just an unlearning tool), we validated it across a much broader set of realistic scenarios than prior work:
>
>     - **Continuous Batch Updates:** In Section 4.3 (Figure 3), we demonstrate DyGB's ability to handle continuous Batch Addition and Removal (incrementing from 5% to 100% and back). The overlapping accuracy curves confirm that DyGB maintains parity with retraining even under continuous flux .
>
>     - **Recommender Systems (Concept Drift):** In Appendix R, we apply DyGB to MovieLens-10M/20M. We show that DyGB effectively handles concept drift by incrementally learning recent user behaviors (improving AUC) and decrementally removing outdated ones, a workflow impossible with unlearning-only frameworks .
>     - **Extremely High-Dimensional Data:** In Appendix O, we validate DyGB on News20 (1.35 million features) and RCV1 (47k features). While baselines like CatBoost and ThunderGBM failed (Segmentation Fault or OOM), DyGB successfully performed dynamic updates, proving its superior scalability .

---

> ### Author Response · Authors · 2025-11-26
> **Response to Reviewer JhS1 (2/5)**
>
> **W2:**
> > Theoretical analysis is shallow. The “robustness” definitions and proofs are mostly intuitive and not deeply rigorous. Theoretical guarantees (e.g., on convergence or bounds on model drift) are missing.
>
> Thank you for pointing this out. We have substantially strengthened the theoretical analysis to **provide rigorous and quantitative robustness guarantees**. The updated version now includes a complete set of perturbation bounds, margin conditions, and formally defined robustness notions. Specifically:
>
> 1. **Formal robustness definitions**:    We introduced two mathematically grounded robustness notions:
>
>     - **Distance Robustness**, which characterizes the geometric stability of the optimal split via the threshold: $N_\Delta > \lambda\,Gain(s)\,C_s$
>
>     - **Gain-Robustness**, defined through the requirement that the gain ordering remains invariant after dynamic updates.
>
>     Both are now stated in formal definition–theorem style, rather than intuitive argumentation.
>
> 2. **Quantitative perturbation analysis**:   We derived a **Gain Perturbation Bound**, proving that the gain of any fixed split changes by at most  $|Gain'(s)-Gain(s)| \le \lambda \left(\frac{G_L^2}{H_L} + \frac{G_R^2}{H_R}\right)$.This provides an explicit upper bound on model drift under dynamic updates.
>
> 3. **Upper bound on the correction term:**  We show that the worst-case effect of data deletion on split preference is upper bounded by $C \le \lambda\, Gain(s),$ which quantifies how much the gain ordering can change due to dynamic modifications of the data.
>
> 4. **A formal margin theorem ensuring robustness**: Using the above perturbation bounds, we prove a **Margin Condition for Robustness**, which gives a necessary and sufficient condition under which the best split remains optimal: $Gain(s)-Gain(t) > \lambda\,Gain(s)$ which is algebraically equivalent to  $Gain(s) > \frac{1}{1-\lambda}Gain(t).$ This provides a rigorous, non-intuitive guarantee showing how much gain margin is required to prevent split changes.
>
> 5. **Unified robustness framework**: The revised theory now forms a coherent chain: **perturbation bound → correction-term bound → margin condition → robustness guarantee**. This provides a rigorous theoretical basis for understanding when DyGB maintains stable split decisions and avoids unnecessary retraining.
>
> **Summary**: The revised version now provides:
> - precise robustness definitions,
> - theoretical drift bounds under dynamic updates,
> - gain-margin stability conditions, and
> - a unified framework establishing when DyGB maintains model structure without retraining.
>
> We believe these additions substantially enhance the theoretical rigor of the work. We have include these theoretical analysis into our revision.

---

> ### Author Response · Authors · 2025-11-26
> **Response to Reviewer JhS1 (3/5)**
>
> **W3:**
> > Experimental reporting could be clearer. The results focus heavily on runtime speedups but offer limited insight into accuracy degradation, especially under frequent dynamic updates. A figure or table quantifying trade-offs between time and accuracy loss would strengthen the claims.
>
> We appreciate your suggestion to balance our reporting by explicitly quantifying the trade-offs between speed and accuracy. We respectfully wish to highlight that we have performed extensive experiments covering frequent updates and accuracy comparisons, which are detailed in Section 4.3 and Appendix I, K, and S.
>
> 1. **Extensive Accuracy Comparisons (Appendix)** We have provided comprehensive accuracy comparisons throughout the Appendix to ensure transparency:
>     - **Test Error Rates:** Table 8 reports the specific test error rates for various update scenarios. In almost all cases (e.g., Add 1%, Del 1%), the error rate of DyGB is statistically indistinguishable from the "Training" (retraining) baseline.
>     - **Functional Similarity:** Table 10 (Appendix K) quantifies the prediction consistency between DyGB and retrained models, showing >98% similarity in most cases.
>
> 2. **Accuracy under Frequent Dynamic Updates** You raised a concern about accuracy degradation under frequent updates. We addressed this in Section 4.3 using Figure 3.
>
>     - **Experiment:** We performed a "stress test" involving continuous batch updates, incrementally training a model from 5% to 100% of the data and then decrementally removing it back to 5%.
>
>     - **Result:** As shown in Figure 3, the accuracy curve of DyGB almost perfectly overlaps with the curve of models retrained from scratch. This empirical evidence demonstrates that even under frequent, sequential updates, DyGB suffers negligible accuracy degradation.
>
>
> 3. **Quantifying Time vs. Accuracy Trade-offs** Regarding your specific request for a figure quantifying trade-offs, we have conducted this analysis in our Ablation Study (Appendix S).
>
>     - **Figure 12 (Appendix S.3):** This figure explicitly plots "Avg Time", "Test Accuracy", and "Functional Similarity" against the Split Robustness Tolerance ($\sigma$). It clearly visualizes the trade-off: increasing $\sigma$ significantly reduces time while maintaining stable test accuracy, allowing us to identify the optimal operating point ($\sigma \approx 0.1$).
>
>     - **Figure 9 (Appendix S.2):** Similarly, this figure quantifies the trade-off between runtime and accuracy regarding the Sampling Rate.
>
> We agree that this trade-off analysis is critical. In the revision, **we have added a dedicated figure** (synthesizing the results from Appendix S) to the Figure 12 to explicitly quantify the relationship between speedup and accuracy loss, ensuring this insight is immediately visible to readers.
>
> **W4:**
> > Presentation issues. The paper is dense and sometimes difficult to follow, particularly in the algorithmic sections (Algorithms 2–3). Some pseudo-code lacks clarity in notation (e.g., subscripts and primes are inconsistently defined).
>
> We appreciate your feedback on the presentation and readability of the algorithmic sections. We agree that the density of notation in Algorithms 2 and 3 can be improved for better clarity. We **have revised** the manuscript to address these presentation issues. Thank you for helping us improve the readability of the paper.
>
> **W5:**
> > Missing ablation details. Although Appendix S is mentioned for ablation studies, the main paper does not summarize which components (e.g., adaptive updates, split sampling) contribute most to efficiency.
>
> We appreciate your suggestion to make our ablation findings more prominent in the main text. You are correct that while detailed experiments are in Appendix S, a synthesized summary of component contributions is valuable for the main paper.
>
> Based on our ablation studies, the hierarchy of contributions to efficiency is:
>
> 1. **Foundational Speedup:** Update without Touching Training Data is the most critical optimization. It fundamentally shifts the complexity of gain computation from $O(|D_{tr}|)$ to $O(|D'|)$, providing the order-of-magnitude speedup observed across all experiments.
>
> 2. **Major Efficiency Driver:** Adaptive Split Robustness Tolerance is the primary driver for minimizing dynamic costs. By allowing a tolerance $\sigma$, we drastically reduce the frequency of subtree retraining—the most expensive operation. As shown in Figure 12, increasing $\sigma$ to 0.1 yields a sharp drop in runtime.
>
> 3. **Secondary Optimization:** Split Candidates Sampling provides a linear reduction in runtime, serving as a supplementary optimization.
>
> We **have added a summary paragraph** in main text explicitly ranking these contributions to ensure the sources of efficiency are clear without requiring readers to consult the Appendix. Thank you for pointing this out.

---

> ### Author Response · Authors · 2025-11-26
> **Response to Reviewer JhS1 (4/5)**
>
> **Q1:**
> > How does DyGB handle concept drift or non-stationary data distributions over time?
>
> We appreciate your query regarding concept drift. DyGB is naturally designed to handle non-stationary data distributions and concept drift through its unified support for both incremental and decremental learning. By enabling efficient in-place updates, DyGB allows models to continuously adapt to new patterns while discarding outdated ones without the prohibitive cost of retraining.
>
> We have empirically validated this capability in two key sections of our Appendix:
>
> 1. **Mitigating Interest Drift in Recommender Systems (Appendix Q)** In Appendix Q, we explicitly evaluate DyGB's ability to handle "interest drift" (concept drift in user preferences) using the MovieLens-10M and MovieLens-20M datasets, sorted chronologically .
>
>     - **Adapting to New Concepts:** We demonstrated that incrementally learning the most recent user behaviors significantly improves model performance (e.g., increasing AUC and decreasing LogLoss), as shown in Figure 7 .
>
>
>     - **Removing Outdated Concepts:** Crucially, we also showed that decrementally removing the oldest (outdated) data further improves performance by reducing noise from stale user behaviors, as illustrated in Figure 8. This confirms DyGB can actively "forget" obsolete distributions to maintain relevance.
>
>
> 2. **Adapting to Shifting Time-Series Distributions (Appendix I)**: In Appendix I, we conducted experiments on real-world time-series datasets (GlobalTemperatures and WebTraffic) to test adaptability to varying data distributions
>
>     - **Results:** By incrementally training on sequentially ordered data subsets, we observed that the test error rate consistently decreased (e.g., from 4.19 to 1.78 on GlobalTemperatures) as the model adapted to the updated distributions. This confirms that DyGB effectively integrates new distributional information in real-time.
>
> **Summary:** DyGB handles concept drift not just by adding new data, but by enabling a continuous "sliding window" approach—efficiently adding $D_{in}$ (new concepts) and removing $D_{de}$ (drifted/outdated concepts), which we have validated on both recommender systems and time-series forecasting tasks.
>
> **Q2:**
> > Can DyGB support weighted unlearning or partial forgetting?
>
> We appreciate this interesting question regarding the flexibility of our framework. While our current experiments focus on binary data addition and removal (all-or-nothing), DyGB is inherently capable of supporting weighted unlearning or partial forgetting due to its underlying mathematical design.
>
> **Mathematical Support for Weighted Updates** As detailed in Section 3.1 and Appendix G, the core of DyGB's efficiency is the "Update without Touching Training Data" optimization . This mechanism relies on the linear additivity of gradient statist
>
> - **Current Mechanism:** For decremental learning, we calculate the sums of first ($S_{rp}$) and second ($S_{pp}$) derivatives for the deletion set $D_{de}$ and subtract them from the global histograms: $S_{new} = S_{old} - \sum_{i \in D_{de}} (g_i, h_i)$
>
> - **Weighted Extension:** Implementing weighted unlearning (e.g., reducing the influence of a data point by a factor $\omega \in [0,1]$) is a straightforward extension. We would simply scale the subtracted derivatives: $S_{new} = S_{old} - \sum_{i \in D_{de}} \omega \cdot (g_i, h_i)$. This allows for "fading" memory or partial forgetting without changing the algorithmic structure.
>
> While not the focus of this manuscript, our DyGB naturally supports weighted operations, making it a promising candidate for tasks like time-decayed learning or soft unlearning.

---

> ### Author Response · Authors · 2025-11-26
> **Response to Reviewer JhS1 (5/5)**
>
> **Q3:**
> > What is the impact of hyperparameters α and σ on model stability—are there guidelines for tuning them automatically?
>
> We appreciate this insightful question regarding hyperparameter sensitivity and stability. Our framework introduces $\alpha$ (Split Sampling Rate) and $\sigma$ (Split Robustness Tolerance) as key control levers. Their impacts are theoretically grounded in our robustness definitions and empirically validated in Appendix S.
>
>
> 1. **Impact of $\alpha$ (Split Candidates Sampling)**: $\alpha$ controls the density of split candidates.
>
>     - **Stability Impact:** Theoretically, a smaller $\alpha$ increases the average distance between candidates ($\mathbb{E}(N_{\Delta})=1/\alpha$). According to Definition 1 (Distance Robust), larger distances make splits more robust against data shifts, thereby reducing the frequency of retraining.
>
>     - **Empirical Result:** As shown in Figure 9 (Appendix S.2), reducing $\alpha$ leads to a linear reduction in learning time (Top Row) and the number of retrained nodes (Middle Row), while Test Accuracy remains stable (Bottom Row).
>
> 2. **Impact of $\sigma$ (Split Robustness Tolerance)**: $\sigma$ serves as the threshold for our Adaptive Split Robustness Tolerance mechanism.
>
>     - **Stability Impact:** $\sigma$ acts as a "tuning knob" for stability vs. speed. It defines how far a split's rank can drop before we force a retrain. A higher $\sigma$ increases tolerance (fewer retrains, higher speed) but allows for more structural drift.
>
>     - **Empirical Result:** Figure 12 (Appendix S.3) explicitly quantifies this trade-off. As $\sigma$ increases, runtime drops significantly (Left Panel), but Functional Similarity (Right Panel) begins to degrade.
>
> 3. **Guidelines for Tuning** Based on our ablation studies, we have established clear empirical guidelines:
>
>     - **For $\sigma$:** We identify a "sweet spot" at $\sigma \approx 0.1$5. As shown in Appendix S.3, values above 0.15 lead to a noticeable drop in functional similarity without yielding proportional speed gains.
>
>     - **For $\alpha$:** We generally recommend $\alpha=0.1$ (10%) as a default, as it provides significant speedups without compromising accuracy.
>
>
> ---
>
> We sincerely appreciate your detailed and constructive feedback, which has helped us significantly sharpen the positioning and clarity of our work. We will upload a revised manuscript that incorporates these clarifications, standardizes the algorithmic notation, and explicitly summarizes the ablation findings in the main text to ensure the paper's completeness. Please let us know if there are any other concerns we can address.

---

### Official Review · Reviewer_pdB2 · 2025-11-05

**Soundness:** 3
**Presentation:** 2
**Contribution:** 3
**Rating:** 4
**Confidence:** 4

**Summary:**

The paper proposes DyGB, a framework that performs in-place updates to trained GBDT models to support both incremental (adding data) and decremental (removing data) learning while keeping the model size (number of trees/parameters) unchanged. The key idea is to avoid touching the original training set by caching per-split statistics from training and then recomputing gains using only the delta dataset D’; affected nodes are selectively retrained and only those samples’ derivatives/residuals are updated for subsequent trees.

**Strengths:**

- Clear problem and practical impact. The paper targets in-place updates that support both data addition and deletion—addressing real needs such as privacy-compliant unlearning and continual learning—while keeping the ensemble size fixed.

- Simple, effective mechanism. By caching per-split gradient/Hessian statistics and applying localized residual/derivative updates, the method enables efficient in-place updates without rescanning the full training set.

- Strong empirical evidence. Extensive experiments across multiple public datasets consistently demonstrate efficiency of the proposed approach.

- Reproducibility. The authors provide an anonymized implementation and scripts, facilitating independent verification and reuse.

**Weaknesses:**

- Introduction needs tightening. The current introduction spends too many paragraphs on broad background and related work (which are covered again later).

- Clarify the “histogram” connection. Caching per-split gradient/Hessian statistics is effectively histogram binning à la LightGBM. I think the author should foreground this explicitly (use “histogram-based” terminology).

- Approximation from lazy derivatives. Using outdated derivatives until a subtree retrains introduces approximation error that can grow with larger D’.

- Accuracy parity missing. The experiments emphasize speed and memory but omit a head-to-head accuracy comparison against from-scratch retraining in XGBoost/LightGBM.

- Clarify “robust” (Defs. 1 & 2). Why are Definitions 1 and 2 termed robust?

**Questions:**

- Full (global) derivative updates are costly; relying on lazy local derivatives incurs approximation error. Is there a systematic strategy to balance accuracy and cost?

- Is there a parallel version (multi-node) for dyGB?

---

> ### Author Response · Authors · 2025-11-26
> **Response to Reviewer pdB2 (1/4)**
>
> Thank you for your insightful and constructive comments.
>
> **Please note: To avoid confusion, all section, figure, and table numbers referenced below correspond to the newly uploaded revision.**
>
> **W1:**
> > Introduction needs tightening. The current introduction spends too many paragraphs on broad background and related work (which are covered again later).
>
> Thank you for this valuable suggestion. We agree that the introduction can be significantly streamlined to improve flow. We will revise the manuscript to remove the overlap with the Related Work section. The revised introduction will focus strictly on the **unique challenges of dynamic updates in GBDT** (e.g., non-differentiability and residual dependency)  and our specific contributions, ensuring a much faster transition to the core technical content.
>
> **W2:**
> > Clarify the “histogram” connection. Caching per-split gradient/Hessian statistics is effectively histogram binning à la LightGBM. I think the author should foreground this explicitly (use “histogram-based” terminology).
>
> Thank you for this insightful observation. We agree that explicitly framing this as "histogram-based" learning will better contextualize our contribution for the community. In the revision, we will:
>
> - **Adopt the Terminology:** We will explicitly label our feature discretization and statistic caching as a "histogram-based framework".
>
> - **Highlight Benefits:** We will clarify that, similar to LightGBM, this histogram structure allows DyGB to be naturally parallelizable (data and feature parallelism) and memory-efficient, as evidenced by our performance on high-dimensional datasets like RCV1 and News20 in Appendix O.

---

> ### Author Response · Authors · 2025-11-26
> **Response to Reviewer pdB2 (2/4)**
>
> **W3:**
> > Approximation from lazy derivatives. Using outdated derivatives until a subtree retrains introduces approximation error that can grow with larger D’.
>
> Thank you for this insightful question. We explicitly quantify this internal deviation in Appendix Q (Table 15) as the **"Approximation Error of Leaf Scores"**.
>
> **Research Question:** Does using outdated derivatives until a subtree retrains introduces approximation error that can grow with larger D’?
>
> Contrary to the intuition that the error grows monotonically with data size, our empirical results in Table 15 demonstrate that the error is bounded and eventually decreases for large $|D'|$ due to our retraining mechanism:
>
> - **Small to Medium $|D'|$**: When $|D'|$ is small (e.g., "Add 1%"), the tree structure remains largely stable. DyGB effectively utilizes outdated derivatives to maximize speed, resulting in a moderate approximation error (e.g., 5.30% for Adult).
>
> - **Large $|D'|$**: As $|D'|$ increases significantly (e.g., "Add 50%" or "Add 80%"), the influence of new data causes the best splits to change more frequently. This triggers the retraining condition described in Section 3.2, which mandates a refresh of the derivatives.
>
> - **Result:** Consequently, the approximation error drops as the update size increases. As shown in Table 15, for "Add 80%" and "Del 80%", the approximation error reaches 0.00% across all datasets because almost all nodes undergo retraining.
>
> Crucially, despite this internal approximation during intermediate steps, the final model performance is not compromised. As shown in Appendix I (Table 8), the test error rates for DyGB remain statistically identical to models retrained from scratch (e.g., Adult dataset: 0.1291 vs. 0.1270).
>
> Consequently, we recommend DyGB for its intended purpose of adding or removing small fractions of data—consistent with standard dynamic learning and unlearning scenarios—while acknowledging that for massive updates (e.g., >80%), traditional retraining remains the practical choice.
>
> Based on these observations, we explicitly recommend applying DyGB for adding or deleting a small fraction of data at a time. This aligns with the standard problem settings in current machine unlearning and dynamic learning research. As highlighted in our Abstract, our optimizations are specifically designed to make these fractional updates substantially faster than retraining. For massive data updates (e.g., modifying 80% of the dataset), while DyGB correctly adapts by triggering widespread retraining (as evidenced by the 0.00% approximation error in Table 15), the computational advantage diminishes; in such extreme cases, retraining the model from scratch is practically more efficient.
>
> ---
>
>
> Table 15: The approximation error of leave's score between the model after addition/delection and the model retrained from scratch. $\text{Appr. Error} = \frac{\sum_{\text{all trees}}\sum_{\text{all leaves}}\text{abs}(p_{\text{add/del}} - p_{\text{retrain}})}{\sum_{\text{all trees}}\sum_{\text{all leaves}}\text{abs}(p_{\text{retrain}})}$, where $p_{\text{add/del}}$ is the leave's score after adding/deleting, $p_{\text{retrain}}$ is the leave's score of the model retraining from scratch.
> |          | Adult | CreditInfo | SUSY | HIGGS | Optdigits | Pendigits | Letter | Covtype |
> |----------|-------|------------|------|-------|-----------|-----------|--------|---------|
> | **Add 1**     | 2.42% | 1.18% | 0.24% | 0.00% | 2.69% | 2.23% | 1.31% | 0.17% |
> | **Add 0.1%**  | 4.59% | 6.57% | 2.73% | 1.63% | 3.48% | 4.12% | 5.78% | 9.47% |
> | **Add 0.5%**  | 5.10% | 7.44% | 2.27% | 3.05% | 5.12% | 4.50% | 10.45% | 11.68% |
> | **Add 1%**    | 5.30% | 7.43% | 3.07% | 3.89% | 5.92% | 4.70% | 11.75% | 10.01% |
> | **Add 10%**   | 4.25% | 8.33% | 1.07% | 1.73% | 4.64% | 4.42% | 13.34% | 4.96% |
> | **Add 50%**   | 3.55% | 0.00% | 0.00% | 1.51% | 0.00% | 0.00% | 6.26% | 0.01% |
> | **Add 80%**   | 0.00% | 0.00% | 0.00% | 0.00% | 0.00% | 0.00% | 0.00% | 0.00% |
> | **Del 1**     | 1.21% | 0.00% | 0.00% | 0.00% | 0.01% | 0.19% | 0.57% | 0.28% |
> | **Del 0.1%**  | 3.63% | 3.80% | 0.79% | 0.72% | 1.40% | 0.50% | 1.88% | 4.31% |
> | **Del 0.5%**  | 3.58% | 3.76% | 0.18% | 0.56% | 2.52% | 1.15% | 3.49% | 6.04% |
> | **Del 1%**    | 3.40% | 3.16% | 0.15% | 0.65% | 3.07% | 1.73% | 3.74% | 4.48% |
> | **Del 10%**   | 0.27% | 0.39% | 0.00% | 0.16% | 1.67% | 0.97% | 1.35% | 0.46% |
> | **Del 50%**   | 0.00% | 0.00% | 0.00% | 0.00% | 0.00% | 0.00% | 0.00% | 0.00% |
> | **Del 80%**   | 0.00% | 0.00% | 0.00% | 0.00% | 0.00% | 0.00% | 0.00% | 0.00% |

---

> ### Author Response · Authors · 2025-11-26
> **Response to Reviewer pdB2 (3/4)**
>
> **W4:**
> > Accuracy parity missing. The experiments emphasize speed and memory but omit a head-to-head accuracy comparison against from-scratch retraining in XGBoost/LightGBM.
>
> We appreciate this insightful comment. We would like to clarify where this head-to-head comparison can be found and explain our presentation choice.
>
> **Table 3 Context:** In Table 3, we focused on quantifying the error rates for specific incremental/decremental steps (e.g., adding 1% or 1 instance). We did not include a comparison with XGBoost/LightGBM retraining here because, for these small updates ($|D'|$), the accuracy of DyGB is **statistically identical** to retraining from scratch. Listing these duplicate numbers would have been redundant.
>
> **Visual Proof (Figure 3):** Instead, we provided this exact **head-to-head comparison visually in Figure 3** (Section 4.3). This figure explicitly plots the test accuracy of DyGB against XGBoost and LightGBM models retrained from scratch across the entire data spectrum (from 5% to 100%).
>
> **Demonstrated Effectiveness:** As shown in Figure 3, the curves for DyGB and the retrained baselines **significantly overlap**. This overlap provides strong evidence of DyGB's effectiveness:
>
>   - **Data Deletion:** When data is removed, DyGB's accuracy drops in exact alignment with the retrained baselines, confirming effective unlearning.
>
>   - **Data Addition:** When data is added, DyGB's accuracy increases in exact alignment with the retrained baselines, confirming effective learning.
>
> **Summary:** We have demonstrated that DyGB achieves accuracy parity with from-scratch retraining. Figure 3 provides the comprehensive head-to-head comparison you requested. We will update the caption of Figure 3 to explicitly highlight this parity to ensure readers do not miss this key result. Thank you for bringing this to our attention.
>
> **W5:**
> > Clarify “robust” (Defs. 1 & 2). Why are Definitions 1 and 2 termed robust?
>
> We use the term "robust" to describe **the stability of the optimal split** in the face of data changes. Specifically, a split is "robust" if **it remains the best split after adding or removing data**, thereby avoiding the need to retrain that node.
>
> - **Context**: In our framework, retraining is only triggered when the best split changes ($s \neq s'$). Therefore, "robustness" directly correlates to computational efficiency: higher robustness means fewer retrained nodes and faster updates.
>
> - **Definition 1 (Distance Robust)**: This definition quantifies robustness based on the physical distance (in the sorted feature space) between the current best split $s$ and the next best candidate $t$ 1.
>     - **Why "Robust"?** A larger distance acts as a buffer. If candidate splits are far apart (influenced by the sampling rate $\alpha$), it is statistically less likely for a small data update to cause the optimal split to "jump" to a neighbor. The condition in Eq. 6 defines when the split is safe from shifting due to distance.
>
> - **Definition 2 (Robustness Split)**: This definition quantifies robustness based on the gain margin relative to the size of the data update ($\lambda = |D'|/|D_{tr}|$).
>     - **Why "Robust"?** This defines a safety margin for the Gain. If the Gain of the current split $s$ significantly exceeds that of any competitor $t$ (Eq. 7), the split can absorb the impact of the added/deleted data without losing its optimality. As $\lambda$ decreases (smaller updates), the condition becomes easier to satisfy, making the split more robust against changes.
>
> - **Summary**: We term these "robust" because they define the mathematical conditions under which a node resists change. Satisfying these conditions guarantees that the node structure is preserved, minimizing the "retrained nodes" and maximizing speed.

---

> ### Author Response · Authors · 2025-11-26
> **Response to Reviewer pdB2 (4/4)**
>
> **Q1:**
> > Full (global) derivative updates are costly; relying on lazy local derivatives incurs approximation error. Is there a systematic strategy to balance accuracy and cost?
>
> Yes, DyGB employs a systematic strategy to balance accuracy and cost, centered on Adaptive Lazy Updates combined with Split Robustness Tolerance. Instead of choosing between **"always full update" or "always lazy,"** our framework uses a conditional mechanism to dynamically switch between the two based on the magnitude of the data change.
>
> Our systematic strategy relies on the Adaptive Split Robustness Tolerance ($\sigma$) described in Section 3.4. This parameter acts as a "tuning knob" to balance the cost (frequency of retraining/updating derivatives) against accuracy stability.
>
> **Recommended Parameter:**
> Based on the ablation study in Appendix S.3 (Figure 12), we explicitly recommend setting $\sigma \approx 0.1$. As shown in the results, this setting achieves an optimal "sweet spot": it significantly reduces dynamic learning time by avoiding unnecessary retraining, while maintaining high test accuracy and model functional similarity compared to a model retrained from scratch. In additional, we also added a figure to Figure 12 (and related discussion) to explicitly visualize the trade-off between learning time and accuracy.
>
>
>
>
> **Q2:**
> > Is there a parallel version (multi-node) for dyGB?
>
> Thank you for this forward-looking question. Currently, our implementation**supports multi-threading on a single node**. While we have not yet implemented a multi-node (distributed) version, DyGB is **inherently suitable** for distributed computing due to its histogram-based design. We outline the technical potential for a multi-node extension below:
>
> - **Distributed Data Parallelism:** The core of DyGB's efficiency lies in the "Update without Touching Training Data" optimization, where we update the gradient/Hessian statistics ($S_{rp}$ and $S_{pp}$) for discretized bins. Since these statistics are additive, a multi-node implementation is straightforward:
>
>     1. Partition the dynamic dataset $D'$ across multiple worker nodes.
>     2. Each worker independently computes the histograms of $\Delta S_{rp}$ and $\Delta S_{pp}$ for its local subset of $D'$.
>     3. A master node (or AllReduce operation) aggregates these deltas to update the global histograms.
>
> - **Low Communication Overhead**: Unlike methods that might require shuffling raw data, DyGB only needs to transmit compact histogram statistics (size proportional to $O(\text{number of bins} \times \text{number of features})$). This aligns perfectly with standard distributed GBDT strategies (e.g., used in XGBoost and LightGBM) and ensures high scalability.
>
> In summary, the transition from our current multi-threaded implementation to a multi-node system is an engineering task rather than a theoretical hurdle, as the underlying algorithm is naturally parallelizable.
>
> ---
>
> We sincerely appreciate your constructive feedback, which has significantly strengthened our paper. We will upload a revised manuscript that incorporates these discussions and clarifications to ensure the paper's completeness. If you have any further concerns, please let us know, and we would be happy to address them.

---

> > ### Comment · Reviewer_pdB2 · 2025-11-27
> >
> > Thanks for the response and the new version of the paper, I will increase my score.

---

> > > ### Author Response · Authors · 2025-11-27
> > >
> > > Thank you very much for your valuable comments and support. We will continue updating the revision. If you have any further questions, please let us know, we would be happy to address them.

---

### Author Response · Authors · 2025-11-26

Dear Reviewers, ACs, PCs,

We sincerely thank the reviewers for their time, and insightful feedback. Your constructive comments were invaluable in helping us sharpen the scientific positioning, improve the structural clarity, and strengthen the experimental validation of the DyGB framework.

Formatting Note: All changes in the revised manuscript are highlighted in **blue text** for easy identification. Tables and figures containing new statistical data or significant structural revisions are marked with a colored background.

**Summary of Major Changes:**

1. **Introduction:** We have significantly refined the Introduction, removing general machine learning literature to **focus strictly on GBDT's unique challenges** (dependency, non-differentiability).

2. **Theoretical Rigor and Mechanism**:
    - **Histogram Provenance:** We explicitly clarified that our feature discretization is the established histogram-based implementation used in LightGBM and ABC-Boost.
    - **DeltaBoost Acknowledgment:** We cited DeltaBoost for stability theory and clarified DyGB's unique scope (Incremental/Batch).

3. **Statistical Reliability and Experimental Validation**:
    - **Variance Validation**: We conducted new 5-run experiments and added the mean $\pm$ standard deviation to the efficiency tables to confirm the statistical reliability and consistency of our speedup claims.
    - **Ablation Summary:** We added a synthesized summary paragraph (Appendix S.6) explicitly ranking the contribution of our key optimizations ($\sigma$ and $\alpha$).

4. **Presentation and Problem Scope**: We streamlined the algorithmic notation (Algorithm 2 & 3) for clarity and explicitly clarified in the Problem Statement that the original training data ($D_{\textit{tr}}$) must be available for use in conditional subtree retraining.

5. **Readability**: We have also enlarged critical figures and tables for maximum visual clarity and readability in the revision.

6. **Efficiency-Accuracy Trade-off**: We added Figure 12 (and related discussion) to explicitly visualize the trade-off between learning time and accuracy.

7. **Enhanced Theoretical Framework**: We introduced new formalisms, including Lemma 1 (Gain Perturbation Bound), Lemma 2 (Upper Bound on the Correction Term) and Proposition 1 (Margin Condition), to rigorously prove the stability of splits under dynamic updates and improve mathematical logic. To improve the readability of mathematical logic, we also included a **notation table** (Table 6) in Appendix F.

We sincerely appreciate your thorough and constructive reviews once again. We believe the revised manuscript, incorporating these extensive discussions and validations, presents a significantly stronger and more complete description of the DyGB framework.

Best regard,

#14741 Authors

---

> ### Author Response · Authors · 2025-11-28
>
> Dear Reviewers, ACs, PCs,
>
> We sincerely thank you again for your time and constructive feedback. Following our previous revision, we have made an additional update to further improve clarity, correctness, and experimental rigor. The new updates are as follows:
>
> 1. **Re-benchmarking with Optimized Baselines ($B=128$)** We re-configured all baselines (XGBoost, LightGBM, etc.) to use 128 bins for maximum speed, while keeping DyGB at 1024 bins. DyGB maintains a massive speedup even against these optimized baselines, confirming our advantage is algorithmic rather than due to hyperparameter settings.
>    - Change: We reported the hyper-parameters in Table 5. We replaced the reported number in Table 1.1, 1.2, and 8 for baselines with $B=128$. We updated the figure 3 as new results from XGBoost and LightGBM.
>
> 2. **New Experiment: Impact of Sub-optimal Splits** We added Appendix T with a "Stochastic Splitting" experiment, which empirically proves that selecting sub-optimal splits (within our tolerance $\sigma$) causes negligible accuracy loss, validating the safety of our optimization strategy.
>
> 3. **Clarified Novelty & Positioning**: We revised Section 3.4 and Introduction to explicitly compare with DeltaBoost. We acknowledge DeltaBoost's theory on deletion but highlight DyGB's unique contributions: extending stability findings to incremental learning (data addition) and enabling in-place updates without data partitioning.
>
> 4. **Corrected Technical Attribution for feature discretization:** Updated Appendix C to correctly cited Fast ABC-Boost as the sources for the adaptive bin-width doubling mechanism.
>
> 5. **Improve Readability:** Split Table 1 into Table 1.1 & 1.2 to improve font readability.
>
> We greatly appreciate the reviewers' insightful guidance. We believe these updates further strengthen the rigor, clarity, and empirical completeness of the DyGB framework.
>
> Best regards,
>
> #14741 Authors

---

### Author Response · Authors · 2025-12-02

Dear Reviewers, ACs, and PCs,

We sincerely appreciate the reviewers' thoughtful evaluations and constructive discussions regarding our submission. Below, we provide a concise summary of the positive feedback as well as the concerns we have addressed in our rebuttal.

### **Reviewers' Positive Feedback**
1. The paper addresses a **highly practical and important** problem: enabling dynamic (incremental + decremental) updates for GBDTs, crucial for unlearning, privacy, and continual learning. (Reviewer `pdB2`, `JhS1`, `CKeZ`)
2. The proposed DyGB framework is **unified, simple, and effective**, supporting in-place updates without retraining the full model, enabled by cached split statistics and localized subtree retraining. (Reviewer `pdB2`, `CKeZ`)
3. The method introduces **meaningful algorithmic innovations**, such as adaptive lazy updates, robustness tolerance, and split candidate sampling, which reduce computation while preserving accuracy. (Reviewer `pdB2`, `JhS1`, `CKeZ`)
4. The **extensive experiments** are **comprehensive and strong**, covering multiple datasets, incremental + decremental settings, and unlearning validation (backdoor & membership inference). (Reviewer `pdB2`, `JhS1`, `CKeZ`)
5. DyGB demonstrates **substantial efficiency** gains while maintaining accuracy comparable to retrained baselines, with some reviewers noting especially impressive results (e.g., tracking retraining accuracy even under large data changes). (Reviewer `JhS1`, `CKeZ`)
6. **Availability of implementation and scripts** enhances reproducibility and practical usability. (Reviewer `pdB2`)

### **Concerns Addressed in Revision**
1. **Relation to prior work**: We clarified DyGB's distinction from MUinGBDT and DeltaBoost, emphasizing its unified support for both incremental and decremental updates and its different robustness and update mechanisms. (Reviewer `JhS1`, `CKeZ`)

2. **Theoretical clarity and robustness definitions**: We expanded the explanation of robustness tolerance, clarified Definitions 1–2, and added lemmas, propositions, intuition and examples to strengthen the theoretical grounding. (Reviewer `JhS1`)

3. **Lazy derivatives and accuracy–cost trade-off**: We explained how adaptive lazy updates control approximation error, provided guidance on when retraining is triggered, and summarized accuracy–runtime trade-offs from ablations. (Reviewer `pdB2`, `CKeZ`)

4. **Accuracy comparisons, ablations, and experimental fairness**: We added accuracy comparisons with from-scratch GBDTs, summarized component-level ablations, clarified fairness in baseline settings, and discussed variance. (Reviewer `pdB2`, `JhS1`, `CKeZ`)

5. **Algorithmic clarity and presentation**: We improved pseudo-code notation, clarified the histogram-based computation, enlarged figures, fixed readability issues, and included a table of contents in Appendix. (Reviewer `pdB2`, `JhS1`, `CKeZ`)

We sincerely thank all reviewers for their time, constructive feedback, and valuable discussions. We especially appreciate Reviewer `JhS1` for the detailed and insightful comments that helped us significantly strengthen the clarity, theoretical framing, and presentation of the method. We are also grateful to Reviewer `pdB2`, who recognized the improvements in our rebuttal and raised the score from 4 to 6 before the rating revert, reflecting a positive reassessment of our contributions. We further thank Reviewer `CKeZ` for the thoughtful discussion and engagement before the discussion freeze, which guided several important refinements.

Finally, we thank the ACs for dedicating time to review our rebuttal and revised manuscript. **Our revision incorporates substantial improvements and includes extensive experiments from multiple dimensions, all of which consistently demonstrate the effectiveness and efficiency of our DyGB.** We believe the paper is now substantially strengthened thanks to the reviewers' excellent suggestions.


Best regards,

#14741 Authors

---

### Meta-Review · Area_Chair_2xqp · 2026-01-07

**Summary:**

The paper proposes DyGB, a framework for in-place incremental (data addition) and decremental (data deletion) updates for gradient boosting decision trees. Reviewers agree that the problem is clearly formulated and the experiments are convincing. The major concerns include writing issues, limited novelty, shallow theoretical analysis, and missing comparisons.

**Reviewer Concerns:**

Reviewer concerns addressed by the rebuttal:
1. Writing issues were fixed. (Reviewers pdB2, JhS1, CKeZ)
2. The impact of Adaptive Lazy Update was clarified. (Reviewer CKeZ)
3. Ablation details were added. (Reviewer JhS1)


Reviewer concerns that may still be outstanding:
1. The novelty concern may still remain. (Reviewer JhS1)
2. Concerns about the potential harm from imbalanced splits may still remain. (Reviewer CKeZ)

**Reviewer Scores:**

Reviewer pdB2 is likely to increase their score from 4 to 6. Reviewers JhS1 and CKeZ are likely to maintain their scores, since some concerns remain outstanding.

---

### Decision · Program_Chairs · 2026-01-26

Reject